# Multivalent coiled-coil interactions enable full-scale centrosome assembly and strength

Manolo U. Rios[1], Małgorzata A. Bagnucka[1]*, Bryan D. Ryder[3]*, Beatriz Ferreira Gomes[2]*, Nicole E. Familiari[1], Kan Yaguchi[1], Matthew Amato[1], Weronika E. Stachera[1], Łukasz A. Joachimiak[3], and Jeffrey B. Woodruff[1]

**The outermost layer of centrosomes, called pericentriolar material (PCM), organizes microtubules for mitotic spindle assembly. The molecular interactions that enable PCM to assemble and resist external forces are poorly understood. Here, we use crosslinking mass spectrometry (XL-MS) to analyze PLK-1-potentiated multimerization of SPD-5, the main PCM scaffold protein in *C. elegans*. In the unassembled state, SPD-5 exhibits numerous intramolecular crosslinks that are eliminated after phosphorylation by PLK-1. Thus, phosphorylation induces a structural opening of SPD-5 that primes it for assembly. Multimerization of SPD-5 is driven by interactions between multiple dispersed coiled-coil domains. Structural analyses of a phosphorylated region (PReM) in SPD-5 revealed a helical hairpin that dimerizes to form a tetrameric coiled-coil. Mutations within this structure and other interacting regions cause PCM assembly defects that are partly rescued by eliminating microtubule-mediated forces, revealing that PCM assembly and strength are interdependent. We propose that PCM size and strength emerge from specific, multivalent coiled-coil interactions between SPD-5 proteins.**

## Introduction

Centrosomes comprise structured centrioles surrounded by a micron-scale, supramolecular mass of protein called pericentriolar material (PCM) that nucleates and anchors microtubule arrays (Conduit et al., 2015; Vasquez-Limeta and Loncarek, 2021; Woodruff et al., 2014). Self-assembly of large coiled-coil proteins, such as SPD-5 (*Caenorhabditis elegans*) and Cnn (*Drosophila melanogaster*) form the underlying PCM "scaffold" (Conduit et al., 2014a; Feng et al., 2017; Fong et al., 2008; Hamill et al., 2002; Megraw et al., 1999; Woodruff et al., 2015). This scaffold recruits "client" proteins that regulate PCM assembly, material properties, and function. The PCM clients γ-tubulin, TPX2, and ch-TOG concentrate α/β tubulin and nucleate microtubules (King and Petry, 2020; Moritz et al., 1995; Roostalu et al., 2015; Wieczorek et al., 2015; Zheng et al., 1995; Zhu et al., 2023). Polo Kinase, SPD-2/Cep192, and Aurora A Kinase potentiate PCM scaffold assembly (Conduit et al., 2010; Conduit et al., 2014a; Haren et al., 2009; Woodruff et al., 2015). PP2A-B55a Phosphatase weakens PCM and promotes its disassembly at the end of each cell cycle in *C. elegans* embryos (Enos et al., 2018; Magescas et al., 2019; Mittasch et al., 2020).

PCM must withstand forces transmitted by the microtubules it nucleates. Astral microtubules emanating from PCM often contact dynein motors anchored on the cell cortex, which can exert a pulling force of several pN (Laan et al., 2012). Motor proteins in the mitotic spindle also generate both pulling and pushing forces. The collective action of these microtubule-bound motors position and elongate the mitotic spindle (Dumont and Mitchison, 2009). Forces exerted on the PCM are primarily cortically directed, revealing that the major stress on PCM is tensile (Farhadifar et al., 2020). During anaphase in *C. elegans* embryos, these forces fracture and disperse the PCM scaffold and accelerate its disassembly (Enos et al., 2018; Magescas et al., 2019). Elimination of microtubules causes PCM compaction and prevents "flaring" (Enos et al., 2018; Magescas et al., 2019; Megraw et al., 2002; Rathbun et al., 2020). Although much is known about the generation of microtubule-mediated forces, very little is known about how PCM resists these forces without fracture. Therefore, to reveal the molecular basis of PCM force resistance, it is important to identify molecular connections that prevent PCM fracture under native microtubule-mediated forces.

SPD-5 and Cnn are each sufficient to multimerize into micron-scale assemblies in a manner potentiated by PLK-1 phosphorylation (Conduit et al., 2014b; Feng et al., 2017; Woodruff et al., 2015, 2017). Cnn contains a central coiled-coil region, termed "PReM," that forms dimers in isolation. Upon phosphorylation by PLK-1, PReM forms homopentamers (Conduit et al., 2014a) or anti-parallel, heterotetramers with

[1]Department of Cell Biology, Department of Biophysics, The University of Texas Southwestern Medical Center, Dallas, TX, USA;   [2]Max Planck Institute of Molecular Cell Biology and Genetics, Dresden, Germany;   [3]Department of Biochemistry, Center for Alzheimer's and Neurodegenerative Diseases, Peter O'Donnell Jr. Brain Institute, University of Texas Southwestern Medical Center, Dallas, TX, USA.

*M.A. Bagnucka, B.D. Ryder, and B. Ferreira Gomes contributed equally to this paper.   Correspondence to Jeffrey B. Woodruff: jeffrey.woodruff@utsouthwestern.edu.

the C-terminal CM2 domain (Feng et al., 2017). Yeast-2-hybrid analyses and pull-downs of SPD-5 fragments showed a similar interaction between the CM2-like domain and a central region containing coiled-coil domains and key PLK-1 phosphorylation sites (a.a. 272–732, which the authors termed the "PReM region") (Nakajo et al., 2022). While these domains are clearly important for PCM assembly, there are likely additional interactions needed to make a micron-scale scaffold. Li et al. (2012) demonstrated that a minimum valence of 3 (i.e., 3 distinct binding domains) is required to achieve three-dimensional protein phases and network structures. Increasing the valence beyond this value further decreases the threshold concentration for network formation. How many motifs contribute to PCM scaffold assembly (valence), their identities (coiled-coil versus disordered regions), and their relative importance are unclear. Nor is it known as to what extent interactions between scaffold proteins determine the overall force resistance of the PCM.

*C. elegans* is ideally suited to address how the molecular interactions underlying PCM give rise to mesoscale properties such as size and strength. First, *C. elegans* PCM is relatively simple, and functional aspects (e.g., assembly and microtubule aster formation) can be reconstituted with a minimal set of components in vitro (Woodruff et al., 2015, 2017). Second, PCM in *C. elegans* embryos undergoes stereotyped transitions between high strength and weakness that are easily visualized by live-cell imaging (Mittasch et al., 2020). Third, it is possible to manipulate PCM proteins and alter force generation mechanisms using genetic and pharmacological interventions.

Here, we mapped putative molecular interactions underlying the SPD-5 scaffold using crosslinking mass spectrometry (XL-MS) and tested their importance for PCM assembly and force resistance in *C. elegans* embryos. Our results revealed that unphosphorylated SPD-5 folds back on itself in the unassembled state and opens up after phosphorylation by PLK-1. The PReM region of SPD-5 contains an alpha-helical hairpin that dimerizes into a tetrameric coiled-coil. Mutations within this region perturbed PCM assembly in vivo, which was partly rescued by eliminating microtubule-mediated pulling forces. Mutating mapped interacting regions in the N-terminus (a.a. 1–566) and a long coiled-coil (a.a. 734–918) of SPD-5 impair PCM assembly in vivo and make PCM weak and susceptible to microtubule-dependent pulling forces. In vitro, these regions are necessary for the micron-scale assembly of SPD-5 scaffolds. We concluded that multivalent coiled-coil interactions between SPD-5 proteins enable both full-scale PCM assembly and force resistance.

## Results

### XL-MS reveals interactions during SPD-5 self-assembly

SPD-5 is essential for PCM assembly in *C. elegans* embryos (Hamill et al., 2002) and sufficient to form micron-scale scaffolds that recruit client proteins and nucleate microtubule asters in vitro (Woodruff et al., 2015, 2017) (Fig. 1 A). Alphafold predicts that SPD-5 contains 14 alpha helices connected by disordered linker regions (Jumper et al., 2021). MARCOIL (50% threshold; MTK matrix) predicts that nine disconnected regions within the

alpha helices have the potential to form coiled-coils (Fig. 1 B). PLK-1 phosphorylation accelerates SPD-5 multimerization invitro and is required for full-scale PCM assembly in vivo (Woodruff et al., 2015). We used mass spectrometry (MS) to identify SPD-5 residues phosphorylated by PLK-1 (Fig. 1 C and Data S1). To reveal how phosphorylation regulates SPD-5 assembly, we performed XL-MS on in vitro SPD-5 scaffolds assembled in the presence of constitutively-active (CA) or kinase-dead (KD) PLK-1 (Fig. 1 D and Materials and methods). We used the zero-length crosslinker DMTMM, which captures interactions between primary amines of lysines and the carboxylates of aspartic or glutamic acids (Leitner et al., 2014). When separated by SDS-PAGE, crosslinked samples migrated as monomeric or multimeric species (MW = 135 kDa versus multiples thereof), which could be excised and analyzed separately by MS (Fig. 1 D). It is possible that a protein in a complex was not crosslinked to its binding partners, in which case it would run as a monomer on an SDS-PAGE gel. Thus, we refer to our samples as "SDS-PAGE monomers" and "SDS-PAGE multimers." The SDS-PAGE monomers contain exclusively intramolecular interactions, while the SDS-PAGE multimers contain both intra- and intermolecular interactions. For all experiments, we performed six replicates and calculated the frequency that crosslinks appeared across replicates, which we term "occurrence."

Analysis of unphosphorylated SPD-5 SDS-PAGE monomers (+PLK-1[KD]) revealed 221 intramolecular crosslinks (Fig. 1 E). Many short-range crosslinks (<10 a.a. in a linear sequence) were highly reproducible and corresponded to the capture of neighboring residues (e.g., K844-E853, E952-K956; occurrence = 5). Highly reproducible long-range crosslinks likely represent structured folds (K75-D293, E601-K621, E605-K621; occurrence = 6). Less reproducible crosslinks (occurrence <2) likely represent transient or weaker interactions (Chen et al., 2019; Hou et al., 2021). The variability in crosslink occurrence suggests that our crosslinking map represents a structural ensemble, where SPD-5 can transition between configurations. Crosslinks primarily connected coiled-coil domains (49% of total crosslinks) or a coiled-coil domain with a non-coiled-coil region (44%) (Fig. 1 E). Only 7% of crosslinks connected two non-coiled-coil regions. These results suggest that unassembled, unphosphorylated SPD-5 folds back on itself.

In the phosphorylated SPD-5 SDS-PAGE monomer sample (+PLK-1[CA]), we identified only 69 intramolecular crosslinks. While many short-range crosslinks remained, almost all long-range crosslinks seen in the unphosphorylated state were not identified (Fig. 1 F). Lost crosslinks typically corresponded to those originating outside the coiled-coils, which shifted the balance of interaction motifs captured: only 35% of crosslinks involved a non-coiled-coil domain versus 51% in the unphosphorylated sample. Thus, PLK-1 phosphorylation limits interactions between coiled-coil domains and non-coiled-coil regions, thus preventing intramolecular folding within SPD-5.

To identify the most physiologically relevant interactions underlying the assembly of the PCM scaffold, we then analyzed SPD-5 SDS-PAGE multimers assembled in the presence of PLK-1(CA). We found 289 crosslinks that were primarily concentrated in numerous coiled-coil domains and not the linkers

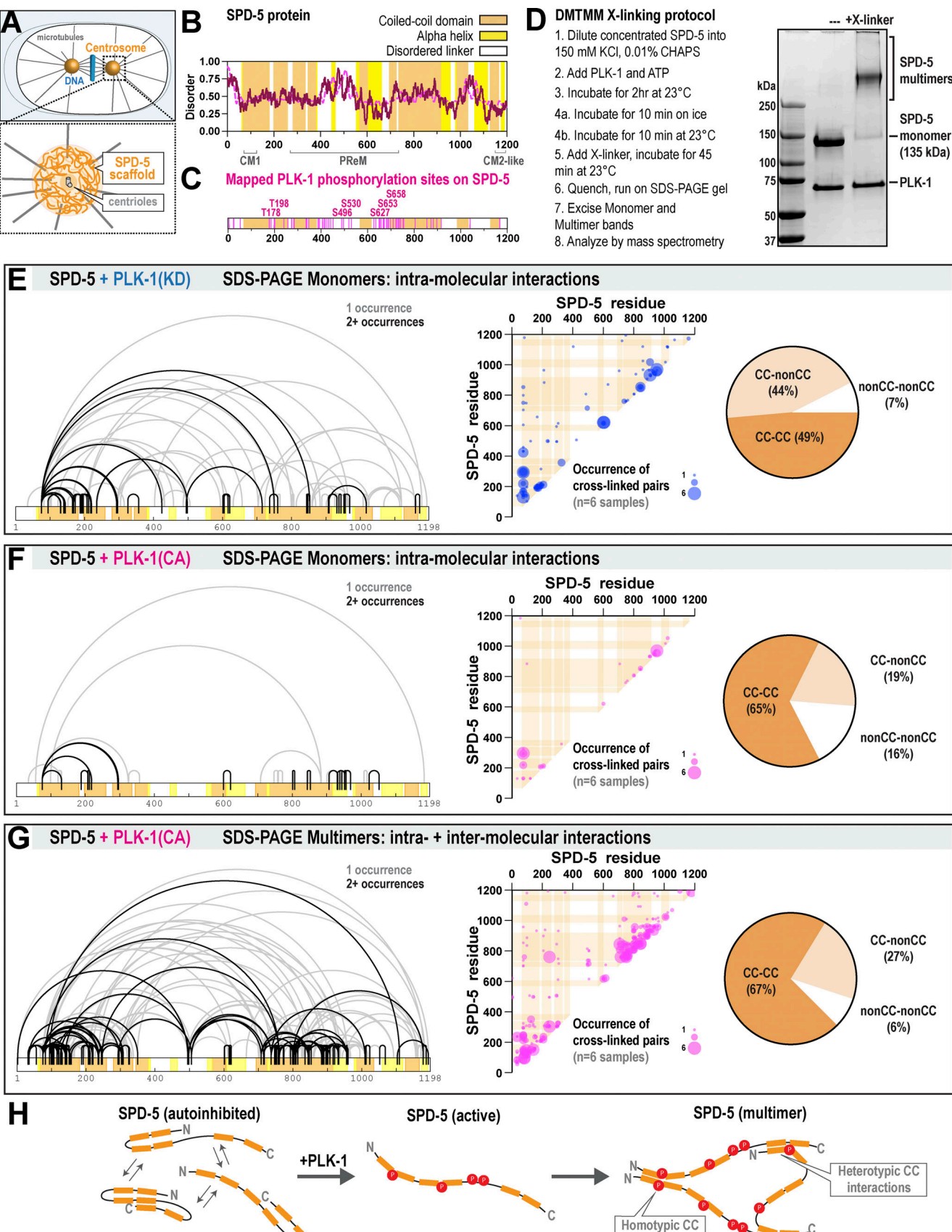

Figure 1. **XL-MS reveals SPD-5 interactions during multimerization. (A)** Centrosome architecture in a *C. elegans* embryo. **(B)** Secondary structural analysis of SPD-5 featuring predicted coiled-coil domains (MARCOIL 50%), alpha helices (Alphafold, >70% confidence), and disorder scores (IUPRED2 score in purple

and ANCHOR score in magenta). **(C)** Phosphorylation sites on SPD-5 (+PLK-1[CA]) identified by MS. All sites are listed in Data S1. **(D)** Flow chart of the DMTMM crosslinking procedure. Samples were split into two at step 4 to enrich for monomers (4a) or multimers (4b) (see Materials and methods). Right panel: A representative gel showing samples incubated with or without crosslinker. SPD-5 that has exclusively intramolecular crosslinks migrate as monomers on a gel (SDS-PAGE monomers). Crosslinked SPD-5 complexes migrate slower (SDS-PAGE multimers). **(E)** XL-MS analysis of SPD-5 SDS-PAGE monomers incubated with kinase-dead PLK-1 (*n* = 6 replicates). Left panel: Linear map of crosslinks. Center panel: Location and occurrence (indicated by bubble size) of crosslinked pairs across replicates. Orange bars indicate predicted coiled-coil domains. Right panel: Percentage of crosslinked pairs involving predicted coiled-coil domains (CC) or other regions (nonCC). All crosslinked pairs are listed in Data S2. **(F)** XL-MS analysis of SPD-5 SDS-PAGE monomers incubated with constitutively active PLK-1. **(G)** XL-MS analysis of SPD-5 SDS-PAGE multimers incubated with constitutively active PLK-1. **(H)** Model for SPD-5 assembly. Unphosphorylated SPD-5 folds back on itself in multiple configurations, which prevents intermolecular interactions. Phosphorylation blocks intramolecular interactions, allowing coiled-coil domains to interact in trans. Interactions can be between the same (homotypic) or different coiled-coil domains (heterotypic). Source data are available for this figure: SourceData F1.

---

(Fig. 1 G). By subtracting out crosslinks that appear in the monomer data sets (i.e., bona fide intramolecular crosslinks), we identified 269 crosslinks (93% of the total) that are likely intermolecular (Fig. S1 A). This data set includes two crosslinks (D349–K1183, E665–K1160) between the PReM-containing (a.a. 272–732) and CM2-like (a.a. 1061–1198) regions. This result is consistent with pull-down experiments showing that PLK-1 phosphorylation enhances the binding between reconstituted PReM and CM2 regions (Nakajo et al., 2022). In addition, this data set revealed additional uncharacterized interaction sites (explored further below). We noticed that most linked residues were close in sequence space, appearing near the diagonal on our occurrence plot (middle panel, Fig. 1 G). Of all crosslinks between coiled-coil domains, 72% were between the same domains, suggesting that coiled-coil domains in SPD-5 can homo-multimerize. There were also several long-range crosslinks, the most prominent being K35-E235, K126-D350, and E247-K760 (occurrence >4), suggesting that certain coiled-coil domains in SPD-5 can also hetero-multimerize.

Unphosphorylated SPD-5 can assemble in vitro, albeit slowly in comparison with PLK-1-phosphorylated SPD-5 (Woodruff et al., 2015). In embryos, phospho-mutant SPD-5, which lacks four key PLK-1 phosphorylation sites, cannot assemble on its own but can bind to preassembled PCM (Wueseke et al., 2016). Thus, SPD-5 can self-interact weakly without the need for phosphorylation. To understand the molecular basis of this interaction mode, we used XL-MS to analyze unphosphorylated SPD-5 SDS-PAGE multimers (+PLK-1[KD]). We identified 677 crosslinks, most of which included at least one coiled-coil domain (Fig. S1 B). Of these, 612 crosslinks (90%) appeared only in the multimer sample and likely represent intermolecular interactions.

By comparing these data sets, we could assess how phosphorylation affects motif preference during SPD-5 scaffold assembly (Fig. S1 C). After normalizing for differences in the overall crosslink number, we found that phosphorylation of SPD-5 increases its preference for coiled-coil interactions and reduces the number of non-coiled-coil interactions (Fig. S1, D and E). The most prominent gains were detected in coiled-coils 1, 2, and 7. The most prominent losses were at residues K191 and K495 residing in predicted disordered regions. The change at residue K191 is noteworthy as it is close to PLK-1 phosphorylation sites (T178 and T198) that regulate the docking of the γ-tubulin complex (Ohta et al., 2021). Interacting residues had overall fewer crosslinked partners in the phosphorylated state

(Fig. S1 F), suggesting that phosphorylation reduces binding promiscuity. Distributions of sequence separation between crosslinked sites were similar between samples, suggesting that phosphorylation does not induce obvious changes in the coiled-coil register (Fig. S1 G).

Our XL-MS data reveal that multiple coiled-coil domains in SPD-5 engage in both intra- and intermolecular interactions. PLK-1 phosphorylation limits the intramolecular folding of SPD-5 and makes these domains accessible to engage other SPD-5 molecules (Fig. 1 H). Our results are consistent with a model wherein PLK-1 phosphorylation of SPD-5 eliminates weaker, less-specific intra- and intermolecular linker contacts to favor more specific and stronger intermolecular crosslinks between coiled-coil domains. Together, these data suggest that dispersed coiled-coil domains are the primary drivers of SPD-5 scaffold assembly. For the remainder of our study, we tested the importance of three newly identified interaction regions for PCM assembly and strength: (1) two helices within the PReM region (a.a. 541–677), (2) a C-terminal, long coiled-coil domain (a.a. 734–918), and (3) four coiled-coil domains in the N-terminus (a.a. 1–566).

**The PReM region of SPD-5 contains a dimerizing helical hairpin**
SPD-5 was originally identified in a screen for temperature-sensitive mutations (Hamill et al., 2002). Expression of one allele (*spd-5(or213)*; R592K) severely inhibits PCM assembly and is 100% embryonic lethal at the restrictive temperature (25°C). At the permissive temperature (16°C), *spd-5(or213)* embryos build diminutive PCM and are sick (68% viability) (Hamill et al., 2002). How a conservative substitution causes such a dramatic PCM phenotype is unknown. R592 lies within the PReM region, close to PLK-1 phosphorylation sites needed for PCM expansion in preparation for mitosis (Ohta et al., 2021; Woodruff et al., 2015). We identified high-occurrence, short-range crosslinks within this region and low-occurrence, long-range crosslinks with other regions of SPD-5 (Fig. 1, D–F). These results suggest that interactions mediated by the PReM region promote PCM assembly.

An AlphaFold-derived model predicts that the PReM region contains two alpha helices connected by a loop, thus forming a helical hairpin (Fig. S2 A). We analyzed the structure and stoichiometry of the predicted hairpin region by reconstituting a fragment of SPD-5 containing residues 541–677 (Fig. 2 A). Circular dichroism revealed that purified SPD-5(541–677) is 72% alpha-helical (Fig. 2 B), similar to 80% predicted by AlphaFold

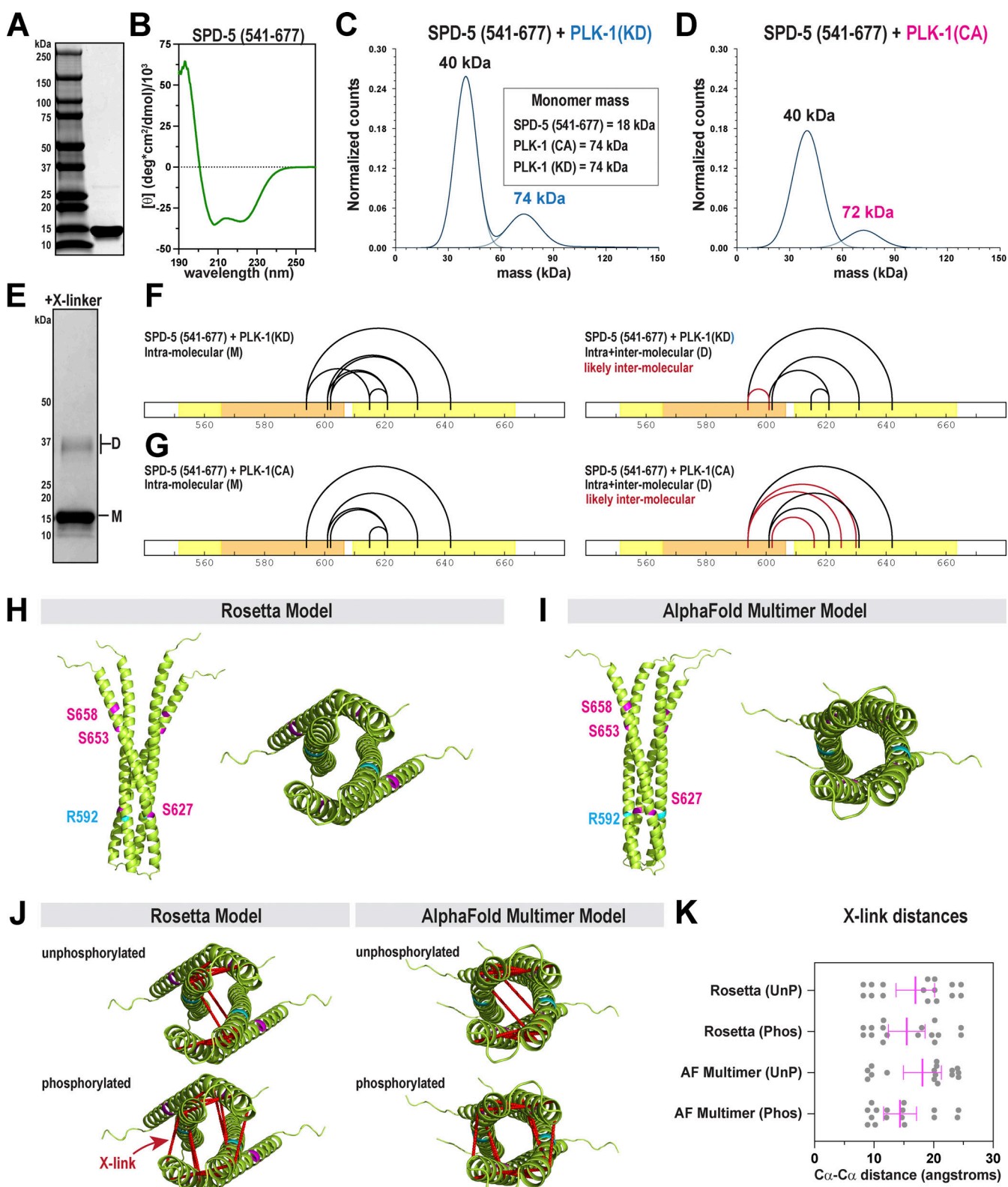

Figure 2. **SPD-5 contains a central helical hairpin motif that homo-dimerizes. (A)** SDS-PAGE gel showing purified SPD-5(a.a. 541–677). **(B)** Circular dichroism of SPD-5(541–677) reveals a strong alpha-helical signature. **(C)** 50 nM SPD-5 (541–677) was incubated with 20 nM kinase dead PLK-1 for 20 min and then analyzed by mass photometry. The inset lists the predicted monomeric masses of each protein. The calculated mass for each species is listed above the curves. **(D)** Mass photometry of SPD-5(541–677) incubated with active PLK-1. **(E)** Subsaturation crosslinking of SPD-5(541–677) reveals monomeric (M) and dimeric (D) species on an SDS-PAGE gel. **(F)** XL-MS of samples of SPD-5(541–677) incubated with PLK-1(KD). Crosslinks found in the monomers represent intramolecular interactions. Crosslinks unique to the dimeric state (red lines) are likely intermolecular. **(G)** XL-MS of samples of SPD-5(541–677) incubated with

PLK-1(CA). **(H)** Best fit Rosetta model for dimeric SPD-5(541–677). PLK-1 phosphorylation sites are shown in magenta. Left, top view; right, en face view. **(I)** AlphaFold Multimer model for dimeric SPD-5(541–677). Left, top view; right, en face view. **(J)** Overlay of experimentally obtained crosslinks onto Rosetta and AlphaFold models. **(K)** Crosslink distances when mapped onto Rosetta and Alphafold models. Distance was measured between alpha carbons of linked residues (n = 18; mean ± 95% C.I.). UnP, links from the unphosphorylated sample (+PLK-1[KD]). Phos, links form the phosphorylated sample (+PLK-1[CA]). Source data are available for this figure: SourceData F2.

(confidence score >70) (Fig. S2 A). Mass photometry revealed that SPD-5(541–677) forms a dimer by itself and in the presence of PLK-1(KD) or PLK-1(CA) (Fig. 2, C and D; and Fig. S2 B). We verified that SPD-5(541–677) is phosphorylated by PLK-1 using gel migration shift and MS (Fig. S1 C). We conclude that this region is largely alpha-helical and homodimerizes in a phosphorylation-independent manner.

Having shown that phosphorylation does not affect the dimerization of the hairpin motif, we then tested if phosphorylation affects its three-dimensional structure. We performed XL-MS on SPD-5(541–677) incubated with PLK-1(KD) or PLK-1(CA). By using subsaturation amounts of crosslinker, we could generate SPD-5(541–677) species that migrate as monomers or dimers on an SDS-PAGE gel (Fig. 2 E), which allowed categorization of intra- and intermolecular crosslinks in the dimer. Intramolecular crosslinks in SPD-5(541–677) were nearly identical in the phosphorylated and unphosphorylated states. We observed differences in intermolecular crosslinks: one link appeared only in the unphosphorylated sample (K594-E601), while three links appeared only in the phosphorylated sample (K594-E625, K594-D630, E602-K616) (Fig. 2, F and G). These results suggest that phosphorylation causes subtle changes in the arrangement of the two helices with respect to each other.

We then used Rosetta ab initio structural predictions to build a three-dimensional model of the hairpin dimer. We tested 10,000 possible docking configurations and identified best-fit models based on two constraints: energy of the dimer assembly and compatibility with the experimental crosslinks, defined by the cumulative cα-cα distance of crosslink pairs for each model (Fig. S2 D). For all samples, one major model class emerged containing flared parallel dimers that form tetrameric coiled-coils near the bend in the hairpin (Fig. 2 H). Due to the limited number of intermolecular contacts in the unphosphorylated hairpin, we could not confidently model structural differences between the phosphorylated and unphosphorylated states. As an independent assessment, we used AlphaFold Multimer, which also predicted that this region forms a tetra-helical bundle with flared ends (Fig. 2 I). This model exhibited high confidence in the central helical bundle and lower confidence at the flared ends (Predicted Alignment Error shown in Fig. S2 E).

We evaluated the Rosetta and AlphaFold models by overlaying the experimentally derived cross-links and checking for distance violations (Fig. 2 J). For all models, cross-links were <24 Å (cα-cα distance) (Fig. 2 K), within the acceptable range for DMTMM as demonstrated previously (Leitner et al., 2014). We conclude that residues 541–677 in SPD-5 constitute an alpha-helical hairpin that dimerizes to form a tetrameric coiled-coil with flared ends. While phosphorylation of SPD-5 does not influence the dimerization of the hairpin, it could alter its connectivity.

## Residues within the helical hairpin are essential for PCM assembly and strength

We addressed the importance of the helical hairpin for PCM assembly and strength using mutational analysis and confocal microscopy of embryos. We hypothesized that PCM strength and assembly could be interdependent: if a structural deficiency causes PCM weakness, then pulling forces would deform and prematurely rupture PCM. This could disassemble PCM, giving the false impression of failed assembly. Thus, for a true PCM weakness phenotype, we should be able to rescue PCM shape, integrity, or mass by eliminating pulling forces.

We used immunofluorescence to visualize PCM assembly in spd-5(or213) embryos which express SPD-5(R592K) from the endogenous genomic locus. We compared one-cell embryos that had begun or completed nuclear envelope breakdown. At 25°C, SPD-5 localized strongly to centrosomes in wild-type but very weakly in spd-5(or213) embryos, as expected (Fig. 3 A) (Hamill et al., 2002). Western blotting revealed similar levels of full-length SPD-5 in control and spd-5(or213) embryos grown at 25°C; thus, the PCM assembly defect is not due to lower expression levels or protein degradation (Fig. S3 A). Application of nocodazole rescued SPD-5(R592K) accumulation at centrosomes fivefold compared with the DMSO-treated controls. However, nocodazole did not fully restore centrosome size to wild-type levels (Fig. 3, A and B). Thus, SPD-5(R592K) could build small mitotic PCM, but it was too weak to withstand physiological pulling forces.

To characterize the assembly and strength defects of SPD-5(R592K) more carefully, we used live-cell imaging of an independently generated mutant. We used MosSCI to create transgenic worms expressing either wild-type or R592K versions of SPD-5 tagged with GFP (Fig. 3 C). These transgenes are RNAi-resistant, allowing the knockdown of endogenous SPD-5 and observation of mutant phenotypes (Fig. S3 B). Both control and mutant constructs were expressed at similar levels (Fig. S3 A). In spd-5(RNAi) embryos, GFP::SPD-5(WT) built a full-sized, functional PCM that was spherical in shape, as expected (Fig. 3, C and D). GFP::SPD-5(R592K) localized weakly to centrioles but was not sufficient to expand the PCM. In GFP::SPD-5(R592K) embryos, treatment with nocodazole increased PCM mass 10-fold compared with the DMSO controls (Fig. 3, C and D). Thus, the R592K mutation weakens the SPD-5 scaffold, making it susceptible to premature disassembly by microtubule-mediated pulling forces. However, nocodazole did not completely restore PCM mass to wild-type levels, indicating that the R592K mutation disrupts SPD-5 scaffold assembly, even in the absence of forces. Unexpectedly, unlike spd-5(or213) embryos, the MosSCI mutant line was not temperature sensitive: embryos expressing

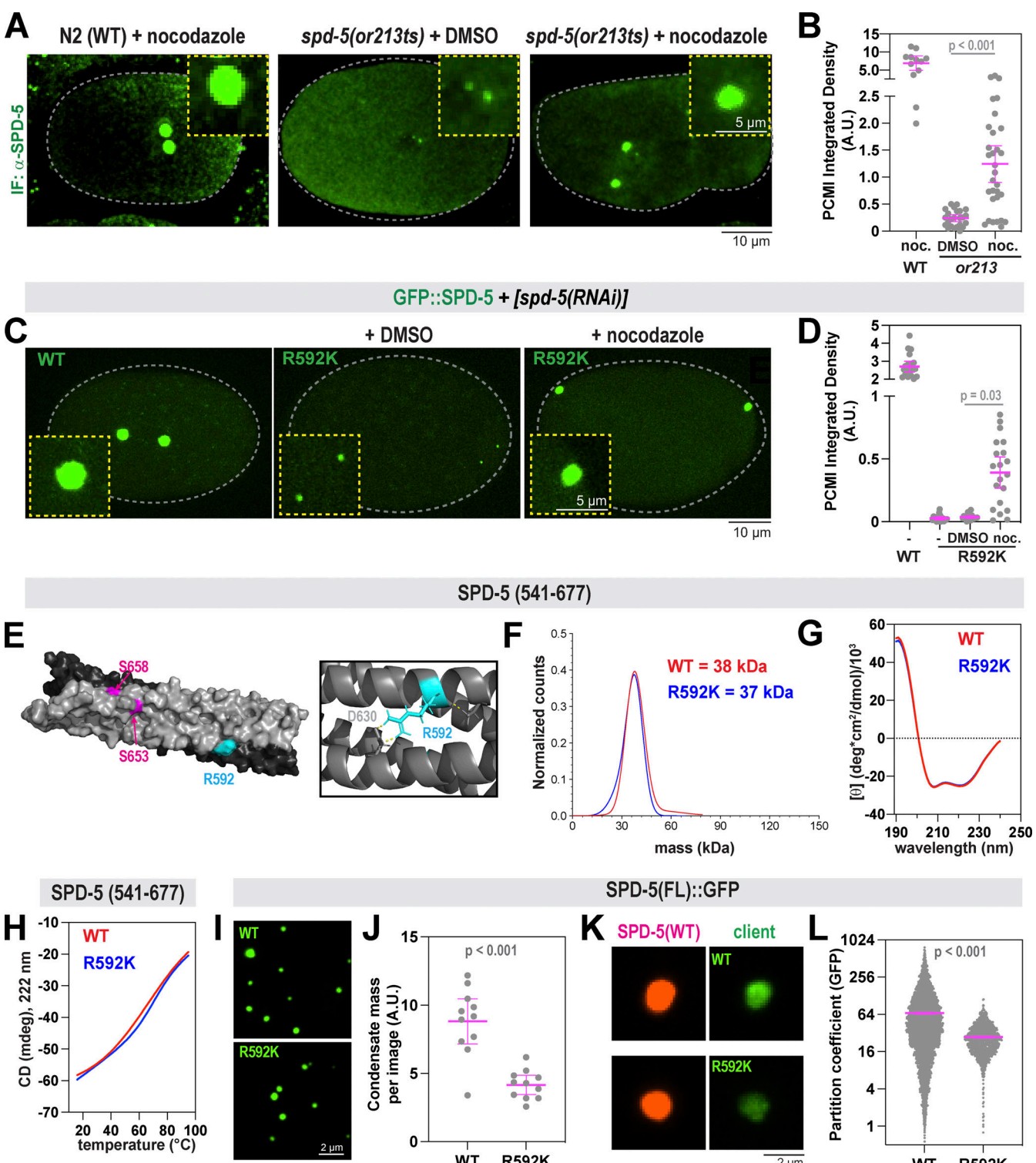

Figure 3. **R592 in SPD-5 is essential for full-scale PCM assembly and strength. (A)** Wild-type (N2) or *spd-5(or213 ts)* worms were grown at 25°C for 16 h, then embryos were extracted and exposed to DMSO or 20 µM nocodazole for 2 min, then fixed. Embryos were permeabilized with *perm-1(RNAi)*. SPD-5 was detected using immunofluorescence. Insets show zoomed-in images of centrosomes. **(B)** Quantification of fluorescence integrated density of SPD-5 signal localized at PCM. Mean ± 95% C.I.; *N* = 12–31 centrosomes; P value from a Kruskal–Wallis test. **(C)** MosSCI-generated transgenes (GFP::SPD-5, WT or R592K) were expressed from chromosome II and are re-encoded to be resistant to RNAi knockdown of endogenous *spd-5*. Worms were treated with RNAi against endogenous SPD-5 for 24 h, then embryos were excised and imaged by fluorescence confocal microscopy. Embryos were permeabilized using light pressure. Insets show zoomed in images of centrosomes. **(D)** Quantification of fluorescence integrated density of SPD-5 signal localized at PCM in C. Mean ± 95% C.I.; *N* = 36–40 centrosomes; P value from a Kruskal–Wallis test. **(E)** Surface-rendered model of the SPD-5(541–677) dimer. Key residues for one side are labeled. Inset: Diagram of SPD-5 structure surrounding R592; dashed yellow lines represent hydrogen bonds. **(F)** 50 nM of wild-type (WT) or mutant (R592K) SPD-5(541–677) was analyzed by mass photometry. The calculated molecular weights are indicated. **(G)** CD spectroscopy of WT and R592K SPD-5(541–677). **(H)** Thermal

denaturation of WT and R592K SPD-5(541–677) was measured by CD spectroscopy at 222 nm. **(I)** 500 nM of purified full-length SPD-5(WT)::GFP or SPD-5(R592K)::GFP were incubated in 150 mM KCl + 7.5% PEG (MW: 3,300) for 10 min to form condensates, then imaged with fluorescence confocal microscopy. **(J)** Quantification of total condensate mass per field of view in I. Mean ± 95% C.I.; N = 11 images; P value from Mann–Whitney test. **(K)** 1,000 nM of SPD-5(WT)::RFP was assembled into condensates for 2 min, then 10 nM of the client (SPD-5(WT)::GFP or SPD-5(R592K)::GFP) was added, incubated for 8 min, and then imaged. **(L)** Quantification of client partitioning into preassembled WT condensates in K. Mean ± 95% C.I. (N = 4,702 (WT) and 1122 (R592K) condensates; P value from Welch's *t* test.

only SPD-5(R592K) were inviable at 16 and 25°C (Fig. S3 C). The basis of this difference is unclear. Transgenic SPD-5(R592K) contains a GFP tag and is expressed at slightly lower levels compared with SPD-5(R592K) in *spd-5(or213)* embryos (Fig. S3 A), which could decrease viability at 16°C. Nevertheless, analyses of both mutant strains revealed that R592 is essential primarily for full-scale PCM assembly and secondarily for PCM strength.

How does R592K disrupt SPD-5 scaffold assembly at the molecular level? Our structural models predict that R592 is exposed to the solvent and should only participate in hydrogen bonding with neighboring residues (Fig. 3 E). In silico analysis of energetics suggests that a change to lysine at this position should not dramatically affect the stability of the hairpin dimer (Fig. S3 D). To test this idea, we purified SPD-5(541–677) harboring the R592K mutation and assessed its stability and dimerization capacity. Mass photometry and crosslinking revealed that both wild-type and mutant proteins are dimeric (Fig. 3 F and Fig. S3 E). We observed dimers for both species even at the lowest concentrations of protein observable by mass photometry (10 nM), indicating a relatively high affinity. CD spectra and thermal denaturation profiles were similar between wild-type and mutant SPD-5(541–677) (Fig. 3, G and H). Thus, R592K does not detectably affect the dimerization or stability of the helical hairpin, consistent with predictions from our structural models. The thermal denaturation profiles revealed a linear transition, rather than a non-cooperative transition, suggesting that the dimer lacks an extensive hydrophobic core. One interpretation is that the SPD-5(541–677) dimer has a dynamic core that is not ideally packed.

Since R592 is found on the surface of the hairpin dimer, we wondered if it constitutes a binding interface to promote the multimerization of full-length SPD-5. This theory is supported by our XL-MS data of full-length SPD-5, which revealed long-range crosslinks emanating from the helical hairpin region (Fig. 1). Therefore, we purified full-length, GFP-tagged SPD-5 (WT or R592K) and performed scaffold assembly and recruitment assays in vitro. 500 nM SPD-5(WT)::GFP formed micron-scale, spherical condensates in the presence of 7.5% PEG (3,000 MW), as reported previously (Woodruff et al., 2017). SPD-5(R592K)::GFP also formed spherical condensates, but the total mass of condensates per image was ~twofold lower compared with WT (Fig. 3, I and J). For the recruitment assay, we first assembled condensates using RFP-labeled SPD-5(WT) (1,000 nM) and then added GFP-labeled variants (10 nM). Recruitment of the R592K mutant was, on average, ~2.5-fold lower than WT (Fig. 3, K and L). Thus, R592, which resides within the helical hairpin motif, is important for full-scale assembly of SPD-5 scaffolds in vitro. We conclude that the R592K mutation directly affects the assembly of the SPD-5 scaffold without affecting the local structure of the helical hairpin.

As an alternative way to disrupt the helical hairpin, we used CRISPR to mutate residues 610–640, which constitute the second helix in this region. We made the deletion in a background strain expressing SPD-5 tagged with RFP at its endogenous locus (RFP::SPD-5[WT]). Control embryos built two centrosomes that amassed spherical PCM, which cohered until rupturing and fragmenting in late anaphase, when PCM normally disassembles (Fig. 4 A). Embryos expressing RFP::SPD-5(Δ610–640) were viable but formed diminutive PCM that was distorted in shape and prematurely fractured (Fig. 4, A–D and Fig. S4 A). RFP-positive fragments emanated from the main PCM body in *spd-5(Δ610–640)* embryos prior to anaphase, indicating premature PCM rupture. This phenotype was not due to poor expression of mutant SPD-5, as Western blotting revealed that the mutant protein was expressed at a higher level than the control protein (Fig. S4 B). Nor was this phenotype due to elevated pulling forces, as spindle rocking was similar between mutant and control conditions (Fig. S4 C). In *spd-5(Δ610–640)* embryos, the elimination of microtubule-mediated pulling forces via nocodazole treatment partially rescued the PCM assembly defect (1.8-fold increase) and completely rescued the PCM sphericity and premature rupture phenotypes (Fig. 4, A–D). These data indicate that PCM built from SPD-5(Δ610–640) is materially weak and cannot withstand microtubule-mediated pulling forces during spindle assembly, consistent with our observations of embryos expressing SPD-5(R592K). We conclude that residues within the helical hairpin are critical for SPD-5 to assemble into a strong scaffold that resists microtubule-mediated forces. Our data also indicate that PCM scaffold weakness manifests as impaired assembly (strongest phenotype), premature rupture, or a decrease in sphericity (weakest phenotype).

### A long, C-terminal coiled-coil domain is essential for SPD-5 assembly and strength

We then characterized the other interaction hotspots in SPD-5 identified by XL-MS. In the SPD-5 multimer sample (+PLK-1 [CA]), we mapped 119 intermolecular crosslinked pairs where at least one residue resided within a predicted coiled-coil domain spanning a.a. 734–918, which we termed "CC-Long." To test the importance of CC-Long for PCM assembly and strength, we used CRISPR to delete a.a. 734–918 in a strain expressing RFP::SPD-5. Embryos expressing RFP::SPD-5(Δ734–918) were viable (Fig. S5 A), but PCM was smaller and more irregular in shape compared with the control (Fig. 5, A–D). PCM also fragmented prematurely in *spd-5(Δ734–918)* embryos (Fig. 5, B and E). These phenotypes were not due to reduced expression of the mutant constructs (Fig. S5 B). Nocodazole treatment increased the mean intensity of PCM-localized RFP::SPD-5(Δ734–918) and restored the spherical shape of the PCM scaffold in metaphase (Fig. 5, A–C). These

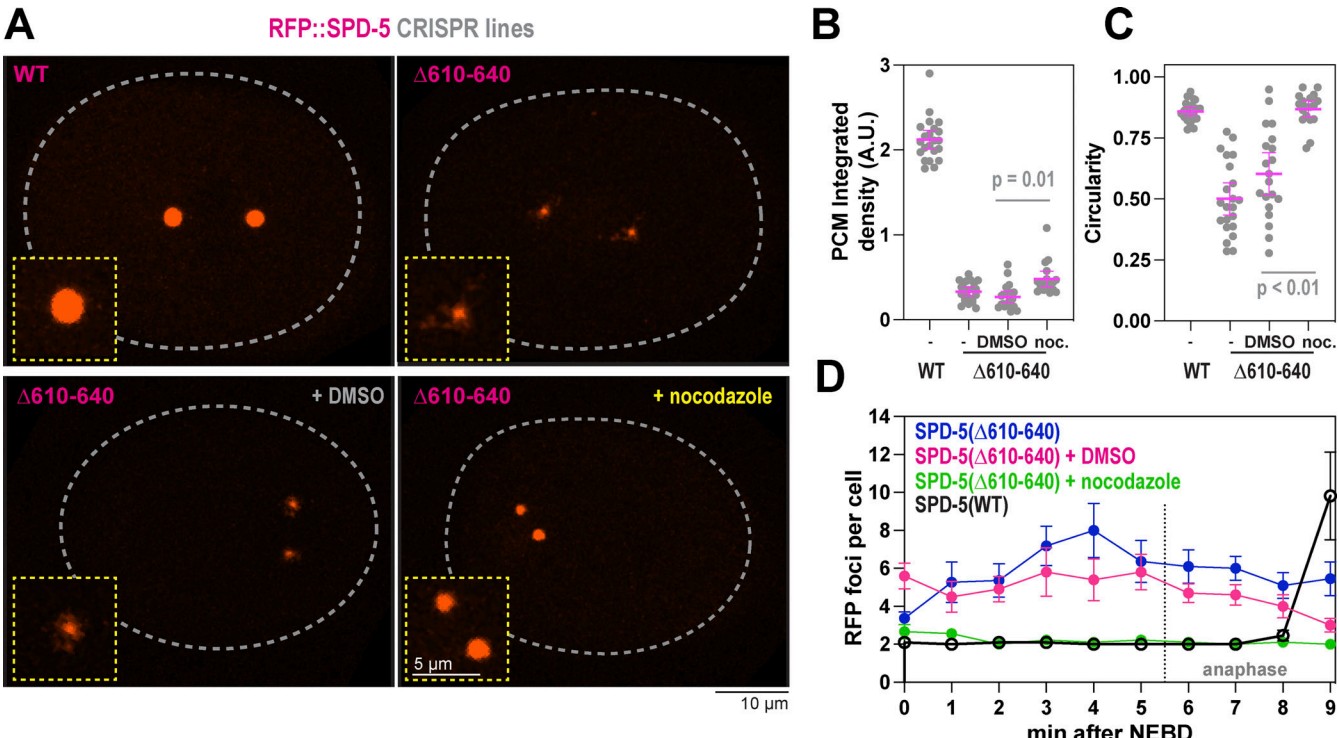

**Figure 4. Residues within the helical hairpin are essential for PCM assembly and strength. (A)** Endogenous RFP-tagged SPD-5 was modified by CRISPR to delete residues 610–640. Embryos expressing the unmodified (WT) and mutant (Δ610–640) RFP::SPD-5 were visualized with fluorescence confocal microscopy. Embryos were permeabilized with *perm-1(RNAi)* to allow entry of DMSO or 20 µM nocodazole. **(B)** Quantification of fluorescence integrated density of RFP signal localized at PCM. Mean ± 95% C.I.; *N* = 18–22 embryos; P value from a Kruskal–Wallis test. **(C)** Quantification of PCM circularity. Mean ± 95% C.I.; *N* = 18–22 embryos; P value from a Kruskal–Wallis test. **(D)** PCM fragmentation from nuclear envelope breakdown onward in one-cell embryos. Mean ± 95% C.I.; *N* = 9–11 embryos.

data indicate that irregular PCM morphology is caused by microtubule-mediated pulling forces, suggesting that loss of CC-Long leads to material weakness in the PCM scaffold. While nocodazole treatment rescued PCM circularity, it only increased PCM assembly ~1.4-fold compared with the DMSO control, which was well below wild-type levels (Fig. 5 D). Thus, CC-Long is required for full-scale SPD-5 assembly even in the absence of pulling forces. This phenotype was less severe compared with that seen with Helical Hairpin mutations, suggesting that interaction domains are not equivalent.

Since ablating pulling forces rescues PCM circularity and assembly defects in *spd-5(Δ734–918)* embryos, we next asked whether increasing pulling forces would worsen the phenotype. RNAi knockdown of casein kinase 1 gamma (CSNK-1), which increases cortical pulling forces by ~1.5-fold (Panbianco et al., 2008), exacerbated the PCM fragmentation phenotype in embryos expressing SPD-5(Δ734–918) but not SPD-5(FL) (Fig. 5, E and F). Thus, PCM built with SPD-5(Δ734–918) is too weak to stay connected under physiological pulling forces. We conclude that CC-Long is required for the SPD-5 scaffold to self-assemble and generate enough strength to resist microtubule-mediate pulling forces.

**The N-terminus of SPD-5 is essential for full-scale PCM assembly and strength**

The N-terminus of SPD-5 (a.a. 1–566) contains four predicted coiled-coil domains, the first of which contains a CM1-like region that recruits the γ-tubulin complex (Ohta et al., 2021) (Fig. 6 A). Our XL-MS of phosphorylated SPD-5 multimers identified 135 intermolecular crosslinked pairs where at least one residue resided within a.a. 1–566. Thus, we hypothesized that this region could mediate SPD-5 scaffold assembly and strength. We used MoSCI to create a transgenic version of SPD-5 missing the N-terminus (GFP::SPD-5[566–1198]) and compared it with transgenic full-length SPD-5 (GFP::SPD-5[FL]) (Fig. S6, A and B). These transgenes are resistant to RNAi knockdown of endogenous *spd-5*. After RNAi depletion of endogenous *spd-5*, GFP::SPD-5(566–1198) formed the PCM that was 17-fold less massive than in control embryos (Fig. 6, B and C). Under these conditions, mutant embryos were inviable (Fig. S6 A). We concluded that the N-terminus of SPD-5 is essential for full-scale PCM assembly and embryo development.

Treatment with nocodazole had no effect on PCM assembly in embryos expressing GFP::SPD-5(566–1198). This result is expected since the N-terminus of SPD-5 anchors microtubules though the γ-tubulin complex (Ohta et al., 2021); thus, pulling forces are not transmitted and are not expected to strain the PCM. We reasoned that if SPD-5(566–1198) is defective in generating strength, it might disrupt the integrity of PCM built with endogenous SPD-5. In embryos expressing endogenous SPD-5, GFP::SPD-5(566–1198) localized to PCM but did not cause PCM fragmentation or premature disassembly (Fig. 6 F and Fig. S6 C). To test whether increasing pulling forces would reveal a

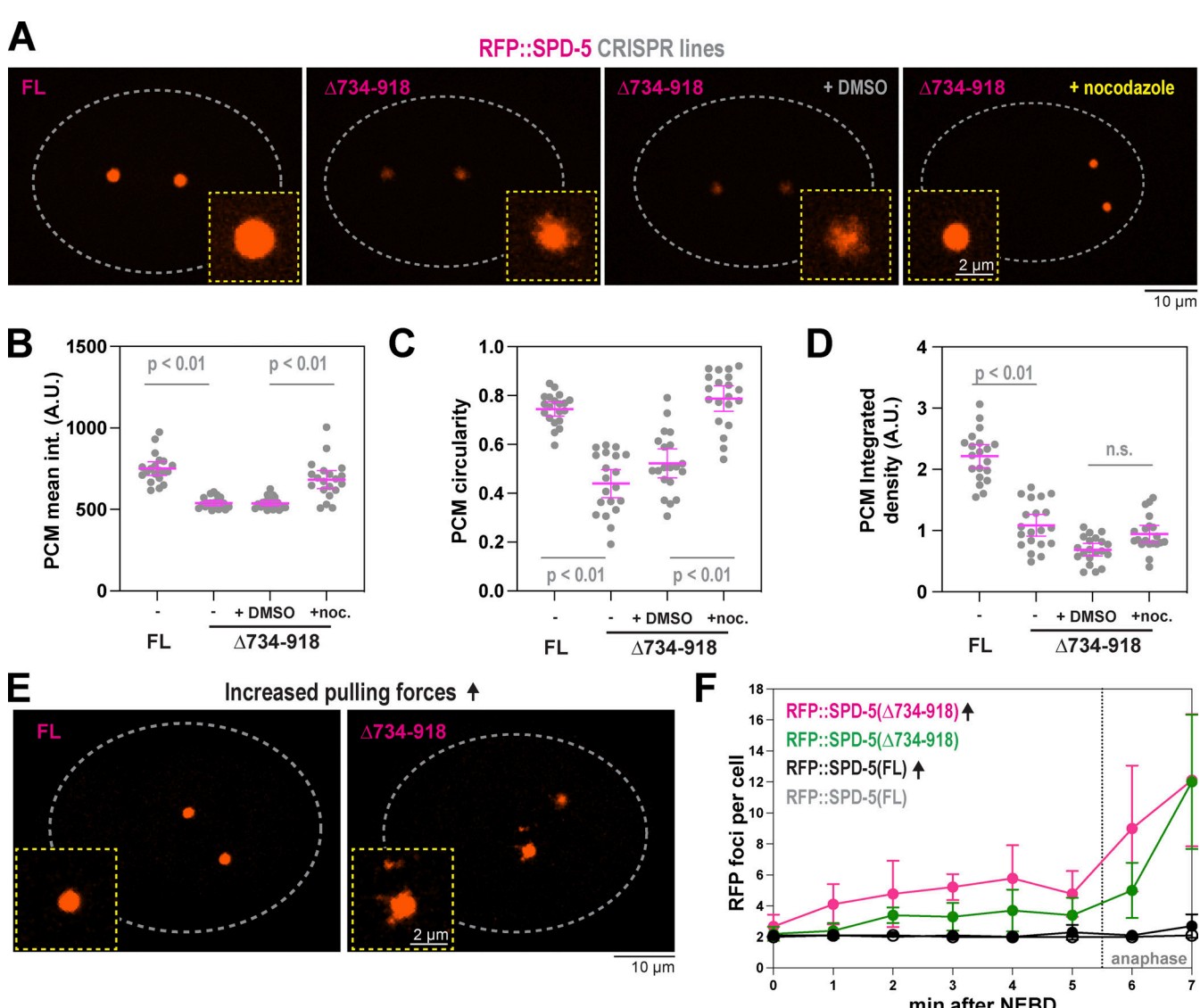

Figure 5. **A long central coiled-coil motif in SPD-5 is essential for PCM assembly and strength. (A)** Endogenous RFP-tagged SPD-5 was modified by CRISPR to delete residues 734–918. Embryos expressing the unmodified (WT) and mutant RFP::SPD-5 were visualized with fluorescence confocal microscopy. 20 μM nocodazole or 1% DMSO were introduced via laser puncture of the eggshell. **(B)** Quantification of mean intensity of RFP signal localized at PCM during metaphase. Mean ± 95% C.I.; N = 20 centrosomes; P value from a Kruskal–Wallis test. **(C)** Quantification of PCM circularity during metaphase in indicated strains. Mean ± 95% C.I.; N = 20 centrosomes; P value from a Kruskal–Wallis test. **(D)** Quantification of fluorescence integrated density of RFP signal localized at PCM during metaphase. Mean ± 95% C.I.; N = 20 centrosomes; P value from a Kruskal–Wallis test. **(E)** Embryos were treated with RNAi against *csnk-1*, which increases microtubule-mediated pulling forces. Insets show zoomed in images of centrosomes. **(F)** PCM fragmentation in one-cell embryos measured from nuclear envelope breakdown onward. Mean ± 95% C.I.; N = 10 embryos. Arrows indicate embryos were treated with *csnk-1* RNAi.

phenotype in this mixed scenario, we depleted CSNK-1 using RNAi. PCM prematurely ruptured and became less spherical in *csnk-1(RNAi)* embryos expressing GFP::SPD-5(566–1198), while no effect was observed for control embryos (Fig. 6, D–F). We concluded that SPD-5(566–1198) interferes with the proper assembly of endogenous SPD-5, resulting in structurally weak PCM. Our results suggest that interactions mediated by SPD-5's N-terminal helical domains are essential to achieve full-scale PCM assembly and strength.

We then tested if the N-terminus is sufficient to assemble into supramolecular scaffolds. We used MosSCI to create embryos expressing N-terminal SPD-5 (a.a. 1–566) tagged with GFP

(Fig. 6 G; and Fig. S6, A and B). In the absence of endogenous SPD-5, GFP::SPD-5(1–566) was not sufficient to form PCM in one-cell embryos or rescue viability, consistent with a prior study (Fig. 6 G and Fig. S6 A) (Nakajo et al., 2022). However, in further developed multicell embryos, GFP::SPD-5(1–566) assembled into foci (Fig. 6 G). This phenomenon showed cell-to-cell variability and was most prominent in embryos depleted of endogenous SPD-5 (Fig. 6 H). We noticed that transgene expression increased after the knockdown of endogenous SPD-5 and during development (Fig. S6, D and E). Thus, we speculate that SPD-5(1–566) assembly in vivo is highly sensitive to protein concentration and only occurs after crossing a critical threshold.

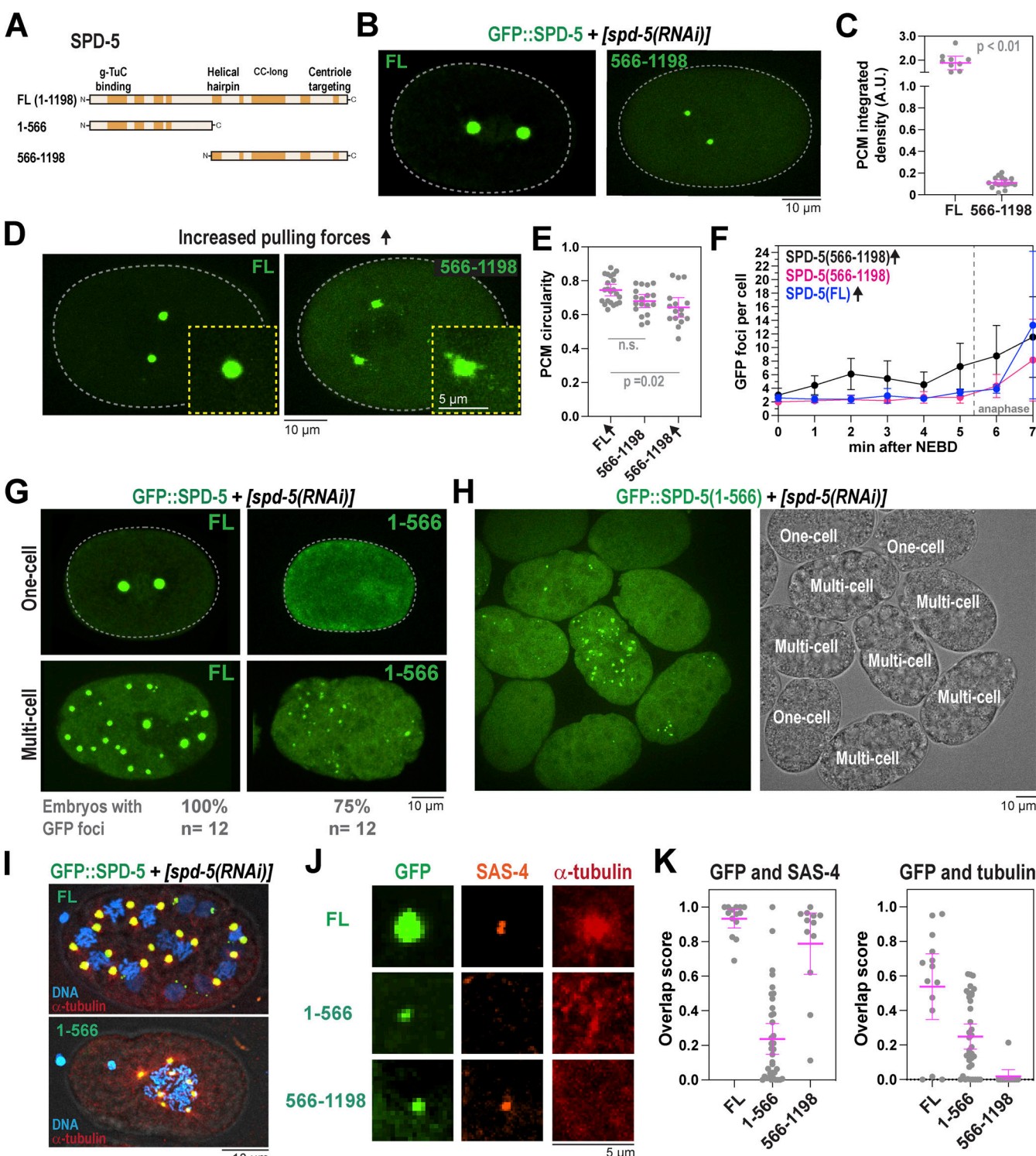

Figure 6. **The coiled-coil-containing N-terminus of SPD-5 is required for PCM strength and assembly. (A)** Diagram of SPD-5 domains and truncation mutants. Predicted coiled-coil domains (MARCOIL, 50%) are indicated in orange. g-TuC, gamma-tubulin complex. **(B)** Fluorescence confocal images of embryos expressing full-length (a.a. 1–1198; FL) or truncated (a.a. 566–1198) *spd-5* transgenes after knockdown of endogenous SPD-5 by RNAi. **(C)** Quantification of fluorescence integrated density of GFP signal localized at PCM during metaphase. Mean ± 95% C.I.; *N* = 10 (FL) and 17 (566–1198) centrosomes; P value from a Mann-Whitney test. **(D)** Images of *csnk-1(RNAi)* embryos expressing endogenous *spd-5* and transgenic *gfp::spd-5*. **(E)** Quantification of PCM circularity during metaphase in B and D. Arrows indicate embryos were treated with *csnk-1* RNAi. Mean ± 95% C.I.; *N* = 16–22 centrosomes; P value from a Kruskal–Wallis test. **(F)** PCM fragmentation from nuclear envelope breakdown onward in one-cell embryos. Mean ± 95% C.I.; *N* = 6–10 embryos. Arrows indicate embryos treated with *csnk-1* RNAi. **(G)** Images of one-cell and multicell embryos expressing full-length (FL) or truncated SPD-5 (a.a. 1–566). The number of multicell embryos displaying GFP-positive foci is indicated below. **(H)** Embryos expressing GFP::SPD-5(1–566) in the absence of endogenous SPD-5. Fluorescence image on the left, DIC image on the right. Cells are labeled based on developmental stage. **(I)** Immunofluorescence of transgenic embryos stained for α-tubulin and DNA.

Embryos were lightly fixed to preserve the native GFP fluorescence. **(J)** Immunofluorescence of transgenic embryos stained for a centriolar marker (SAS-4) and α-tubulin. **(K)** Left: Overlap scores between SAS-4 and GFP::SPD-5 signal. Right: Overlap scores between α-tubulin and GFP::SPD-5 signal. Mean ± 95% C.I.; $n$ = 12–36 GFP-positive foci.

Immunofluorescence revealed that GFP::SPD-5(1–566) foci could nucleate microtubules and did not localize with the centriolar marker SAS-4, indicating that these structures represent semifunctional, PCM-like assemblies, but not true centrosomes (Fig. 6, I–K). Endogenous SPD-5 was not detected at GFP::SPD-5(1–566) foci, indicating efficient RNAi knockdown (Fig. S6 F). Conversely, GFP::SPD-5(566–1198) foci colocalized with SAS-4 and not microtubules (Fig. 6, J and K). This is consistent with previous studies showing that the N-terminus of SPD-5 binds γ-tubulin complexes (Ohta et al., 2021), while the C-terminus is required for centriole binding (Nakajo et al., 2022). These results suggest that N-terminal coiled-coils mediate both self-assembly and the microtubule-nucleation capacity of SPD-5. We conclude that the assembly of full-sized, functional PCM scaffold requires multiple domains in both the N- and C-termini of SPD-5.

### Multiple SPD-5 domains are sufficient to form micron-scale structures in vitro

Our data, as well as previous studies, indicate that SPD-5 contains several coiled-coil-rich modules that mediate self-assembly. This type of multivalent architecture is common among proteins that form supramolecular networks, gels, and liquid droplets (Banani et al., 2017; Li et al., 2012). Our data also revealed that the coiled-coil modules vary in degree of importance for PCM assembly, suggesting a hierarchy. To clarify the nature of these hierarchical, multivalent interactions, we purified nine SPD-5 fragments (F20-28) and compared their ability to multimerize in vitro (Fig. 7 A and Fig. S7 A). Full-length (FL) SPD-5 and fragments that were predicted to contain coiled-coil domains showed alpha-helical signatures using CD spectroscopy (Fig. 7 B). F24 (a.a. 566–1198) and F28 (CC-long; Δ734-918) proteins were sufficient to assemble strongly fluorescent, micron-scale condensates, but not to the same degree as FL (mass of SPD-5 assemblies = $14 \times 10^6$ A.U [FL], $9 \times 10^6$ A.U.[F24], and $9.5 \times 10^6$ A.U.[F28]; Fig. 7 C). Other fragments (F20, F21, F22, F25, F26, F27) assembled into weakly fluorescent, diffraction-limited spots. F23 signal was diffuse, indicating no assembly (Fig. 7 C and Fig. S7 B). At equimolar concentrations, SPD-5 assembly scaled non-linearly with the molecular mass of coiled-coil domains ($R^2$ = 0.78; Fig. 7 D). These data indicate that no fragment can fully recapitulate the assembly properties of full-length SPD-5. Thus, a high degree of valency is required to achieve proper scaffold mass and morphology. Our results also indicate that C-terminal fragments assemble better than N-terminal fragments; for example, compare F24 ($9 \times 10^6$ A.U.) and F21 ($0.6 \times 10^6$ A.U.), which mirrors our findings in one-cell embryos (Fig. 6). We saw a striking reduction in the assembly of F26 versus F24, which differ by the presence of the helical hairpin domain. This result is consistent with our previous results (Fig. 3) and further illustrates the importance of the hairpin domain for SPD-5 self-assembly. These results support our

conclusion that a hierarchy of interactions between coiled-coil domains drives SPD-5 multimerization.

Next, we investigated which domains are sufficient for recruitment to a preassembled SPD-5 scaffold. We assembled condensates containing 1 μM full-length SPD-5::RFP and then added 10 nM GFP-labeled SPD-5 variants (Fig. 7 E). After 10 min, we analyzed the partition coefficient ($P_c$) of the GFP signal within the RFP-labeled condensates. SPD-5(FL)::GFP partitioned strongly into existing condensates ($P_c$ = 67), as expected. All other fragments showed poor to intermediate partitioning ($P_c$ = 4–43). Like the assembly assay, we saw that C-terminal fragments were recruited better than N-terminal fragments; for example, compare F24 ($P_c$ = 43) and F21 ($P_c$ = 7.4). We saw a 2.6-fold reduction in the partitioning of F26 versus F24, again indicating the importance of the Helical Hairpin domain for recruitment. Overall, there was a strong, non-linear relationship between coiled-coil content and $P_c$ ($R^2$ = 0.82; Fig. 7 F). Our results demonstrate that SPD-5 assembly and recruitment scale with coiled-coil content, but the domains do not play equivalent roles.

To investigate whether the serial linkage of coiled-coil motifs confers an advantage, we tested if increasing the concentration of a smaller fragment would make it equivalent to full-length protein in terms of assembly capacity. We thus performed SPD-5 assembly and recruitment assays using equivalent molar concentrations of coiled-coil motifs. Our results show that SPD-5(FL) outperforms the fragments (Fig. S7, B and C). We conclude that serial linkage of coiled-coil domains synergistically improves self-association, which could be explained as an increase in functional affinity through avidity. These data support a model where the PCM scaffold achieves material strength and maximal size through multivalent coiled-coil interactions between SPD-5 proteins.

## Discussion

In this study, we identified interaction modules and material design principles that enable the PCM scaffold to assemble and resist microtubule-mediated tensile stresses without fracture. We comprehensively analyzed the interaction landscape of SPD-5, the sole PCM scaffold protein in *C. elegans* (domain map shown in Fig. 8 A). Our data support a model where specific, multivalent coiled-coil interactions drive the multimerization of SPD-5 into micron-scale scaffolds that resist microtubule-mediated pulling forces (Fig. 8 B). Prior to assembly, unphosphorylated SPD-5 transitions between autoinhibited states, where the N- and C-termini fold back on themselves and across the whole protein. PLK-1 phosphorylation of SPD-5 eliminates most of these intramolecular interactions, thus opening SPD-5 so that its coiled-coil domains can engage other SPD-5 molecules. Phosphorylation also limits promiscuous intermolecular interactions. High-affinity dimerization of helical hairpins from separate SPD-5 molecules forms a "molecular linchpin" that

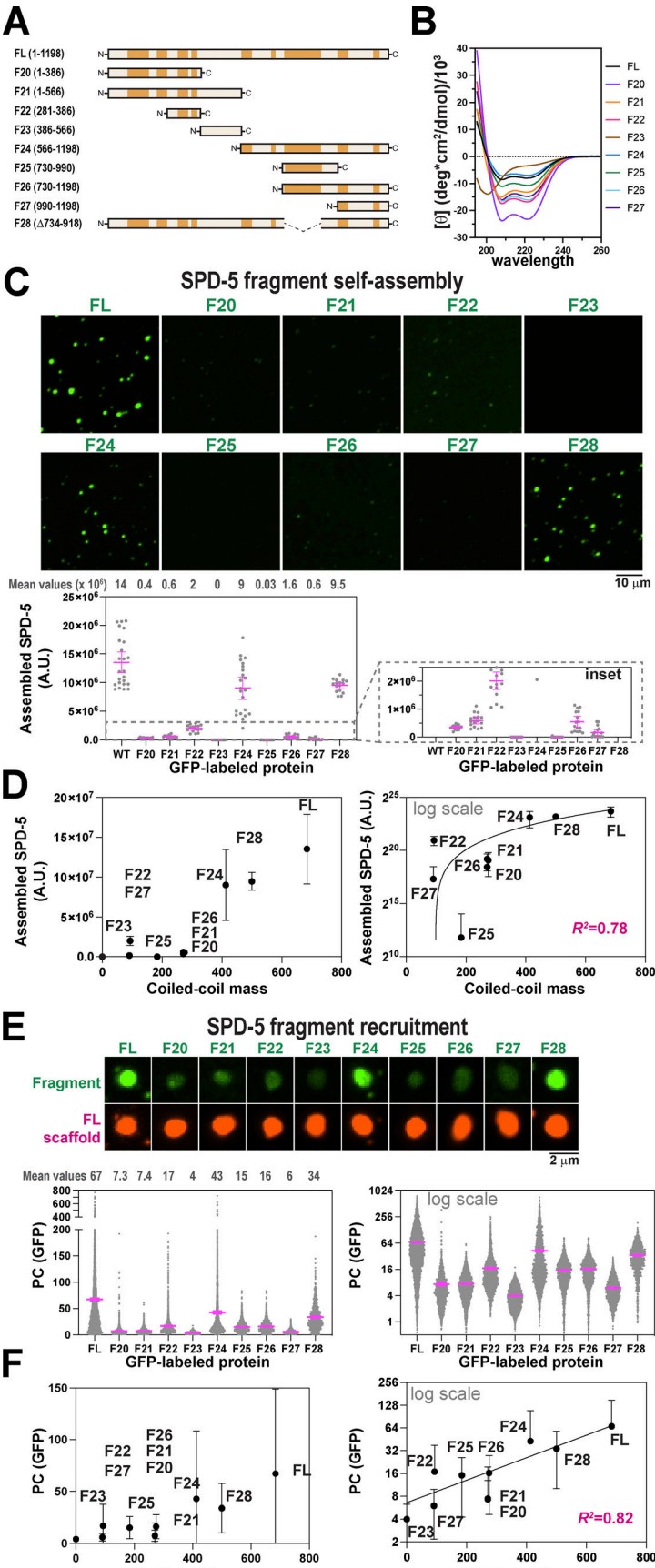

Figure 7. **Multiple SPD-5 regions are sufficient to form micron-scale assemblies in vitro. (A)** Domain maps of nine different SPD-5 variants. **(B)** Circular dichroism spectroscopy of purified SPD-5 fragments. **(C)** SPD-5 self-assembly assay. 500 nM of purified GFP-labeled SPD-5 proteins were incubated for 10 min in 7.5% (wt/vol) PEG-3350 then imaged. Mean ± 95% C.I.; $n$ = 11–22 images per condition. **(D)** Comparison of in vitro condensate mass with coiled-coil content. Mean ± 95% C.I.; $n$ = 11–22 images per condition. The data are fit with a quadratic model ($R^2$ = 0.78). **(E)** SPD-5 recruitment assay. 1 µM SPD-5::RFP was incubated in 7.5% PEG-3350 for 2 min to form condensates. 10 nM SPD-5::GFP proteins were added, incubated for 8 min, then imaged. Mean ± 95% C.I.; $n$ = 875–4,702 condensates per condition. **(F)** Comparison of in vitro recruitment with coiled-coil content. Mean ± 95% C.I.; $n$ = 875–4,702 condensates per condition. The data are fit with an exponential model ($R^2$ = 0.82).

Figure 8. **Model for SPD-5 scaffold assembly via multivalent coiled-coil interactions. (A)** Domain map of SPD-5. CM1, γ-tubulin complex binding domain; Phos. Hotspot, a region containing essential PLK-1 sites; PReM, phospho-regulated multimerization domain; CL, centriole-localization domain; CM2-like, essential region for SPD-5 assembly. The N-terminal assembly region, Helical Hairpin, and long coiled-coil domain (CC-Long) are new motifs described in this study. **(B)** SPD-5 is an elongated protein with coiled-coil domains separated by disordered linkers. Multiple coiled-coil domains interact to form intermolecular connections that drive multimerization of SPD-5 into micron-scale assemblies. The linkers allow chain flexibility so that SPD-5 can sample multiple configurations. Phosphorylation of SPD-5 prevents autoinhibitory intramolecular folding and promiscuous intermolecular interactions between linker regions. Strong interactions connect alpha-helical hairpins to form the structural core of SPD-5. The cumulative action of these domains enables full-scale PCM assembly and strength in *C. elegans* embryos.

drives the assembly of the PCM scaffold and holds it together under mechanical load. The serial linkage of coiled-coil domains in SPD-5 creates multivalency, which favors self-assembly through avidity. Disordered regions connecting the coiled-coils engage in limited interactions and likely create intraprotein flexibility, allowing the coiled-coils to sample different configurations. The C-terminus of SPD-5 anchors it at the centriole and makes connections with other SPD-5 molecules through a central phospho-regulated region (PReM) (Nakajo et al., 2022). Full-scale PCM size and strength thus emerge from the collective intermolecular interactions between coiled-coil domains of SPD-5.

Multivalency is a core principle underlying the supramolecular assembly of polymers. In typical polymer systems, interaction sites are equivalent and act as general stickers. By contrast, we demonstrate that a hierarchy of specific interactions between coiled-coil domains underlies SPD-5 multimerization. One major finding is that the C-terminal half of SPD-5, which contains CC-Long, the Helical Hairpin, and CM2-like domain, is more important than the N-terminal half for mediating SPD-5

multimerization, both in vitro and in vivo. While the N-terminal half of SPD-5 contributes weakly to SPD-5 multimerization in vitro, it contributes moderately to PCM assembly in vivo. What could explain this difference? It is possible that SPD-5 clients, which bind to the N-terminus (e.g., SPD-2, AIR-1, RSA-2) (Boxem et al., 2008), provide heterotypic interactions that also contribute to SPD-5 multimerization. Future work should characterize how these clients bind to SPD-5 and tune its multimerization.

At face value, our multivalent model seems to be at odds with previous observations of *spd-5(or213)* embryos, which express SPD-5 harboring a single mutation (R592K) that prevents PCM assembly and causes lethality at 25°C (Hamill et al., 2002). Why should mutation of any single domain dramatically affect the assembly of a multivalent protein? Here, we address this issue and show that R592 lies within an alpha-helical hairpin (a.a. 541–677) essential for PCM scaffold assembly and strength. PCM assembly in *spd-5(or213)* embryos is partly restored by the removal of microtubule-mediated forces. Thus, external forces

sensitize the system, such that small perturbations cause fracture and disassembly of the SPD-5 scaffold under stress. Similar cases are seen in engineering, where structural failure can be attributed to local discontinuities (Oliver et al., 2004). Our structural analyses suggest that SPD-5(541–677) dimerizes into a flared tetrameric coiled-coil and participates in long-range homotypic interactions. Thus, the Helical Hairpin represents a core structured module within the broader multivalent interaction network of SPD-5. The two other interaction modules we characterized (the N-terminus and CC-Long) also contribute to both PCM assembly and strength but to a lesser degree. We conclude that assembly and material properties are interrelated: PCM must be strong enough to resist microtubule-mediated pulling or else it will fracture and disassemble.

Could impaired interactions between SPD-5 molecules be the sole molecular cause of PCM weakness? Laser ablation of centrioles causes PCM fragmentation in a microtubule-dependent manner in *C. elegans* embryos (Cabral et al., 2019), suggesting that connection to centrioles is a major determinant of PCM force resistance. The SPD-5 mutants in this study localize to centrioles in the absence of forces, indicating that their major centriolar binding motifs are not disrupted; however, we cannot exclude the possibility that these mutations decrease the affinity of centriolar attachment. Our results nevertheless demonstrate that mechanical properties must be considered when assaying PCM assembly phenotypes.

PCM assembly is potentiated by PLK-1 phosphorylation of SPD-5, but the molecular mechanism is poorly understood. Previous studies proposed that PLK-1 helps SPD-5 transition to an active state that favors multimerization (Nakajo et al., 2022; Wueseke et al., 2016; Zwicker et al., 2014). Our XL-MS data provide structural evidence that unphosphorylated SPD-5 is folded up, such that coiled-coil domains engage in intramolecular interactions. PLK-1 phosphorylation of SPD-5 releases these interactions, freeing the coiled-coil domains so they can interact in trans. Given that PLK-1 phospho-sites occur primarily in non-coiled-coil regions, we speculate that phosphorylation increases linker stiffness, thereby limiting intramolecular folding. Thus, PLK-1 phosphorylation releases SPD-5 from an autoinhibited state. This effect might also explain how binding of the SPD-5 N-terminus to γ-tubulin complexes is regulated: PLK-1 phosphorylation at T170 and T198 releases intramolecular folding, revealing docking sites for the γ-tubulin complex, consistent with the model proposed by Ohta et al. (2021). Such a mechanism could also explain the regulated binding of γ-tubulin to Cnn and CDK5RAP2 (Tovey et al., 2021; Yang et al., 2023).

How does our work inform about the architecture of the PCM scaffold? Our XL-MS data revealed a complex and variable network of crosslinks between SPD-5 molecules, including those between the same coiled-coil domain (homotypic) or different domains (heterotypic). Both interaction types were found in the Helical Hairpin dimers. For some domains (e.g., coiled-coil 1 or CC-Long), the crosslinking patterns for the homotypic interactions are consistent with a parallel configuration. However, an in-depth structural assessment of isolated domains is required to fully understand how they interact with themselves and other domains.

Like SPD-5, PCM scaffold proteins in other species (e.g., Cnn, CDK5RAP2, PCNT) contain numerous dispersed coiled-coil domains and PLK-1 phosphorylation sites. Also, multiple fragments of CDK5RAP2 localize to PCM and make ectopic assemblies in mammalian cells (Kuriyama and Fisher, 2020). Thus, it is possible that PCM scaffold assembly through multivalent coiled-coils domains is a conserved mechanism. More broadly, our work highlights serial coiled-coil motifs as drivers of micron-scale, super-stoichiometric assemblies in cells. These principles could help understand the formation of other supramolecular assemblies rich in coiled-coil proteins, including microtubule-organizing centers, RNP granules, and endocytic initiation sites (Ford and Fioriti, 2020; Kozak and Kaksonen, 2022; Sallee and Feldman, 2021).

## Materials and methods
### Experimental model and subject details
For the expression of recombinant proteins (listed in Table S1), we used SF9-ESF *Spodoptera frugiperda* insect cells (Expression Systems) grown at 27°C in ESF 921 Insect Cell Culture Medium (Expression Systems) supplemented with Fetal Bovine Serum (2% final concentration). *C. elegans* worm strains were grown on nematode growth media (NGM) plates at 16–23°C following standard protocols (http://www.wormbook.org). Worm strains used in this study are listed in Table S2 and created using MosSCI (Frøkjaer-Jensen et al., 2008) or CRISPR (Paix et al., 2015).

### Prediction of coiled-coil motifs and secondary structure
Coiled-coil motifs in SPD-5 were predicted using MARCOIL at a 50% threshold, an MTK matrix, and a high transition probability (https://toolkit.tuebingen.mpg.de/tools/marcoil). Secondary structure was predicted using Alphafold (Jumper et al., 2021; Varadi et al., 2022).

### Protein purification
All expression plasmids are listed in Table S1. Full-length SPD-5 and PLK-1 constructs were expressed and purified as previously described (Woodruff and Hyman, 2015; Woodruff et al., 2015). For newly made proteins, genes encoding SPD-5 fragments were amplified from the full-length *spd-5* coding sequence using PCR and then inserted into a baculoviral expression plasmid (pOCC27 or pOCC28) using standard restriction cloning. Baculoviruses were generated using the FlexiBAC system (Lemaitre et al., 2019) in SF9 cells. Protein was harvested 72 hr after infection during the P3 production phase. Cells were collected, washed, and resuspended in harvest buffer (25 mM HEPES, pH 7.4, 150 mM NaCl). All subsequent steps were performed at 4°C. Cell pellets were resuspended in Buffer A (25 mM HEPES, pH 7.4, 30 mM imidazole, 500 mM KCl, 0.5 mM DTT, 1% glycerol, 0.1% CHAPS) + protease inhibitors and then lysed using a dounce homogenizer. Proteins were bound to Ni-NTA (Qiagen), washed with 10 column volumes of Buffer A, and eluted with 250 mM imidazole. The eluate was then bound to amylose resin (NEB) and washed with five column volumes of Buffer C (25 mM HEPES, pH 7.4, 500 mM NaCl, 0.5 mM DTT, 1% glycerol, 0.1%

CHAPS). Protein was eluted by adding PreScission protease, incubating overnight, and then passing over Glutathione Sepharose 4B (Sigma-Aldrich) to remove the Precission protease. Eluted protein was then concentrated using 3–50 K MWCO Amicon concentrators (Millipore). All proteins were aliquoted in PCR tubes, flash-frozen in liquid nitrogen, and stored at –80°C. Protein concentration was determined by measuring absorbance at 280 nm using a NanoDrop ND-1000 spectrophotometer (Thermo Fisher Scientific).

For SPD-5 (541–677), a codon-optimized gene block was cloned into the 6xHis_1B LIC plasmid (gift from Scott Gradia, UC Berkeley, Berkeley, CA, USA) and transformed into Arctic Express bacteria (Agilent). Expression was induced with 0.3 mM IPTG for 16 h at 20°C. Cell pellets were resuspended in Buffer A + protease inhibitors and then lysed using an Emulsiflex followed by sonication (40% amplitude, 5 s ON, 15 s OFF, for 90 s). The lysate was centrifuged at 24,000 rpm in a JA-25 rotor for 30 min. The supernatant was passed over Ni-NTA agarose, washed, and eluted with 250 mM imidazole. Eluate was concentrated with a 10 K MWCO Amicon concentrator (Millipore), filtered, and then passed over a Superdex 75 increase size exclusion column (Cytiva). Fractions containing SPD-5(541–677) were concentrated and aliquoted in PCR tubes, flash-frozen in liquid nitrogen, and stored at –80°C. Protein concentration was determined by measuring absorbance at 205 nm using a NanoDrop ND-1000 spectrophotometer (Thermo Fisher Scientific).

**Crosslinking mass spectrometry of SPD-5**
To identify PLK-1-driven changes in SPD-5 interactions, we found it was necessary to start with completely dephosphorylated SPD-5. Purification of dephosphorylated SPD-5 was achieved through modification of our standard protocol. In short, insect cell lysates were passed through 4 ml of Ni-NTA beads twice, washed five times with buffer 1 (25 mM HEPES, 500 mM NaCl, 30 mM imidazole, 1% glycerol, 0.1% CHAPS, pH 7.4), then twice with buffer 2 (150 mM KCl, 25 mM HEPES, pH 7.4) at 4°C. Ni-NTA-bound SPD-5 was incubated for 1 h at room temperature in dephosphorylation buffer (1× PMP buffer (NEB) + 1 mM MnCl$_2$), + 40,000 U lambda phosphatase (400,000 U/ml, NEB). Beads were then washed twice with buffer 1 at 4°C. Dephosphorylated SPD-5 was eluted from the Ni-NTA beads using 15 ml of Buffer 3 (25 mM HEPES, 500 mM NaCl, 250 mM imidazole, 1% glycerol, 0.1% CHAPS, pH 7.4). SPD-5 was then bound to 500 μl MBP-trap beads (Chromotek) and the column was washed 3× with buffer 4 (25 mM HEPES, 500 mM NaCl, 1% glycerol, 0.1% CHAPS, pH 7.4). Dephosphorylated SPD-5 was eluted from the MBP-trap beads by overnight incubation in buffer 4 + 100 μl of PreScission protease (1 mg/ml; Acro Biosystems) at 4°C. Eluted protein was further purified and stored as described above. Dephosphorylation of SPD-5 was confirmed by PTM identification using mass spectrometry showing effective removal of 99.8–100% of phosphates.

Crosslinking reactions were prepared at room temperature with 1 μM dephosphorylated SPD-5, 1 μM PLK-1 (KD/CA), 0.2 mM ATP, 10 mM MgCl$_2$, 150 mM KCl, 25 mM HEPES, pH 7.4 and 0.5 mM DTT. To analyze SPD-5 multimers, samples were incubated for 2 h at room temperature, followed by the addition

of 8 mM DMTMM for 45 min at room temperature (shaking at 300 rpm). To enrich for monomers, samples were incubated at room temperature for 2 h, then chilled on ice and gently pipetted for 10 min before adding DMTMM. To quench the reaction, we added 50 mM ammonium bicarbonate for 15 min at room temperature (shaking at 300 rpm). Samples without crosslinker were used for mass spectrometry PTM analysis to identify phosphorylated sites (Data S1).

Samples were run on an SDS-PAGE gel to separate crosslinked species. Bands corresponding to monomeric or multimeric protein were excised from the gel, then digested overnight with trypsin (Pierce), reduced with DTT, and alkylated with iodoacetamide (Sigma-Aldrich). Samples were cleaned using solid-phase extraction with an Oasis HLB plate (Waters), then injected into an Orbitrap Fusion Lumos mass spectrometer coupled to an Ultimate 3000 RSLC-Nano liquid chromatography system. Peptides were separated using a 75-μm i.d., 75-cm long EasySpray column (Thermo Fisher Scientific) and eluted with a gradient at a flow rate of 250 nl/min from 0 to 5% buffer B over 1 min, 5%–40% B over 60 min, 40%–99% over 25 min, and held at 99% B for 5 min before returning to 0% B for column equilibration. Buffer A contained 2% (vol/vol) ACN and 0.1% formic acid in water, and buffer B contained 80% (vol/vol) ACN, 10% (vol/vol) trifluoroethanol, and 0.1% formic acid in water. The mass spectrometer operated in positive ion mode with a source voltage of 1.5–2.4 kV and an ion transfer tube temperature of 275°C. MS scans were acquired at 120,000 resolution in the Orbitrap and up to 10 MS/MS spectra were obtained in the ion trap for each full spectrum acquired using collision-induced dissociation (CID) for ions with charges 3–7. Dynamic exclusion was set for 25 s after an ion was selected for fragmentation.

For data analysis, each Thermo.raw file was converted to.mzXML format for analysis using an in-house installation of xQuest (Leitner et al., 2014). Score thresholds were set through xProphet (Leitner et al., 2014), which uses a target/decoy model. The search parameters were set as follows. For zero-length crosslink search with DMTMM: maximum number of missed cleavages = 2, peptide length = 5–50 residues, fixed modifications carbamidomethyl-Cys (mass shift = 57.02146 Da), mass shift of crosslinker = –18.010595 Da, no monolink mass specified, MS$^1$ tolerance = 15 ppm, and MS$^2$ tolerance = 0.2 Da for common ions and 0.3 Da for crosslink ions; search in enumeration mode. The false discovery rates (FDR) were <20% (SPD-5(FL) or <5% (SPD-5[541–677]) at the link level (see Datas S3, S4, and S5 for further information about crosslinked peptide pairs). Samples were also reanalyzed accounting for mass shifts due to the presence of phospho-serines and phospho-threonines. A cumulative list of crosslinked pairs can be found in Data S2.

**Mass photometry**
500 nM SPD-5(541–677) was centrifuged to remove potential aggregates, then diluted into PBS to a final concentration of 50 nM on a clean glass coverslip. Protein was analyzed using a TwoMP mass photometer (Refeyn) using four species of BSA (monomer, dimer, trimer, and tetramer) to create a molecular weight standard curve.

## Ab initio modeling

The initial conformation of the SPD-5 541–677 fragment was generated using AlphaFold from full-length SPD-5 (Jumper et al., 2021; Varadi et al., 2022). The dimerization of this fragment was evaluated using two parallel strategies: AF multimer module (Evans et al., 2022, *Preprint*) and Rosetta docking (Gray et al., 2003; Chaudhury et al., 2011). For Rosetta modeling of the dimer, 10,000 models were generated in the absence of restraints using the low-resolution centroid mode followed by high-resolution refinement (Chaudhury et al., 2011). Additionally, we compared the energetics of the WT and R592K SPD-5 fragments (541–677) to evaluate the effect of the mutation on the stability of the dimer ensemble. The ensemble of dimeric models was evaluated against crosslinks obtained from the full-length and fragment SPD-5 XL-MS datasets by computing the sum of the distances for all experimentally observed cα-cα distance XL-MS pairs. Because no symmetry was imposed on the dimer generation, the sum of distances was computed for both geometries to test the symmetry of the interfaces. Final models were compared that exhibited low computed assembly energies and low cumulative distance between crosslink pairs.

## RNAi treatment

RNAi was done by feeding. The *spd-5* feeding clone targets a region that is reencoded in our MosSCI transgenes (Woodruff et al., 2015). Bacteria were seeded onto nematode growth media (NGM) supplemented with 1 mM isopropyl β-D-1-thiogalactopyranoside (IPTG) and 100 μg mL⁻¹ ampicillin. L4 hermaphrodites were grown on feeding plates at 23°C for 24 h. For *perm-1(RNAi)*, worms were grown on NGM + 0.1 mM IPTG at 25°C for 16 h (Carvalho et al., 2011).

## Target sequence for spd-5(RNAi)

5′-TGGAATTGTCCGCTACTGATGCAAACAACACAACTGTCG
GATCTTTTCGTGGAACTCTTGATGACATTCTGAAGAAAAACG
ATCCAGATTTCACATTAACCTCTGGTTATGAAGAAAGAAAGA
TCAACGACCTGGAGGCAAAGCTCCTCTCTGAGATCGACAAGGT
AGCTGAGCTGGAAGATCACATTCAGCAGCTCCGTCAAGAAC
TTGACGACCAATCTGCAAGGCTTGCCGATTCAGAAAATGTTC
GCGCTCAGCTTGAAGCGGCCACTGGACAAGGAATCCTCGGAG
CTGCTGGAAACGCTATGGTTCCAAATTCAACGTTCATGATCG
GGAACGGTCGTGAATCACAGACGCGAGACCAGCTCAATTACA
TTGATGATCTTGAAACGAAGTTAGCTGATGCGAAGAAGGAAA
ATGATAAGGCTCGTCAGGCACTCGTTGAATACA-3′.

## Target sequence for perm-1(RNAi):

5′-GCCACCAATCTGTCCAACTTTCTCTTTGGACATTTGACGATT
CAGAATATCCTGGTACGAATTTTGCTCGCTTATTCGCTTAAT
ATATTGATCGAAATTCAGAATTTGAAAATGTCTCCACTTGCC
TTCGTTACCACGGGATGAAGAACCTTCCTAATGCTCTGCTAG
CAGCTGATGTGGTAAACATACTAATATAATTATTTTTCTTCT
TAAAAAATAATTTTCTAGGTCGTCAATCTTCATGAATGCACC
GATTTCTCACTGCTTCCAGACGTTGCTAAACTTCAGATGCAC
AACGTGGAAGTTGTCAGATCAGTACTGTTCCATTGCAATTCG
CCATTGGTTCATCTTTCGACACCCTTCTTGCAATGTTCTCAT
CGCTGGCCAAATGTTTATGAACCCGAGCGTGATCCAGTCGTC
TTCAAACCTCAATGGCCATTCCCTGAGTATTGTGCGAGCAAG
TTTGAAGCCGAGAAACTCGTTAAATCCGCTCCGAACGACTGC

TACATTGTGTTTGTGCAATTACTTTCATTTTGTCTTTTTATT
TTGTATTAATTTCAGAAGATGCGTTCCAACTTATGGCGAAGG
CGACGATTGCTCTATTCTCACTGATTTGATCTATTTCTCCAA
CGACAAACCATTCTATTTCTCGGTGATGATGACGGTCATAT
GCAAATGGCGTACGCTGGAAATGCAGCTGTGGCAATATGGTC
AGCCGTCTGCAGATTACTTTCACAATCCACTAGTTTGAATCT
GAACGAGTCATTTGAAGAGGAATTGGGCGATTTGCTGACTTC
AGCCGAAAGCTCTTTCCGCTTCCATCAGGAGAGTGAGAGAGT
TTTTGAAAAGAGCAAATTGGAATTGTATGCCATTAAAGAAGA
GGACGAAAATCTTGAAGGGTATCGTGCTCGTCATAATACAGT
TCGCACCAGCATCGATGTCGACTCCGAGTCCAAAACAGATAT
CGACGAGAACTTGACCGAAGAAGGCGAAGTGTTCGAGCAAGG
TGATTGCACAAAACAAGGATTCAATTTCGAAATTGAGGACGA
ACCGCAGTCTCAAAATTTGGACTTCTCGATAATCAACGACTC
GAAATTTTCTAAAAATGATCGCGTTTTCGAGGTTATTT-3′.

## Western blotting

60 adult worms were picked and transferred to blank plates for 20 min to remove bacteria from their bodies. Worms were then moved to PCR tubes containing 10 μl of mili-Q water to which 10 μl of SDS loading buffer was added. Samples were separated by SDS-PAGE. Protein from each gel was transferred to a nitrocellulose membrane using a Trans-blot turbo transfer for high molecular weight proteins (10 min). Membranes were incubated in a blocking buffer consisting of 1× TBS-T + 3% Blotting-Grade Blocker (BioRad) shaking at room temperature for 1 h. Membranes were then washed three times with fresh 1× TBS-T and incubated shaking with primary antibodies overnight at 4°C. The primary antibody was washed three times with fresh 1× TBS-T and incubated by shaking with secondary antibodies at room temperature for 1 h. Each membrane was then incubated in ECL reagent (Thermo Fisher Scientific SuperSignal West Femto) for 5 min and imaged with a ChemiDoc Touch Imaging System. Primary antibodies: Mouse anti-alpha tubulin (Product # 3873S Lot #15; 1:1,000; Cell Signaling Technologies); Goat anti-enhanced GFP (1:5,000; Dresden PEP facility, 15 mg/ml; Poser et al., 2008), Rabbit anti-SPD-5 C-terminus (1:1,000, clone 758, Dresden Antibody Facility; Pelletier et al., 2004). Secondary Antibodies (1:50,000 for all): HRP conjugated Goat anti-Rabbit IgG (1 mg/ml) (# 65-6120, Lot # J276300; Invitrogen); HRP conjugated Goat anti-Mouse (1.5 mg/ml) (# 62-6520, Lot # WA312227; Invitrogen ); and HRP conjugated Donkey anti-Goat (1 mg/ml) (# A15999, Lot # 58-155-072318; Invitrogen).

## Embryo viability assay

10 L4 worms were picked and transferred to OP50 plates or RNAi feeding plates. 24 h later, individual adult worms were transferred to individual plates (10 worms per strain) and allowed to lay eggs for 6 h. Adult worms were then removed from the mating plates and eggs were manually counted. Hatched worms were then counted 24 h later. Plate viability is reported as the number of worms hatched divided by the number of eggs laid. Strain viability is reported as the average viability of 10 plates.

## Generation of *C. elegans* embryos expressing *spd-5* transgenes

*C. elegans* worm strains used in this study were created with MosSCI (Frøkjaer-Jensen et al., 2008) and based on constructs

made previously (Woodruff et al., 2015). Briefly, genomic sequences representing *spd-5* fragments were amplified from pOD1021(MosII)*Pspd2::spd-5(1-1198)::3'UTR* and inserted into the pCFJ151-based parent plasmid (pOD1021[MosII]) lacking the full-length insert. GFP::SPD-5(R592K) mutant was created by replacing a section of the wild-type sequence with a mutated gene block. All plasmids were purified using a NucleoBond Xtra Midi Prep Kit (Macherey Nagel), combined with coinjection plasmids, and injected into strain EG6699 (tt5605, Chr II). After 1 wk, worms were heat-shocked for 3 h at 35°C to kill worms maintaining extrachromosomal arrays. Moving worms without fluorescent co-injection markers were selected as candidates. Sequencing was used to confirm transgene integration.

### Generation of CRISPR-modified *C. elegans* mutants
*C. elegans* worms expressing tagRFP::SPD-5 at the endogenous *spd-5* locus (Magescas et al., 2019) were modified by CRISPR-Cas9 to delete the 93 bp sequence encoding a.a. 610–640 or the 555 bp sequence encoding a.a. 734–918 (PHX5737 spd-5[syb5737]). Modified worms were generated by SunyBiotech using gRNAs:

For PHX5737 (tagRFP::spd-5(delta734-918)):
Sg1: 5′-CCATTGAACGGTTCGTCTGGAAG-3′ and Sg2: 5′-CCGACTATTGTTCTACGAAGATG-3′.
For PHX5763 (tagRFP::spd-5(delta610-640)):
Sg1: 5′-CCAAATGCGGATAAGATCAAGAA-3′ and Sg2: 5′-CATCTTGGAAGCAGCGGAGAAGG-3′.

### Microscopy of *C. elegans* embryos
Embryos from adult worms were dissected on a 22 × 50 mm coverslip (Catalog # 2975-225; Coring) in 10-µl of egg salts buffer (ESB) containing 15-µm polystyrene beads (Sigma-Aldrich) using two 22-gauge needles. For microtubule depolymerization assays, ESB was mixed with nocodazole to make a 20 µM nocodazole solution or 1% DMSO. Samples were then mounted onto plain 25 × 75 × 1 mm microscope slides (# 12-544-4; Fisher Scientific). Time-lapse images were acquired with an inverted Nikon Eclipse Ti2-E microscope with a Yokogawa confocal scanner unit (CSU-W1), piezo Z stage, and an iXon Ultra 888 EMCCD camera (Andor), controlled by Nikon Elements software. We used a 60×1.2 NA Plan Apochromat water-immersion objective to acquire 41 × 0.5-µm Z-stacks with 100 ms exposures every min for 10 min starting slightly prior to or during pronuclear meeting. 488-nm excitation (15% laser power) was recorded using 2 × 2 binning followed by DIC imaging (92.3% iris intensity) using 1 × 1 binning.

### In vitro condensate assays
To assess SPD-5 self-assembly, GFP-labeled SPD-5 was incubated in condensate assembly buffer (25 mM HEPES, 150 mM KCl, 0.5 mM DTT, 3–9% [wt/vol] PEG-3350) at 23°C. To assess recruitment, 1,000 nM SPD-5::RFP was incubated in a condensate assembly buffer for 2 min and then 10 nM SPD-5::GFP (final concentration) was added. All condensates were transferred to a pre-cleaned glass-bottom imaging plate (ref#4580; 96 well; Corning) and settled for 8 min before imaging with a Nikon Eclipse Ti-2E spinning disk confocal microscope (described in the live-cell imaging section) and either a 40× 1.25 NA silicone or a 100×1.35 NA silicone immersion objective.

### Circular dichroism
0.3 mg/ml full-length SPD-5 or SPD-5 fragments were incubated in a buffer (20 mM K-Phosphate, pH 7.4, 0.01% CHAPS, 0.05 mM DTT, 100 mM NaCl) for 5 min at room temperature before being loaded into a 0.5 mm quartz cuvette (Hellma). Samples were analyzed using a Chirascan CD Spectrometer (Applied Photophysics). For the SPD-5 F1 fragments (a.a. 541–677), the protein was dialyzed against Working Buffer (5 mM sodium phosphate pH 7.4, 150 mM NaF). Protein was diluted to 0.21 mg/ml, placed in a quartz cuvette with 0.1 cm path length, and then analyzed using a Jasco J-815 CD spectrometer. Data were accumulated 10 times, averaged, and then analyzed with CONTIN, as implemented on the web server DichroWeb using reference set 4.

### Embryo immunofluorescence
*spd-5(or213 ts)* embryos were plated on 0.1 mM IPTG plates seeded with *perm-1(RNAi)* bacteria for 16 h at 25°C to permeabilize the eggshells and inactivate the *or213 ts* allele prior to immunostaining. Embryos were then mounted in ESB (118 mM NaCl, 48 mM KCl, 2 mM CaCl2, 2 mM MgCl2, 25 mM HEPES, balanced to 340 mOsm using sucrose) over Superfrost Plus slides (Fisher Scientific). For microtubule depolymerization, ESB was supplemented t with nocodazole (20 µM final) DMSO (1% final). The effectiveness of nocodazole was assessed using antibodies against alpha-tubulin. All other embryos expressing GFP::SPD-5 fragments were collected from adult *C. elegans* fed on *spd-5(RNAi)* plates for 24 h, mounted in M9 on Superfrost Plus slides. Embryos were then frozen in liquid nitrogen, fixed with methanol at –20°C for 10 min, and then washed twice with TBS-T for 5 min. Samples were blocked in TBST + 3% BSA for 30 min at room temperature and then incubated for 1 h with blocking buffer + primary antibodies (1:1,000 rabbit anti-SAS-4 (Kirkham et al., 2003), 1:1,000 rat anti-alpha tubulin-alexa647 (#ab195884; Abcam), and 1:5,000 rabbit anti-SPD-5 N-terminus (lot 785; Pelletier et al., 2004). Slides were washed three times for 5 min with TBST, incubated for 1 h in blocking buffer + secondary antibodies (1:1,000 donkey anti-rabbit alexa555 (#A31572; Invitrogen); goat anti-rabbit alexa488 (#A11008; Invitrogen)), and then washed three times for 5 min with TBS-T. Cells were mounted in a Vectashield mounting medium with DAPI (Vector laboratories) and imaged by fluorescence confocal microscopy (100× 1.35 NA silicone objective).

### Image quantification and statistical analyses
Images were analyzed using semiautomated, threshold-based particle analysis in FIJI. Data were plotted and statistical tests were performed using GraphPad prism. The sample size, measurement type, error type, and statistical test are described in the figure legends, where appropriate.

### Online supplemental material
Fig. S1 shows further analysis of SPD-5 crosslinks with and without phosphorylation, related to Fig. 1. Fig. S2 shows more structural analysis of the Helical Hairpin, related to Fig. 2. Fig. S3 shows extended analysis of the R592K mutation, related to Fig. 3. Fig. S4 shows extended analysis of SPD-5(Δa.a. 610–640), related

to Fig. 4. Fig. S5 shows extended analysis of CC-Long, related to Fig. 5. Fig. S6 shows extended analysis of the N-terminus of SPD-5, related to Fig. 6. Fig. S7 shows further analysis of SPD-5 assembly and recruitment in vitro, related to Fig. 7. Table S1 lists the plasmids used for protein production. Table S2 lists the *C. elegans* strains used in this study. Data S1 lists all phospho-sites detected in purified SPD-5 by MS. Data S2 lists all interacting pairs identified by XL-MS. Data S3 details XL-MS peptide analysis of SPD-5(a.a. 541–677). Data S4 details XL-MS peptide analysis of SPD-5(full-length) multimers. Data S5 details XL-MS peptide analysis of SPD-5(full-length) monomers.

### Data availability
Further requests and information for resources and reagents should be directed to and will be fulfilled by the corresponding author, Jeffrey Woodruff (Jeffrey.woodruff@utsouthwestern.edu). Raw mass spectrometry files can be found on the MassIVE database: MSV000094191, MSV000094193, MSV000094194, MSV000094195, MSV000094198, MSV000094199, MSV000093775.

## Acknowledgments
We thank Anthony Hyman, Karen Oegema, Bruce Bowerman, and Jessica Feldman for providing strains and antibodies; Andrea Zinke, Andrey Pozniakovsky, Susanne Ernst, and Craig Mello for their help with transgenic worm construction; the Macromolecular Biophysics Resource Facility at UT Southwestern for help with CD spectroscopy and mass photometry; the Protein Expression Facility at MPI-Dresden for help with expressing proteins; the Proteomics Core Facility at UT Southwestern for mass spectrometry; and Jesse Bucksot and Sofia Bali for their help with developing MATLAB analysis scripts.

J.B. Woodruff is supported by a Cancer Prevention Research Institute of Texas grant (RR170063), a Welch Foundation Grant (I-2052-20200401), an R35 grant from the National Institute of General Medical Sciences (1R35GM142522), and the Endowed Scholars program at UT Southwestern. L.A. Joachimiak is supported by a Welch Foundation Grant (I-1928-20200401), a Chan Zuckerberg Initiative Collaborative Grant (2018-191983), an MPI R01 from the National Institutes of Health (1RF1AG065407-01A1), and the Endowed Scholars Program at UT Southwestern. Some data presented in this article were acquired with a mass photometer that was supported by award S10OD030312-01 from the National Institutes of Health. B. Ferreira Gomes was supported by the Max Planck Society. M.U. Rios was supported by a National Research Service Award T32 (GM007062). M. Amato was supported by a National Research Service Award T32 (GM131963). K. Yaguchi was supported by a Human Frontier Fellowship (LT0064/2022-L). Open Access funding provided by The University of Texas Southwestern Medical Center.

Author contributions: M.U. Rios performed and analyzed in vivo experiments, purified dephosphorylated SPD-5, designed, performed, and analyzed XL-MS experiments. M.A. Baugnucka purified SPD-5 and performed SPD-5 XL-MS experiments. M.U. Rios, W. Stachera, and B. Ferreira-Gomes purified SPD-5 protein fragments. B. Ferreira-Gomes and M.U. Rios performed circular dichroism experiments. B.D. Ryder and L.A.

Joachimiak analyzed crosslinking data and performed Rosetta simulations. N. Familiari performed baculovirus-mediated protein expression and immunofluorescence with M.U. Rios. M.U. Rios and K. Yaguchi analyzed mutant embryos expressing RFP::SPD-5 and RFP::SPD-5(Δ734–918). J.B. Woodruff and M. Amato performed in vitro analysis of SPD-5 assembly and recruitment. M.U. Rios and J.B. Woodruff performed and analyzed mass photometry data and optimized XL-MS protocols with M.A. Bagnucka. J.B. Woodruff wrote the manuscript with input from L.A. Joachimiak and M.U. Rios.

Disclosures: The authors declare no competing interests exist.

Submitted: 29 June 2023

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

# Supplemental material

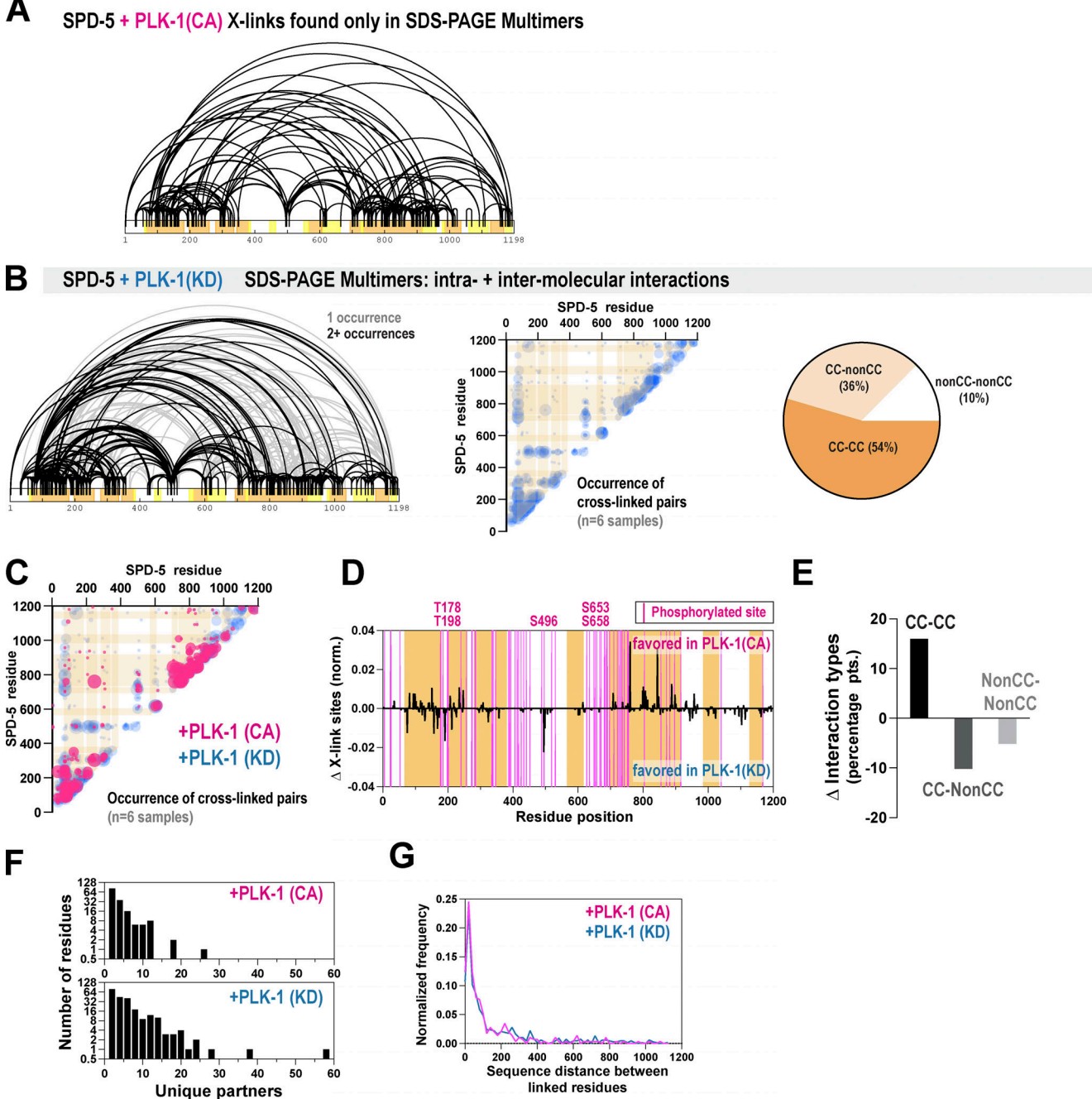

Figure S1. **Analysis of crosslinked residues in phosphorylated and unphosphorylated SPD-5. (A)** Crosslinks were found in SPD-5 (+PLK-1[CA]) SDS-PAGE multimer samples and not in SPD-5(+PLK-1[CA]) SDS-PAGE monomer samples. **(B)** SPD-5 was incubated with kinase-dead PLK-1, then crosslinked using DMTMM. SPD-5 multimers were analyzed by mass spectrometry ($n$ = 6 replicates). Left panel: Map of DMTMM-induced crosslinks. Center panel: Location and occurrence (indicated by bubble size) of crosslinked pairs across replicates. Orange bars indicate predicted coiled-coil domains. Right panel: Percentage of crosslinked pairs involving predicted coiled-coil domains (CC) or other regions (nonCC). **(C)** Quantification and location of crosslinked pairs in the pooled CA (red) or KD (blue) samples. Bubble size indicates number of replicates containing a given cross-linked pair. **(D)** The total number of times a residue was identified in a crosslinked pair was calculated for each condition. Data were normalized to account for overall differences in identified crosslinks. The difference between CA and KD samples is plotted. **(E)** The percentage of crosslinked pairs involving predicted coiled-coil domains (CC) or linker domains was determined for each condition. Shown is the difference in percentage points for each category between the CA and KD samples. **(F)** For each residue identified in a cross-linked pair, the number of unique partners was calculated. Data analyzed from multimer samples. **(G)** Histogram of sequence distances (in residues) between crosslinked pairs. Data were grouped into bins of 20 residues. Data analyzed from multimer samples.

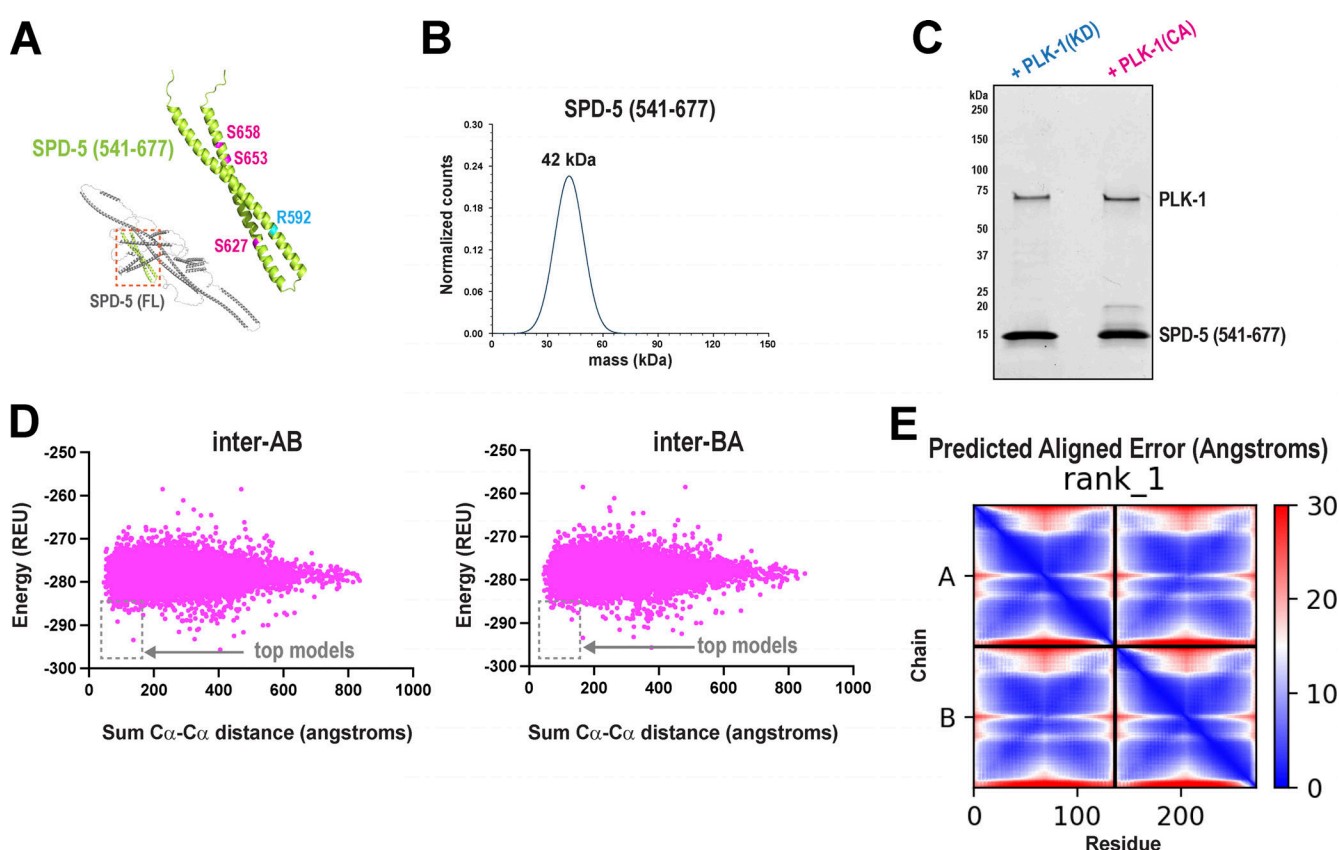

Figure S2.    **Biochemical and structural analysis of SPD-5(541–677). (A)** Alphafold predicts that the central region of SPD-5 forms an alpha-helical hairpin. This motif contains Arginine 592 (blue) and PLK-1 phosphorylation sites (Serines 627, 653, and 658; magenta) that are critical for PCM assembly in *C. elegans* embryos. **(B)** Mass photometry of 50 nM SPD-5(541–677). **(C)** SPD-5(541–677) was incubated with PLK-1(KD) or PLK-1(CA) and analyzed by SDS-PAGE. The slower migrating bands (~20 kDa) represent phosphorylated species of SPD-5. **(D)** Evaluation of 10,000 models built in Rosetta for SPD-5(541–677). **(E)** Predicted aligned error for the AlphaFold Multimer model of SPD-5(541–677). Source data are available for this figure: SourceData FS2.

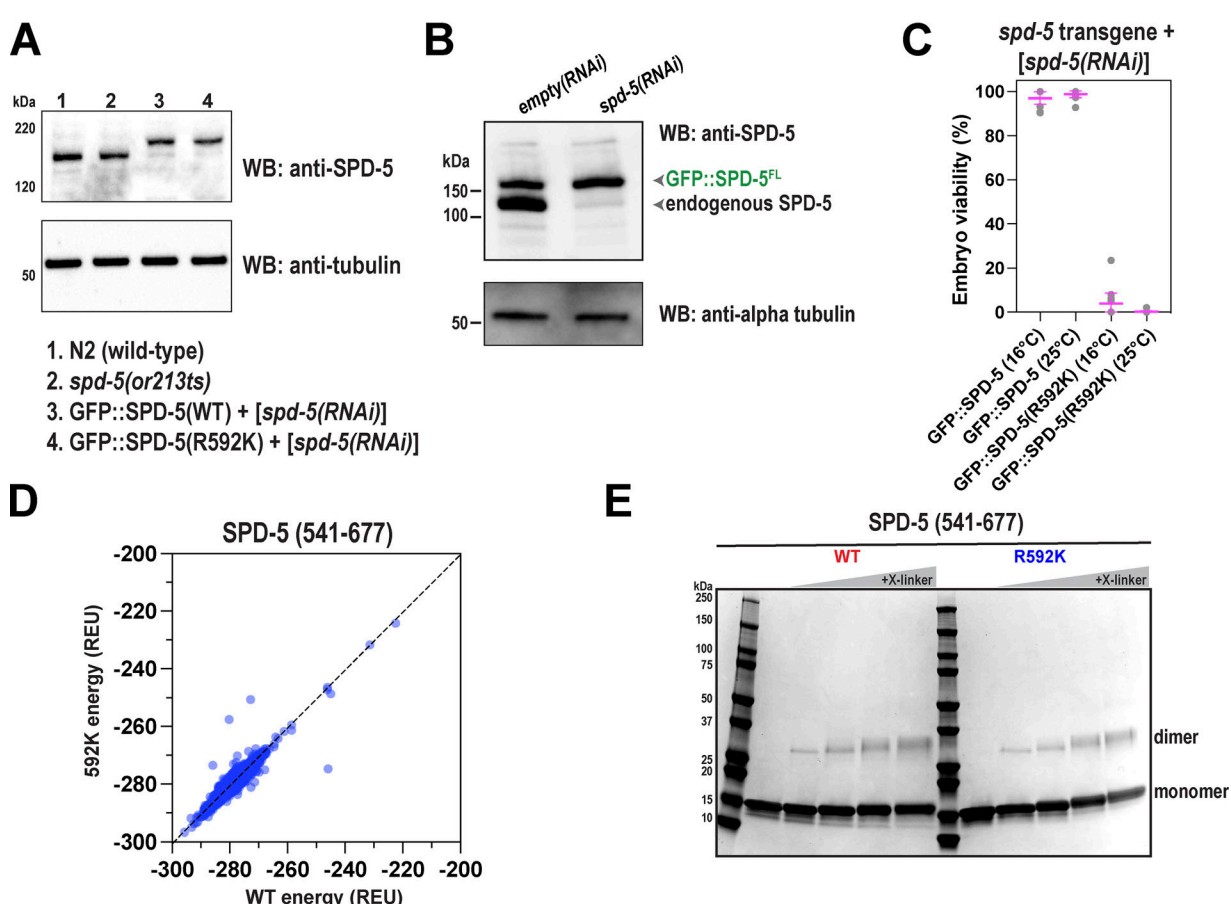

Figure S3. **Extended analysis of SPD-5(R592K). (A)** Western blots depicting expression levels of indicated proteins. Alpha tubulin was detected as a loading control. Worms were grown at 23°C. **(B)** Western blots showing depletion of endogenous SPD-5, but not transgenic GFP::SPD-5, following RNAi. Alpha tubulin was used as a loading control. **(C)** Viability of offspring in worms expressing RNA-resistant GFP::SPD-5 transgenes after depletion of endogenous SPD-5. Mean ± 95% C.I. ($n$ = 11 worms). **(D)** The energetics of 10,000 ab initio models were calculated and then compared with the same model containing the R592K substitution. Each dot represents one model. Data on the diagonal indicate that the mutation does not change the energetics of folding in a particular model. **(E)** SPD-5(541–677) (WT or R592K) was incubated with different amounts of DMTMM crosslinker (0–10 mM) for 45 min, then analyzed by SDS-PAGE. Source data are available for this figure: SourceData FS3.

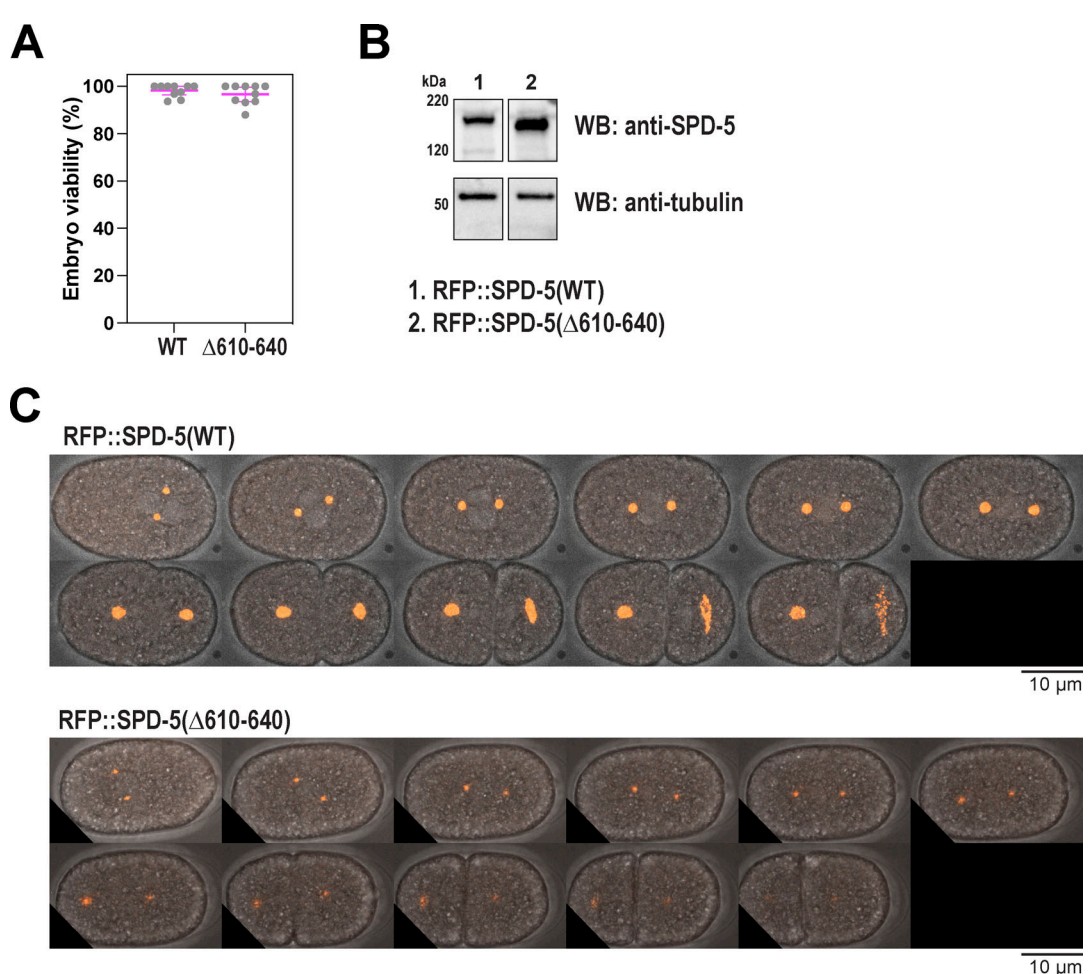

**Figure S4. Extended analysis of SPD-5(Δ610-640). (A)** Viability of offspring in worms expressing wild-type (WT) or mutant (Δ610–640) RFP::SPD-5. Mean ± 9 5% C.I. (*N* = 10 worms). **(B)** Western blots depicting expression levels of indicated proteins. Alpha tubulin was detected as a loading control. The blot for WT protein is the same used for Fig. S5 B. **(C)** Time-lapse confocal images of embryos during spindle assembly, spindle rocking, and PCM disassembly. Images were taken every 20 s. Source data are available for this figure: SourceData FS4.

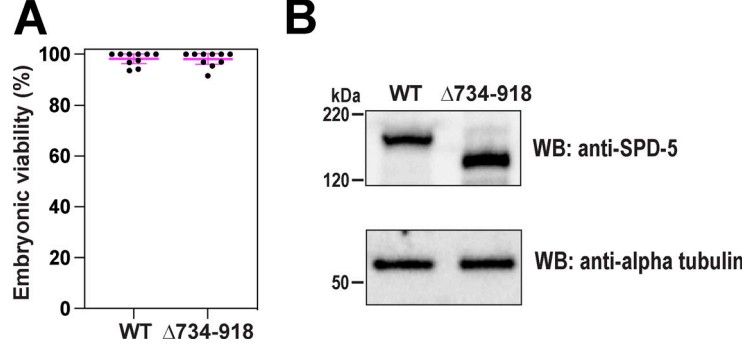

**Figure S5. Control experiments for SPD-5(Δ734–918) mutant analysis. (A)** Viability of embryos expressing either RFP::SPD-5(WT) or RFP::SPD-5(Δ734–918). Mean ± 96% C.I.; *n* = 10 worms per condition, 23–44 embryos each. **(B)** Western blot showing expression of RFP::SPD-5(WT) or RFP::SPD-5(Δ734–918). The blot for WT protein is the same as in Fig. S4 B. Alpha tubulin was used as a loading control. Source data are available for this figure: SourceData FS5.

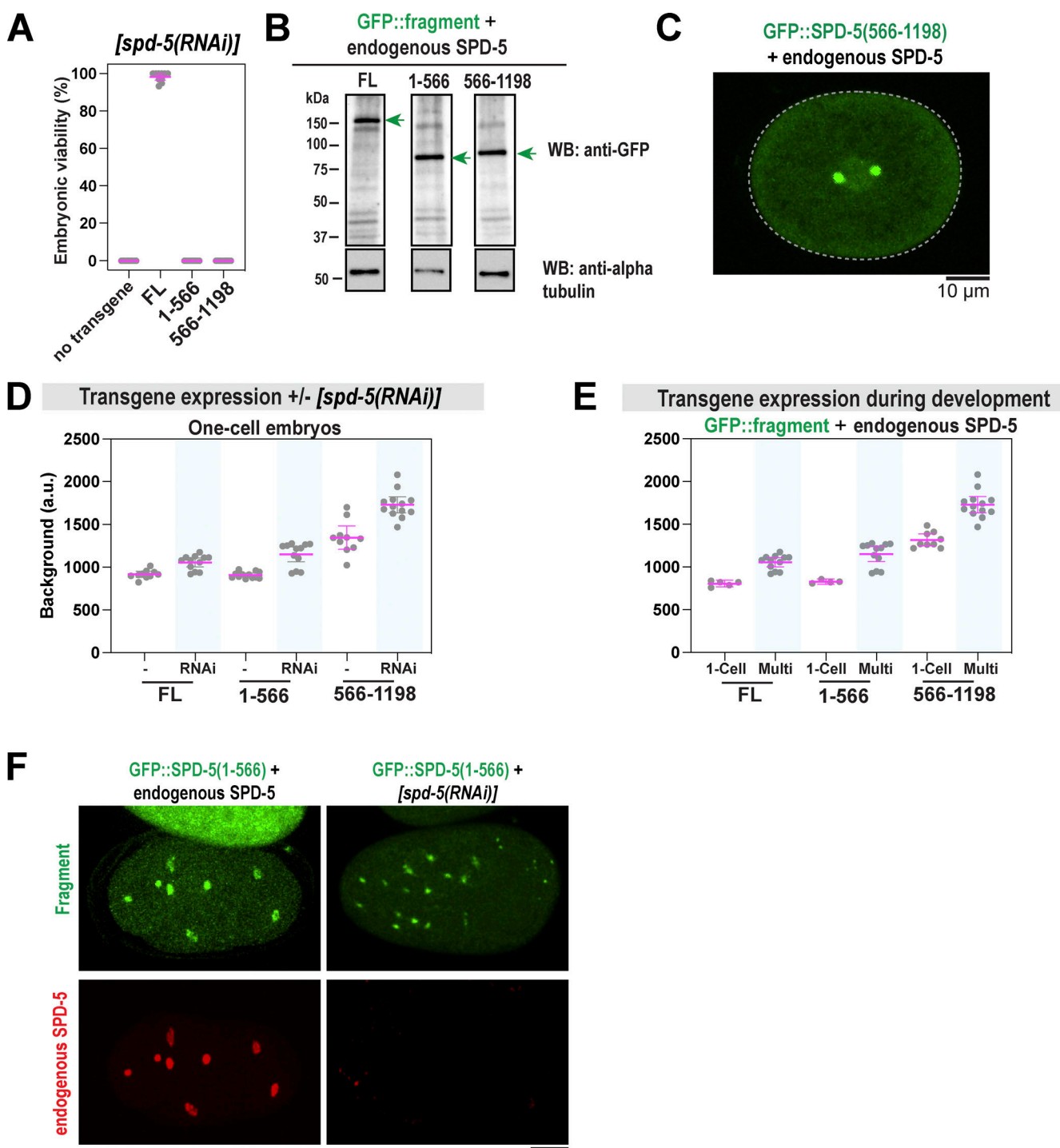

Figure S6. **Control experiments for SPD-5 truncation analysis. (A)** Viability of offspring after mothers were fed for 24 h on *spd-5(RNAi)* plates. No transgene (N2) compared with MosSCI worms expressing transgenic *gfp::spd-5*. Mean ± 95% C.I.; *n* = 10 worms per condition, 25–40 embryos each. **(B)** Western blot showing expression of *gfp::spd-5* transgenes. Alpha tubulin was used as a loading control. **(C)** Fluorescence confocal image of GFP::SPD-5(566–1198) in an embryo expressing endogenous SPD-5. **(D)** Quantification of cytoplasmic fluorescence of transgenic GFP::SPD-5 proteins in one-cell embryos with and without endogenous SPD-5. Mean ± 95% C.I.; *n* = 7–13 embryos. **(E)** Quantification of cytoplasmic fluorescence of transgenic GFP::SPD-5 proteins in one-cell stage versus multicell stage embryos (8-cell or greater). Mean ±95% C.I.; *n* = 5–12 embryos. **(F)** Immunofluorescence of transgenic embryos grown on control or *spd-5(RNAi)* feeding plates. GFP signal was preserved by light fixation. Endogenous SPD-5 was detected by an antibody that targets its C-terminus. Source data are available for this figure: SourceData FS6.

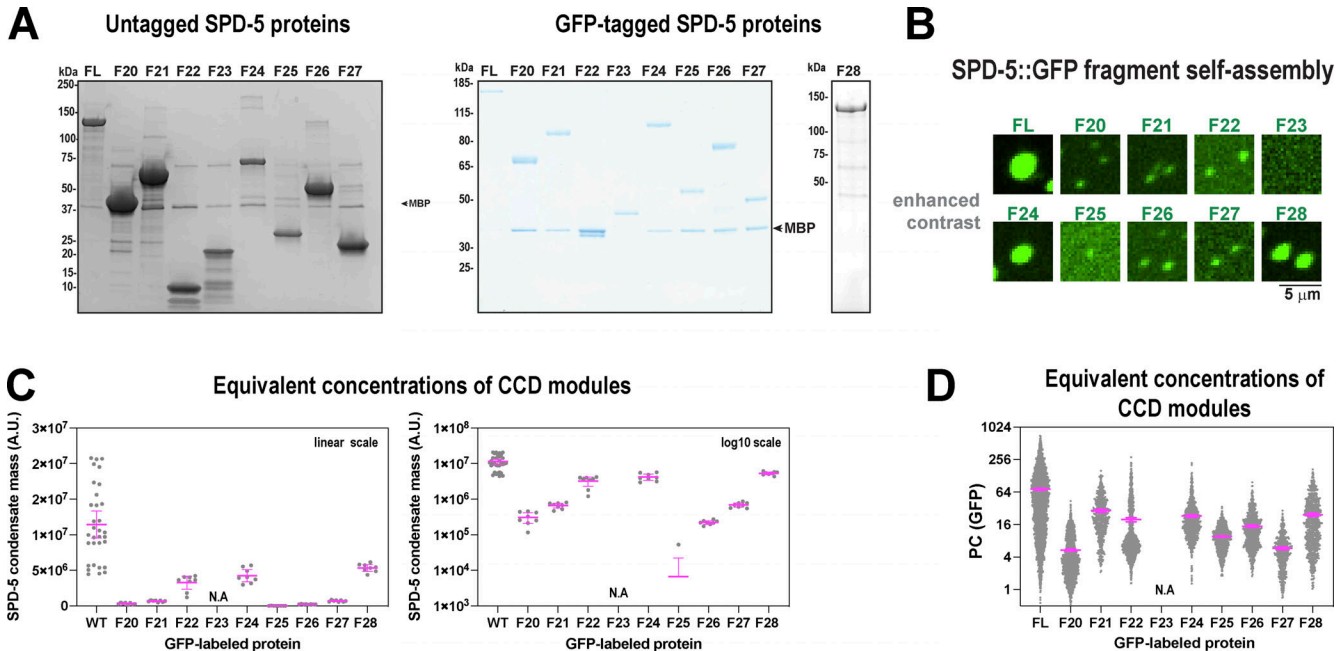

Figure S7.   **Analysis of SPD-5 assembly and recruitment in vitro. (A)** Coomassie-stained SDS-PAGE gels of SPD-5 proteins used in this study. **(B)** Contrast adjusted sections from images in Fig. 7 B highlighting detected assemblies. **(C)** In vitro SPD-5 assembly using equivalent molar concentrations of coiled-coil mass (500 nM (FL), 1,140 nM (F20), 1,140 nM (F21), 3,315 nM (F22), 890 nM (F24), 1,675 nM (F25), 1,120 nM (F26), 3,390 nM (F27), 710 nM (F28)). Mean ± 95% C.I.; n = 11–22 images. **(D)** In vitro SPD-5 recruitment using equivalent molar concentrations of coiled-coil mass (1,000 nM SPD-5::RFP; for GFP proteins: 10 nM (FL), 23 nM (F20), 23 nM (F21), 66 nM (F22), 18 nM (F24), 36 nM (F25), 22 nM (F26), 68 nM (F27), 14 nM (F28)). Mean ± 95% C.I. n = 532–4,702 condensates. Source data are available for this figure: SourceData FS7.

Provided online are Table S1, Table S2, Data S1, Data S2, Data S3, Data S4, and Data S5. Table S1 lists constructs for protein expression. Table S2 lists the *C. elegans* strains used in this study. Data S1 lists all phospho-sites detected in purified SPD-5 by MS. Data S2 lists all interacting pairs identified by XL-MS. Data S3 details XL-MS peptide analysis of SPD-5(a.a. 541–677). Data S4 details XL-MS peptide analysis of SPD-5(full-length) multimers. Data S5 details XL-MS peptide analysis of SPD-5(full-length) monomers.

