## [Peer Review File · The Journal of Cell Biology]

Multivalent coiled-coil interactions enable full-scale centrosome assembly and strength

Manolo Rios, Małgorzata Bagnucka, Bryan Ryder, Beatriz Ferriera Gomes, Nicole Familiari, Kan Yaguchi, Matthew Amato, Weronika Stachera, Łukasz Joachimiak, and Jeffrey Woodruff

Corresponding Author(s): Jeffrey Woodruff, The University of Texas Southwestern Medical Center

Review Timeline:

Submission Date:	2023-06-29
Editorial Decision:	2023-08-15
Revision Received:	2023-11-29
Editorial Decision:	2023-12-20
Revision Received:	2024-01-12

Monitoring Editor: Arshad Desai

Scientific Editor: Dan Simon

Transaction Report:

DOI: <https://doi.org/10.1083/jcb.202306142>

August 15, 2023

Re: JCB manuscript #202306142

Dr. Jeffrey B Woodruff
The University of Texas Southwestern Medical Center
Cell Biology
6000 Harry Hines Blvd
Dallas, TX 75235

Dear Dr. Woodruff,

Thank you for submitting your manuscript entitled "Multivalent coiled-coil interactions enable full-scale centrosome assembly and strength." We apologize for the delay in getting this decision to you and thank you for your patience. After receiving the original reviews, we decided to consult with an additional expert to help us with our decision. All 4 reviews are appended to this letter.

While the reviewers comment favorably on the technical aspects of the work, they all indicated problems with interpretation and limitations that led them to question the degree of advance presented. Reviewer 1 was the most critical, Reviewers 2 and 4 were more positive, and Reviewer 3 in the middle. Given their overall evaluations, we are unable to consider the submitted manuscript further for publication in the journal.

After considering the reviewer feedback and discussing with your colleagues you have the option to appeal this decision if you believe you can substantially revise the work and persuade the reviewers about its suitability for the journal. We believe any such appeal must include a plan to:

- 1) Improve the framing, analysis, and discussion of the cross-linking analysis, along with potential extension - see feedback from Rev 3 and Rev 4 on intramolecular versus intermolecular crosslinks, extension to hotspot fragments, and greater focus on the physiologically relevant phosphorylated state. It seems to us that specifically discriminating intramolecular versus intermolecular crosslinks, which would distinguish between different types of models, would be of great help here.
- 2) Significantly rework the manuscript to reduce emphasis on phase separation (see comments from Rev 1 & Rev 4 which question the relevance of this framework) and instead focus on concrete advances through identification of specific new elements.
- 3) More rigorously match the data presented to its discussion in the text and resolve/address potential inconsistencies (an issue raised by multiple reviewers - e.g. with respect to analysis of the hairpin helix, differentiating PCM assembly versus strength, matching embryo stages given changes in PCM, etc.).
- 4) Address concerns about confidence of structural models raised by Rev 1 and provide any relevant statistics.
- 5) Consider indirect effects of deletions (e.g. removal of kinase-binding sites in the absence of the N-terminus as pointed out by Rev 4) which complicate direct comparisons between in vivo and in vitro conditions.
- 6) Address differences between R592K and hairpin perturbations commented upon by reviewers (e.g. see feedback from Rev 3) and potentially feature the hairpin deletion analysis in the primary figure given that it represents a new element defined by the crosslinking approach.

The issues raised by the reviewers are more substantial than can be addressed in a typical revision period but given interest in the topic, we would be willing to consider a significantly revised and extended manuscript that fully addresses the reviewers' concerns and is subject to further peer-review. If you would like to resubmit this work to JCB, please contact the journal office to discuss an appeal of this decision or you may submit an appeal directly through our manuscript submission system. Please note that priority and novelty would be reassessed at resubmission.

Regardless of how you choose to proceed, we hope that the comments below will prove constructive as your work progresses. We would be happy to discuss the reviewer comments further once you've had a chance to consider the points raised in this letter. You can contact the journal office with any questions, cellbio@rockefeller.edu.

Thank you for thinking of JCB as an appropriate place to publish your work.

Sincerely,

Arshad Desai, PhD
Monitoring Editor
Journal of Cell Biology

Dan Simon, PhD

Reviewer #1 (Comments to the Authors (Required)):

The authors of the paper "Multivalent coiled-coil interactions enable full-scale centrosome assembly and strength" utilize a combination of cross-linking MS, structural modelling, biochemical assays and in vivo studies to characterise the assembly properties of the PCM protein Spd5 from *C.elegans*. Their main conclusions are that Spd5 assembly is driven by hierarchical, multivalent coiled-coil interactions that are (subtly) influenced by phosphorylation and are partly interdependent with microtubule-mediated pulling forces on the PCM.

Overall, I do not recommend publication in JCB, but consider this paper more suitable for a specialised journal.

The centrosome PCM is a dense meshwork that mainly consists of large coiled coil proteins, making the main conclusion of the paper (hierarchical multivalent coiled coil interactions) rather obvious to the point of being trivial. It is hard to see how PCM assembly could work differently given the known facts about the PCM, the proteins that constitute it and the behaviour of coiled coils. If a convincing high-resolution view of the underlying interactions were provided, this would be different, but the corresponding data are not particularly strong (see below). Furthermore, Spd5 is nematode-specific, lowering the appeal to a general readership.

Potentially interesting is the questions how the assembly is regulated and the authors' work on Spd5 phosphorylation could be important in this regard. However, this aspect of the paper is not very convincing. It is unclear what phosphorylation sites are functionally critical and the subtle changes in the observed cross-links are also not controlled for well. Phosphorylation introduces a bulky extra group and a charge, so phosphorylation can be expected to subtly influence the (loose) coiled coil packings in vitro. I suspect that many S/T kinases other than PLK1 added in the corresponding in vitro assay would show something similar (no such controls are provided).

The structural modelling also suffers from shortcomings. While AF2 or RosettaFold have been used to predict inter-domain packing, they mostly/often are not doing a great job on it, especially with coiled coil assemblies. The authors do not provide the relevant statistics to judge the prediction confidence (e.g. the PAE values in AF2). Given the high variability of the predictions that the authors report, I suspect that the corresponding values are in fact rather low. The structural models fail to explain the effect of the functionally well-characterised R592K mutation which is a further worry. The authors' models clearly need experimental high-resolution structural validation.

Overall, I also remain unconvinced by the functional testing of the assembly models. Larger coiled coil regions of Spd5 are removed for the in vivo tests instead of using point mutations based on their models. Even if the structural models provided by the authors were entirely wrong, removing such large regions can be easily imagined to change the fold or interactions with other proteins and is therefore hardly a test of the proposed structural models. This needs to be done better.

Finally, I am also overall sceptical of the authors' in vitro assay. The assemblies are driven by the addition of PEG-3000 as a crowding agent, which results in spherical condensates. Such spherical condensates are typically observed in many, many crystallisation screenings of protein constructs / coiled coil constructs, where such liquid-liquid phase separation (driven by the precipitant / protein super-saturation) is a potential precursor of a crystalline state. Particularly worrying in this respect is that almost all Spd5 fragments (bar one without a coiled coil) appear to lead to these condensates that the authors consider to be an equivalent to the situation in the PCM. I find the authors' conclusion that this shows how redundant the coiled-coils act rather optimistic, as it could be equally concluded that the assay might just not measure what it is supposed to. To be convincing, this will require better mutants and a tightly observed correlation between the in vivo and the in vitro situation.

Reviewer #2 (Comments to the Authors (Required)):

In this paper, Rios et al. investigate how the key centrosomal scaffold protein in *C. elegans*, SPD-5, builds into a force-resisting assembly that ensures centrosome integrity during mitosis. SPD-5 is known to self-associate and form condensates in vitro and this is promoted by Plk1 phosphorylation. Using cross-linking mass spectrometry, the authors now identify several interactions between different regions of SPD-5 and reveal how this interaction landscape is altered by Plk-1 phosphorylation. In general, Plk-1 promotes interactions between coiled-coil (CC) regions and prevents interactions between disordered linker regions. The authors identify 3 interaction hotspots that they then analyse further: two helices in the central PReM domain, a long C-terminal CC domain (named long-CC), and 4 CC's in the N-term that include the γ -TuRC binding CM1 domain. The authors show that the two PReM helices are predicted to form a hairpin, and that this hairpin homo-dimerises in a phosphorylation-independent manner. Computational prediction analysis suggests that the dimer forms a flared structure and that phosphorylation modulates its compactness and promotes the interaction of this region with CM2, while at the same time reducing the interaction of these

helices with other regions of SPD-5. The PReM-CM2 interaction is known to be an important driver of Cnn scaffold assembly in flies, so the authors findings are consistent with these interactions being a conserved feature important for PCM assembly. The authors investigate an R592L temperature-sensitive mutation that had previously been shown to perturb centrosome assembly. R592 is predicted to be on the surface of the hairpin and the authors show it is dispensable for hairpin formation, but important for condensate assembly and interactions between full-length SPD-5 molecules, helping to explain the centrosome assembly phenotypes. In addition, the authors delete a large part of the hairpin to further show its importance for PCM assembly. The authors note that these mutations affect both assembly of SPD-5 into a scaffold and the scaffold's ability to resist forces. The authors then test the importance of the long-CC and the N-terminal CC's. Deletion experiments show their requirement for PCM assembly and strength too, with the long-CC being most important. Finally, the authors purify multiple regions of SPD-5 and test their ability to form condensates - no one fragment can form condensates as well as full-length SPD-5, indicating that multivalent interactions are necessary. Moreover, these final experiments further highlight the importance of the hairpin and long-CC regions.

Overall, the paper is a tour-de-force in characterising the intra and inter-molecular interactions of a key centrosomal component, but I do find the conceptual advance limited considering what is known about SPD-5 and Plk-1's promotion of its assembly. For me the paper is very interesting and I would not be surprised to see it published in JCB, but for a wider readership the JCB editorial team should consider whether elucidating details on an already known, albeit important, process makes a sufficient advance. While the paper focusses on *C. elegans* SPD-5, it is generally agreed that SPD-5 is the functional homologue of fly Cnn and human CDK5RAP2, despite a limited sequence homology (which is common for *C. elegans* proteins). I do therefore feel that information of SPD-5 will be of interest to the wider centrosome and microtubule communities.

Major comments

- 1) The mutants analysed clearly affect both PCM assembly and PCM strength (as there is never a full rescue after nocodazole treatment). The authors do tend mention this but they do tend to overstate the effect on - for example, Figure 4C there is barely any rescue in PCM mass (probably the most important measure) for the C-term deletion mutant when adding nocodazole. The authors say the CC-long is "also required for full-scale assembly" but in reality, I think the CC-long is required mainly for PCM assembly, rather than force resistance.
- 2) Is it surprising that the data shows that interactions between CC regions and not disordered linker regions are important for scaffold assembly in vivo, when I thought the general consensus was that interactions between disordered regions promote condensate formation/liquid phase separation? The authors assume, and have assumed before, that condensate formation via liquid phase separation is the mechanism for PCM assembly in vivo, so this seems to be an important point. Do the authors think that the interactions between CC regions are transient in vivo, or are they maintained within the PCM to create a less dynamic and more scaffold-like structure?

Minor comments

- 1) When mentioning that fly Cnn also forms scaffold (lines 42-45), the authors should cite Conduit et al., 2014, Dev. Cell., as this is the first paper showing that Cnn forms a centrosomal scaffold. It could also be mentioned that this was the paper that first identified the PReM domain.
- 2) Line 498: when saying that the phenotype was rescued, I believe this should be changed to "partially rescued".
- 3) The paper is quite difficult to read, not because of bad English but just because several different techniques and so much data are presented. The authors would help the reader by explaining the techniques in more detail before presenting the data.

Reviewer #3 (Comments to the Authors (Required)):

Centrosomes function as the cell's primary microtubule-organizing center and consist of a pair of centrioles surrounded by pericentriolar material or PCM. The PCM is the site of microtubule nucleation and anchorage but the structure of the PCM is not well defined. PCM assembly likely involves a large number of molecular interactions among its constituent proteins. The identity of these molecular interactions and how they might be affected by polo like kinase 1 which promotes PCM assembly is not known. Likewise almost nothing is known about, how these molecular interactions might help the PCM resist microtubule-dependent forces.

Here the authors use the nematode *C. elegans* as a model system to study the assembly and material properties of the PCM. *C. elegans* PCM is thought to be composed almost entirely by the coiled-coil protein SPD-5. The authors use cross-linking mass spectrometry to map interactions sites between and within SPD-5 molecules in vitro and analyze how phosphorylation by PLK-1 leads to changes in the spectrum of interactions. They find that most interactions involve the coiled-coil regions of SPD-5 and that PLK-1 phosphorylation reduces the overall number of interactions, particularly those in linker regions while promoting interactions between coiled-coil domains. They also investigate the role of various SPD-5 motifs (helical hairpin, a long coiled-coil domain and the n-terminus) in PCM assembly and strength. Finally they analyze nine fragments of SPD-5 in vitro for the ability to form or be recruited into supramolecular structures. Their findings support the idea that multivalent interactions among the SPD-5 coiled-coil domains are a key feature of PCM assembly and strength.

From a technical standpoint, this study is truly impressive. The experiments, particularly the in vitro work, is well thought out and executed. My problem is that I don't think the results offer a significant advance in understanding PCM assembly or strength. The authors do describe the various intra and inter molecular interactions that take place before and after PLK-1 phosphorylation of SPD-5 in vitro, and that is important data, but the rest of the paper is concerned with a structure-function type approach to identify various SPD-5 motifs required for assembly or strength of the PCM. For instance, the helical hairpin and the long-cc are explored but disruption of either one of these structures doesn't even affect embryonic viability, meaning that they don't affect essential processes, such as spindle assembly or position. Moreover, all of the mutations they study strongly affect PCM assembly with only minimal effects on PCM strength (despite what the authors seem to argue in the text). This is not surprising as it seems obvious to me that the protein-protein interactions that drive assembly are also likely to impart material strength to the matrix. So what they really are studying here is mostly the interactions that drive PCM assembly but there just aren't any really exciting findings that make me feel the I can strongly endorse this work for publication in JCB. I'll keep an open mind though and if the other reviewers feel differently, I might reconsider. Below are my recommendations.

Major revisions:

Line 143: I think it would be helpful here to describe the capabilities and limitations of XL-MS. For instance I found myself immediately wondering if the approach was capable of distinguishing intramolecular vs intermolecular interactions but had to wait until later in the manuscript to find this out. Also I had questions about the quantitative nature of the approach. Are the authors able to identify the number of times a specific interaction occurs per unit mass of oligomer? Just a little bit of background here would go a long way in helping the reader appreciate the data.

Line 161. The authors state that interacting residues had overall fewer observed crosslinked partners in the phosphorylated state. How do the authors interpret this finding? Does this mean that phosphorylation results in a less dynamic PCM or does this mean that there are simply fewer static interactions allowed in the phosphorylated state? Or both?

Line 256: The authors wish to test the role of the helical hairpin in PCM strength and assembly. For this they use the *spd-5(or213)* allele that converts arginine 592 to a lysine (R592K). However it is clear that the mutation does not affect the dimerization, stability or the structure of the hairpin. Furthermore, as shown later in the paper, disrupting the hairpin by deleting a portion of the second helix does not have effects as severe as the R592K mutant. Thus the authors can't attribute the effects of the R592K mutation on a disruption of the hairpin. This means that the statement on line 315 that says the hairpin is required for full-scale assembly is incorrect.

Figure 3A and B. At what stage are the embryos being analyzed and how do the authors determine this stage? PCM levels change dramatically during embryogenesis and to properly compare embryos, they all need to be at the exact same stage. The wild-type embryo shown in panel A appears to be a mitotic two-cell embryo while the *spd-5(or213)* embryos appear to be one cell (based on centrosome number). The authors should stain for DNA so that the cell cycle stage can be determined properly. Also the authors state that failed PCM assembly in *spd-5(or213)* embryos is largely due to PCM weakness. I would say this is an overstatement as nocodazole-treatment only partially rescued PCM levels, yielding about half as much as controls. Finally, the authors need controls showing that nocodazole-treatment consistently disrupts the microtubule cytoskeleton. The P1 blastomere of the wild-type nocodazole-treated embryo shown in panel A looks like it has a normal sized and correctly positioned spindle, which you would not expect if microtubules were depolymerized.

Figure 3C and D. The same criticisms that apply to panels A and B apply here. However here the authors find that nocodazole only weakly rescues the PCM accumulation defect of the *spd-5* mutant (I estimate that nocodazole-treatment of the mutant results in only 20% as much PCM as controls based on panel D). Despite this, on lines 282-287, the authors seem to be emphasizing the effect of the R592K mutation on the strength of the SPD-5 scaffold. Looking at the data, it is clear that this mutation primarily affects assembly and not strength and the text should reflect that.

Line 328. The authors refer to a western blot to point out that wild-type SPD-5 and SPD-5(610-640) are expressed at similar levels. First the correct panel is S3F, and not S3E as stated. Second, it looks to me like the SPD-5(610-640) protein is expressed at twice the level of the wild-type. Again this is one of several places where the text doesn't accurately reflect the results.

Lines 319-229 and Figure S3G, H, and I. The authors disrupt the helical hairpin by deleting about 31 residues of the second alpha helix. Unlike the R592K mutant, his mutant is viable, but has very low levels of PCM that are only very weakly rescued by nocodazole treatment. Based on the author's own rational for using nocodazole to test the material strength of the SPD-5 network, I would say given the results shown in Figure S3H, that the helical hairpin plays only a modest role in PCM strength and that it's predominantly required for PCM assembly. Again, the conclusions on lines 335-339 don't line up accurately with the data as the authors stress a role for this motif in resisting microtubule-mediated pulling forces. Also in thinking about the effects of the SPD-5(610-640) mutant on PCM size, I began to wonder if a diminutive PCM might experience relatively higher pulling forces. That is, a wild-type PCM nucleates a certain percentage of microtubules that grow long enough to reach the cell cortex, where pulling forces are concentrated. However a small PCM might nucleate a higher percentage of long microtubules that

make it to the cortex. This might occur because the free tubulin pool is increased due to a reduction in overall microtubule nucleation. The result could be a higher pulling force per unit area (or mass, not sure which is most appropriate) of PCM for smaller centrosomes. This of course could affect interpretation of the nocodazole experiments. Do the authors have any way to measure the pulling forces on the variously sized centrosomes?

Line 394. The authors state that embryos expressing SPD-5(WT) and SPD-5(566-1198) do not exhibit PCM fragmentation or premature disassembly. Where is the data for this?

Line 402. The authors state that their finding that the N-terminus of SPD-5 is necessary for PCM assembly is inconsistent with a prior study showing that an N-terminal fragment of SPD-5 fails to assemble on its own. Why is this inconsistent? The experiments that the authors mention test two different things: necessity and sufficiency. Deletion of the N-terminus tests for necessity while expression of a n-terminal fragment tests for sufficiency. Therefore there is no inconsistency.

Figure 3I. What temperature is this assay done at? Can the authors recapitulate the ts properties of the R592K mutant?

Figure S3A. Were these animals grown at the nonpermissive temperature? Please specify.

Minor revisions:

Figure 1. Could the authors include the PReM domain in the SPD-5 schematics?

Line 359. The authors cite Figure 4A-C when they should have referenced 4A, B and D.

Line 363. The authors cite Figure 4D when they should have cited 4C.

Reviewer #4 (Comments to the Authors (Required)):

Centrosomes are composed of a centriole surrounded by a pericentriolar material matrix (PCM matrix). The PCM matrix is primarily composed of molecules of the CDK5RAP2 family (CDK5RAP2 in humans, Cnn in *Drosophila* and SPD-5 in *C. elegans*) which contain a series of alpha-helices that are predicted to form coiled-coils. During mitotic entry, the PCM matrix expands via Polo-like kinase 1 (PLK1) phosphorylation-dependent self-assembly of PCM matrix molecules. Major questions in the field are how the PCM matrix is organized, both how the PCM matrix molecules are structured and how they interact with each other to form a matrix, and how the assembly process is controlled by phosphorylation. Prior to this work, the authors developed an in vitro assembly system for the *C. elegans* matrix molecule SPD-5, whose assembly in vitro was augmented by PLK1 phosphorylation. In this manuscript, they combine this in vitro system with a crosslinking-based approach to try to understand how the PCM matrix is organized. They compare SPD-5 matrix assembled in the presence versus the absence of PLK1 phosphorylation and show that they get fewer crosslinks (and a higher percentage of crosslinks between predicted coiled-coil domains in the presence of phosphorylation, suggesting that phosphorylation enhances the specificity of SPD-5-SPD-5 interactions. The work goes on to characterize several regions of SPD-5 that are important for PCM assembly and/or strength; these regions include a nice characterization of SPD-5 residues 541-677, which they show forms a helical hairpin that dimerizes to form a tetrameric coiled-coil and a second a long coiled-coil region.

Overall, the authors employ a combination of interesting approaches to tackle the difficult question of how the PCM matrix assembles around centrioles. I would support publication in the JCB if the points below can be addressed.

Main points:

1. The approach of cross-linking PCM is interesting and potentially useful. However, as it is presented here, it is quite challenging to interpret. This is because it appears that individual SPD-5 molecules may be folding back onto themselves, sort of like intramolecular origami, with an example being the coiled-coil "hairpin" that the authors describe in Fig. 2. On top of this, there is a second layer of inter-molecular interactions between SPD-5 molecules that hold together the assembled PCM network. In the crosslink maps in Fig. 1D and E these layers are mixed together, making it impossible to know how much represents molecular folding and how much represents intermolecular proximity within the assembled networks. The authors mention in passing that they could not tease apart these layers by sub-saturation crosslinking for full-length SPD-5, as they did for the smaller SPD-5 (541-677) construct in Fig. 2E-G, to identify the intra-molecular versus inter-molecular cross-links. If this cannot be done in the context of full-length SPD-5, could it be done in the context of a few smaller regions of the molecule (for example the three crosslinking "hotspots" that the authors identify (aa 1-566, 541-677 and 734-918)? ---noting that the authors already do a nice job of this for the second hotspot aa 541-677 in Fig. 2. If not, the point that these two types of crosslinks cannot be distinguished should be more extensively discussed both in the section where the crosslinking is described and in the discussion.

2. In vivo, SPD-5 only assembles if it is phosphorylated by PLK1. Thus, the meaning of the comparison between the

physiological assemblies that form from phosphorylated SPD-5 and non-physiological assemblies that form in vitro from SPD-5 that has not been phosphorylated (which based on prior work from this group appears to require higher concentrations for assembly compared to phosphorylated SPD-5) is not very clear. It seems to reveal that there is a significant increase in promiscuous crosslinks between disordered linker regions if non-phosphorylated SPD-5, which would not normally assemble in vivo, is forced to do so. One might expect that this could be molecules in an autoinhibited configuration forced together by promiscuous linker interactions. Given that the significance of this material is not clear, it seems like the manuscript might be strengthened by moving some of the panels related to this in Fig. 1D-I to the supplement to better highlight the crosslinking map for the potentially physiologically relevant material. For example, in panel F it is difficult to see the map of the pink (phosphorylated SPD-5) crosslinks because of the blue (not phosphorylated SPD-5) crosslinks that is superimposed over it.

3. Lines 177-182: "In addition, we identified previously uncharacterized interaction sites, including three hotspots: 1) two helices within the PReM region (a.a. 541-677), 2) a C-terminal, long coiled-coil domain (a.a. 734-918), and 3) four coiled-coil domains in the N-terminus (a.a. 1-566)." Looking at the map in Fig. 1D, the region between aa 1 and 345 and the region between 694 and 984, do indeed appear to be cross-linking hot spots. However, the region between 541 and 677 does not appear to be a crosslinking hot spot, should this region be included in this list? As per point 1, it would be interesting to know how many of the cross-links in the two hotspot regions that are not characterized in further detail (1-566 and 734-918) are predicted to be intramolecular, perhaps the crosslinking data is revealing much more extensive intramolecular folding than expected.

4. Lines 198-201: "We identified short-range links within this region (e.g., E602-K616, E601-K618) and long-range links with other regions of SPD-5. Phosphorylation reduced the number of contacts overall originating from this region and created a new contact with the CM2-like domain (E665-K1160)(Figure S2A)." The crosslinks to this region, which are highlighted in Fig. S2 are not prominent in Fig. 1D. This seems to be because these crosslinks were only observed in one occurrence, as opposed to other crosslinks that were observed in 2+ occurrences. Are these crosslinks reliable-should any crosslinks that were only observed 1 time be featured in the manuscript.

5. In Fig. 2F,G, is it possible to color-code the crosslinks in the intra + inter-molecular maps on the right so that the intramolecular ones are one color and the ones not in the intra-molecular maps on the left are a second color? This would make it easier to spot the potential inter-molecular crosslinks.

6. In general, I like the analysis in Figures 2-4 testing the role of the various regions in PCM assembly/strength. However, it is not clear that deletion of the N-terminal region of SPD-5 (the 566-1198 construct) can be analyzed in the same way. This N-terminal region of SPD-5 has previously been shown to interact with SPD-2, AIR-1 and the RSA complex (PMID:18692475) and is thought to play a signaling role in PCM assembly by delivering active PLK1 kinase to the PCM (PMID: 21802300). Thus, removing the SPD-2 binding site might prevent SPD-5 assembly in vivo by preventing SPD-5 phosphorylation, rather than due to loss of structural elements that contribute to PCM assembly via interaction with other structural elements. Thus, it cannot really be concluded that the N-terminal coiled-coils structurally contribute to SPD-5 self-assembly. The idea that the N-terminal coiled-coils do not directly mediate SPD-5 self-assembly is supported by the in vitro data shown in Figure 6, which show that the N-terminal constructs F20-F23 do not assemble well on their own and do not associate with pre-assembled PCM, suggesting that the helices in this region do not make promiscuous contacts with helices in the remainder of SPD-5. In addition this data shows that the F24 construct which contains only the C-terminus-assembles in vitro (which PLK1 delivery is likely not limiting) in a fashion that is pretty comparable to full-length SPD-5. These points should be discussed in the text.

7. Although, the data in the paper is interesting and important, it does not provide support for the multivalent "stickers" and "linkers" model outlined in Figure 7, which seems to imply promiscuous association of different coiled-coil regions with multiple binding partners. There is no data in the paper that supports the promiscuous interaction of a coiled-coil region with multiple other coiled-coil regions. Instead, when the authors dig down into a structural element that they identify as important for PCM assembly they find evidence for very specific interactions-identifying a pair of adjacent helices that fold back on each other to form a helical hairpin that subsequently dimerizes to form a tetrameric coiled coil. Thus, one implication of the data is that there may be more intramolecular folding of the SPD-5 helices than previously anticipated-which could be involved in both autoinhibition of the non-phosphorylated protein and in key structural elements within the interacting phosphorylated protein. The authors further show that phosphorylation suppresses the promiscuous interaction of linkers and show that the helices in the N-terminal region of SPD-5 do not drive significant assembly or promiscuously interact with assembled PCM. The "stickers and linkers" model and accompanying discussion should be removed and replaced with a more accurate discussion of the data and significance of the findings in the paper.

Minor comments:

1. In the first paragraph of the introduction, it feels like different types of things are being mixed and matched (1) mitotic PCM in which PCM matrix molecules self-assemble in a PLK1 phosphorylation-dependent fashion, (2) interphase PCM which usually assembles in a single layer around centrioles (PMID: 36107993), but also seems to be present at the base of cilia when centrioles are absent (PMID: 33798428, PMID: 33798427) and (3) the microtubule-nucleating centers in meiotic mouse oocytes. Also, relevant here is recent work suggesting that there are at least two robust microtubule generation pathways anchored at centrioles, one of which is independent of the PCM matrix molecules (CDK5RAP2/Cnn/SPD-5; PMID: 33170211; <https://www.biorxiv.org/content/10.1101/2022.09.23.509043v4>). Overall, the intro would be much stronger if it stayed focused on the mitotic self-assembling CDK5RAP2/Cnn/SPD-5 type matrix that is the subject of the paper.

2. Lines 127-128 "AlphaFold predicts that SPD-5 contains 14 alpha helices connected by disordered linker regions (Jumper et al., 2021)". It may have to do with the overlapping orange and yellow colors, but I only see 12 alpha helices indicated on Fig. 1B.
3. Please label SPD-5 amino acid numbers on the SPD-5 diagram in Fig. 1E as well as in Fig. 1D.
4. Discussion of the PReM region in the intro should include a reference to the paper where it was originally defined in *Drosophila Cnn* (PMID:24656740).
5. The reference on line 81 to Feng et al., 2017, should also include a reference to PMID: 35362532.

Dear Arshad,

Thank you for the careful assessment of our manuscript. As you have seen, 3 out of 4 reviewers responded positively to the technical advance of this study:

Reviewer 2: “Overall, the paper is a **tour-de-force** in characterising the intra and inter-molecular interactions of a key centrosomal component . . .”

Reviewer 3: “From a technical standpoint, **this study is truly impressive**. The experiments, particularly the in vitro work, is well thought out and executed.”

Reviewer 4: “Overall, the authors employ **a combination of interesting approaches to tackle the difficult question** of how the PCM matrix assembles around centrioles”

However, the reviewers had several issues related to interpretation of the XL-MS data, the final model, and description of some of the data, as outlined in your 6 points below.

We have addressed all the reviewers' points. Most importantly, we provide additional experiments that distinguish intra- vs. inter-molecular interactions in the XL-MS data and reveal PLK-1-induced structural changes in the SPD-5 monomer. We feel that these revision experiments have significantly improved our interpretations, strengthened our conclusions, and provide new structural insight into how SPD-5 is regulated through auto-inhibition. We describe the big picture revisions here, then point-by-point in the following sections.

Sincerely,
Jeff Woodruff

Revision Overview:

1) Improve the framing, analysis, and discussion of the cross-linking analysis, along with potential extension - see feedback from Rev 3 and Rev 4 on intramolecular versus intermolecular crosslinks, extension to hotspot fragments, and greater focus on the physiologically relevant phosphorylated state. It seems to us that specifically discriminating intramolecular versus intermolecular crosslinks, which would distinguish between different types of models, would be of great help here.

In the first part of the results section, we explain better how the XL-MS is performed, quantified, and interpreted. We also followed the suggestion of Reviewer 4 and moved the data on unphosphorylated SPD-5 multimers to the supplement. The focus of Figure 1 is now on the more physiological mechanism of SPD-5 assembly that involves phosphorylation.

Regarding intra- vs. inter-molecular interactions, we went to great efforts to enrich for SPD-5 monomers and analyze them with XL-MS in the presence of kinase dead or active PLK-1 (2 reactions; 6 replicates each). To do this, we set up the reactions in the same way as before, but, before X-linking, we chill the reaction on ice for 45 min. We know from our previous study (Woodruff et al., 2015), that SPD-5 multimers will partly disassemble upon cold shock. Everything else is done the same way (same X-linking time, quench, prep for MS, etc).

Analysis of SPD-5 monomers gives us the true intra-molecular interactions (Figure 1D,E). To identify the inter-molecular interactions in the multimer samples, we identify all X-links (Figure

1F), then simply subtract the bona fide intra-molecular interactions (Figure S1B). The results were striking. Unphosphorylated SPD-5 has many intra-molecular interactions, suggesting that it is folded up, as predicted by Reviewer 4. Phosphorylation removes almost all long-range X-links, suggesting that PLK-1 opens SPD-5, which allows then coiled-coil domains to engage with other SPD-5 molecules (Figure 1G). These data provide structural data to support the idea that SPD-5 is auto-inhibited, and that PLK-1 promotes assembly by relieving this auto-inhibition. This idea had been proposed (by us, Karen Oegema, and Jordan Raff for Cnn) but lacked structural evidence to support it. We feel this adds to the significance of our study, as it illuminates the molecular mechanism by which PLK-1 potentiates PCM assembly.

2) Significantly rework the manuscript to reduce emphasis on phase separation (see comments from Rev 1 & Rev 4 which question the relevance of this framework) and instead focus on concrete advances through identification of specific new elements.

We have changed the model, indicating how specific, multivalent interactions between coiled-coil domains drive SPD-5 assembly (Figure 8). We no longer invoke the “stickers and spacers” model of typical polymers (as proposed by Rohit Pappu).

3) More rigorously match the data presented to its discussion in the text and resolve/address potential inconsistencies (an issue raised by multiple reviewers - e.g. with respect to analysis of the hairpin helix, differentiating PCM assembly versus strength, matching embryo stages given changes in PCM, etc.).

We have addressed each of these, point by point (see below).

4) Address concerns about confidence of structural models raised by Rev 1 and provide any relevant statistics.

We now provide the error statistics for the AF2 model (predicted aligned error) in Figure S2F. The model has low error (high confidence) in the center, where the tetra-helical bundle is predicted. Areas of low confidence correspond to the loop connecting the two helices in the hairpin and the flared ends that contain disordered linkers. We also evaluate the Rosetta and AF2 models using X-linking distance, which is a standard method to evaluate structural models. The models are compatible with the X-links, as no distance violations are reported (all links were <24 Å, acceptable for DMTMM; Figure 2J,K).

5) Consider indirect effects of deletions (e.g. removal of kinase-binding sites in the absence of the N-terminus as pointed out by Rev 4) which complicate direct comparisons between in vivo and in vitro conditions.

We now provide a section in the discussion on this point. This is described in detail in our point-by-point response. Our data show that the N-terminus makes homotypic, inter-molecular interactions (Figure 1) and that these contribute to scaffold assembly directly (Figure 7). Nevertheless, we cannot discount the contribution of SPD-5 binding partners, especially those that bind the N-terminus, to SPD-5 scaffold assembly. Such experiments to understand the impact of these partners are important but beyond the scope of this manuscript.

6) Address differences between R592K and hairpin perturbations commented upon by reviewers (e.g. see feedback from Rev 3) and potentially feature the hairpin deletion analysis in the primary figure given that it represents a new element defined by the crosslinking approach.

We now present the hairpin deletion (delta 610-640) data as main figure 4. We address all specific items in our point-by-point response.

Point-by-point Responses

Reviewer #1 (Comments to the Authors (Required)):

The authors of the paper "Multivalent coiled-coil interactions enable full-scale centrosome assembly and strength" utilize a combination of cross-linking MS, structural modelling, biochemical assays and in vivo studies to characterise the assembly properties of the PCM protein Spd5 from *C.elegans*. Their main conclusions are that Spd5 assembly is driven by hierarchical, multivalent coiled-coil interactions that are (subtly) influenced by phosphorylation and are partly interdependent with microtubule-mediated pulling forces on the PCM.

Overall, I do not recommend publication in JCB, but consider this paper more suitable for a specialised journal.

The centrosome PCM is a dense meshwork that mainly consists of large coiled coil proteins, making the main conclusion of the paper (hierarchical multivalent coiled coil interactions) rather obvious to the point of being trivial. It is hard to see how PCM assembly could work differently given the known facts about the PCM, the proteins that constitute it and the behaviour of coiled coils. If a convincing high-resolution view of the underlying interactions were provided, this would be different, but the corresponding data are not particularly strong (see below). Furthermore, Spd5 is nematode-specific, lowering the appeal to a general readership.

We can appreciate that coiled-coils domains have been described before to mediate assembly of stoichiometric protein complexes. But, it is unclear how they could enable supramolecular, non-stoichiometric assemblies. Previous models only proposed the existence of two specific interactions as driving assembly of the PCM scaffold (the PReM-PReM and PReM-CM2). Whether more modules exist, and if other interaction types (e.g., interactions between disordered regions), was never tested. Thus, we strongly feel that our model is not obvious, and at the very least, never experimentally tested. The assembly principles we define for SPD-5 likely will apply to PCM scaffold proteins in other species, as they have similar secondary structural architecture. Other supramolecular structures form via coiled-coil domains (e.g., endocytic sites, see Kozak and Kaksonen, 2022). Thus, we believe that our findings are relevant to the centrosome field and the broader cell biology community.

Potentially interesting is the questions how the assembly is regulated and the authors' work on Spd5 phosphorylation could be important in this regard. However, this aspect of the paper is not very convincing. It is unclear what phosphorylation sites are functionally critical and the subtle changes in the observed cross-links are also not controlled for well. Phosphorylation introduces a bulky extra group and a charge, so phosphorylation can be expected to subtly influence the (loose) coiled coil packings *in vitro*. I suspect that many S/T kinases other than PLK1 added in the corresponding *in vitro* assay would show something similar (no such controls are provided).

The specificity of PLK-1 for SPD-5, as well as the most important phosphorylation sites, have been previously characterized by us (Woodruff et al., 2015) and the Oegema lab (Ohta et al., 2021). Mutation of four key PLK-1 sites impairs SPD-5 assembly *in vivo* and *in vitro*, as observed by ourselves (Woodruff et al., 2015, Wueseke et al., 2016) and the Sugimoto lab (Nakajo et al., 2022). In the current study, our PTM analysis reveals that we identified the same crucial sites identified in these studies. Thus, there is sufficient evidence that use of PLK-1 is warranted and specific.

In our revision, we have provided more insight into how PLK-1 modulates SPD-5 assembly. By capturing monomeric species during our XL-MS experiments, we reveal that SPD-5 has N- and C-terminal intramolecular folds that are opened up by PLK-1, thus unblocking coiled-coil domains so they can interact *in trans*. We feel that these added experiments strengthen the mechanistic conclusions of the study.

The structural modelling also suffers from shortcomings. While AF2 or RosettaFold have been used to predict inter-domain packing, they mostly/often are not doing a great job on it, especially with coiled coil assemblies. The authors do not provide the relevant statistics to judge the prediction confidence (e.g. the PAE values in AF2). Given the high variability of the predictions that the authors report, I suspect that the corresponding values are in fact rather low. The structural models fail to explain the effect of the functionally well-characterised R592K mutation which is a further worry. The authors' models clearly need experimental high-resolution structural validation.

We acknowledge that *in silico* modeling for coiled-coil regions may not be as effective as traditional structural analysis (e.g., crystallography or cryo-EM). In this spirit, we don't present our data as a structure, but as a model and label it as such. In our revision, we assess the quality

of these models in two ways. First, we provide the PAE values for the AF2 prediction in Figure S2F, which show very low error in the regions corresponding to the tetra-helical bundle of the helical hairpin. Second, we overlay the experimentally defined cross-links onto both the Rosetta and AF2 models and report the cross-linking distances in Figure 2J,K. None of our reported cross-links violate the distance constraints of the DMTMM linker, as all links were less than 24 angstroms (this criterion was defined using globular proteins in Leitner et al., 2014). Thus, the cross-links provide experimental validation of the models.

Importantly, our models predicted that the R592K mutation **should not** have any effect on packing or dimerization of the helical hairpin. Our CD and mass photometry experiments showed this to be the true (Figure 3). This provides additional validation of the structural model.

Overall, I also remain unconvinced by the functional testing of the assembly models. Larger coiled coil regions of Spd5 are removed for the in vivo tests instead of using point mutations based on their models. Even if the structural models provided by the authors were entirely wrong, removing such large regions can be easily imagined to change the fold or interactions with other proteins and is therefore hardly a test of the proposed structural models. This needs to be done better.

We agree with the reviewer that an in-depth analysis of each interaction, with the help of point mutations, would greatly improve our understanding of SPD-5 assembly. However, we consider this study to be a more global analysis of the SPD-5 interaction landscape. We feel that defining the structure of each interaction module, and perturbing those interactions with surgical precision, is beyond the scope of this current study.

Finally, I am also overall sceptical of the authors' in vitro assay. The assemblies are driven by the addition of PEG-3000 as a crowding agent, which results in spherical condensates. Such spherical condensates are typically observed in many, many crystallisation screenings of protein constructs / coiled coil constructs, where such liquid-liquid phase separation (driven by the precipitant / protein super-saturation) is a potential precursor of a crystalline state.

Our assay is a convenient and informative way to measure SPD-5 assembly, even if it does not exactly replicate the exact conditions in vivo. In previous studies, we have shown that this reconstitution assay recapitulates key aspects of PCM function: microtubule aster formation, specific binding of client proteins, exclusion of non-centrosome proteins, and regulated changes in material properties (Woodruff et al., 2017; Mittasch et al., 2020). Furthermore, with regard to mutational analysis, we see a good correlation between the in vitro assay and in vivo experiments. In both kinds of assays the C-terminus of SPD-5 is better at assembling than the N-terminus, elimination of CC-Long (a.a.734-918) or the N-terminus reduces assembly, and inclusion of the Helical Hairpin improves assembly.

Particularly worrying in this respect is that almost all Spd5 fragments (bar one without a coiled coil) appear to lead to these condensates that the authors consider to be an equivalent to the situation in the PCM. I find the authors' conclusion that this shows how redundant the coiled-coils act rather optimistic, as it could be equally concluded that the assay might just not measure what it is supposed to. To be convincing, this will require better mutants and a tightly observed

correlation between the in vivo and the in vitro situation.

We realize that the reviewer has misinterpreted our results, likely due to improper description in the main text and our poor choice of image scaling. In the previous manuscript, we emphasized that different SPD-5 truncations can form puncta visible with a light microscope (which is still true). However, the extent to which different SPD-5 fragments self-assemble, and the size of the assemblies, differ by orders of magnitude. To report the results more faithfully, we now show images of SPD-5 assemblies using the same scaling and report the calculated assembly values (Figure 7C). The data show clearly that some regions of SPD-5 (e.g., the C-terminus) are far more important for self-assembly than others. We never meant to suggest that the coiled-coil domains are redundant, as they clearly are not.

Reviewer #2 (Comments to the Authors (Required)):

In this paper, Rios et al. investigate how the key centrosomal scaffold protein in *C. elegans*, SPD-5, builds into a force-resisting assembly that ensures centrosome integrity during mitosis. SPD-5 is known to self-associate and form condensates in vitro and this is promoted by Plk1 phosphorylation. Using cross-linking mass spectrometry, the authors now identify several interactions between different regions of SPD-5 and reveal how this interaction landscape is altered by Plk-1 phosphorylation. In general, Plk-1 promotes interactions between coiled-coil (CC) regions and prevents interactions between disordered linker regions. The authors identify 3 interaction hotspots that they then analyse further: two helices in the central PReM domain, a long C-terminal CC domain (named long-CC), and 4 CC's in the N-term that include the γ -TuRC binding CM1 domain. The authors show that the two PReM helices are predicted to form a hairpin, and that this hairpin homo-dimerises in a phosphorylation-independent manner. Computational prediction analysis suggests that the dimer forms a flared structure and that phosphorylation modulates its compactness and promotes the interaction of this region with CM2, while at the same time reducing the interaction of these helices with other regions of SPD-5. The PReM-CM2 interaction is known to be an important driver of Cnn scaffold assembly in flies, so the authors findings are consistent with these interactions being a conserved feature important for PCM assembly. The authors investigate an R592L temperature-sensitive mutation that had previously been shown to perturb centrosome assembly. R592 is predicted to be on the surface of the hairpin and the authors show it is dispensable for hairpin formation, but important for condensate assembly and interactions between full-length SPD-5 molecules, helping to explain the centrosome assembly phenotypes. In addition, the authors delete a large part of the hairpin to further show its importance for PCM assembly. The authors note that these mutations affect both assembly of SPD-5 into a scaffold and the scaffold's ability to resist forces. The authors then test the importance of the long-CC and the N-terminal CC's. Deletion experiments show their requirement for PCM assembly and strength too, with the long-CC being most important. Finally, the authors purify multiple regions of SPD-5 and test their ability to form condensates - no one fragment can form condensates as well as full-length SPD-5, indicating that multivalent interactions are necessary. Moreover, these final experiments further highlight the importance of the hairpin and long-CC regions.

Overall, the paper is a tour-de-force in characterising the intra and inter-molecular interactions of a key centrosomal component, but I do find the conceptual advance limited considering what is

known about SPD-5 and Plk-1's promotion of its assembly. For me the paper is very interesting and I would not be surprised to see it published in JCB, but for a wider readership the JCB editorial team should consider whether elucidating details on an already known, albeit important, process makes a sufficient advance. While the paper focusses on *C. elegans* SPD-5, it is generally agreed that SPD-5 is the functional homologue of fly Cnn and human CDK5RAP2, despite a limited sequence homology (which is common for *C. elegans* proteins). I do therefore feel that information of SPD-5 will be of interest to the wider centrosome and microtubule communities.

Major comments

1) The mutants analysed clearly affect both PCM assembly and PCM strength (as there is never a full rescue after nocodazole treatment). The authors do tend mention this but they do tend to overstate the effect on - for example, Figure 4C there is barely any rescue in PCM mass (probably the most important measure) for the C-term deletion mutant when adding nocodazole. The authors say the CC-long is "also required for full-scale assembly" but in reality, I think the CC-long is required mainly for PCM assembly, rather than force resistance.

In our revision, we have reworded our statements to emphasize the importance of CC-Long for PCM assembly. Keep in mind that we use three metrics to assess PCM strength: PCM mass, premature PCM fracture, and PCM circularity. Rescue of any of these phenotypes with nocodazole indicates that PCM is unnaturally weak. The reviewer is likely correct that PCM mass is the most significant of the three criteria: slight weakness causes deformation and irregular PCM shape, moderate weakness causes premature PCM fracture, and severe weakness prevents PCM accumulation. As PCM fracture and circularity are completely rescued by nocodazole in the CC-Long mutant, we still believe it is honest to state that deletion of this region affects PCM strength.

2) Is it surprising that the data shows that interactions between CC regions and not disordered linker regions are important for scaffold assembly *in vivo*, when I thought the general consensus was that interactions between disordered regions promote condensate formation/liquid phase separation? The authors assume, and have assumed before, that condensate formation via liquid phase separation is the mechanism for PCM assembly *in vivo*, so this seems to be an important point. Do the authors think that the interactions between CC regions are transient *in vivo*, or are they maintained within the PCM to create a less dynamic and more scaffold-like structure?

As per the advice of other reviewers and the editor, we have reworked our model to remove references to phase separation. While all phase separating systems have multivalency, not all multivalent proteins assemble through phase separation. Since, we have not convincingly tested this idea in a living cell, we avoid mention of phase separation. We instead highlight the role of specific, multivalent interactions between coiled-coil domains.

To answer first question: the idea that only disordered motifs drive phase separation is a widely held misconception. In fact, the very first paper describing the biochemical basis of phase separating proteins used a synthetic peptide with multivalent PRM (proline-rich motifs) and SH3

modules, which are structured motifs that interact with defined stoichiometry (Li et al., Nature 2012). The nucleolus protein NPM1 also phase separates using a combination of disordered regions and an N-terminal structured motif that pentamerizes. Multivalency is really the key parameter, not the nature of the interacting domains.

To answer the second question: We cannot comment on if the CC interactions in SPD-5 are transient or not. We speculate that the coiled-coil interactions mature, possibly through registration, thus effecting a liquid-to-solid transition. We proposed this theory in a previous paper (Woodruff et al., 2017 Cell) and are currently testing this hypothesis in a separate study. Thus, we feel that this study would not benefit from such a discussion, as we are not testing this idea directly.

Minor comments

1) When mentioning that fly Cnn also forms scaffold (lines 42-45), the authors should cite Conduit et al., 2014, Dev. Cell., as this is the first paper showing Cnn forms a centrosomal scaffold. It could also be mentioned that this was the paper that first identified the PRoM domain.

This has been corrected.

2) Line 498: when saying that the phenotype was rescued, I believe this should be changed to "partially rescued".

This has been corrected.

3) The paper is quite difficult to read, not because of bad English but just because several different techniques and so much data are presented. The authors would help the reader by explaining the techniques in more detail before presenting the data.

Other reviewers made similar comments, especially about the XL-MS technique and display of results. We have expanded the first section to include more details about how the XL-MS is performed and how the results are interpreted.

Reviewer #3 (Comments to the Authors (Required)):

Centrosomes function as the cell's primary microtubule-organizing center and consist of a pair of centrioles surrounded by pericentriolar material or PCM. The PCM is the site of microtubule nucleation and anchorage but the structure of the PCM is not well defined. PCM assembly likely involves a large number of molecular interactions among its constituent proteins. The identity of these molecular interactions and how they might be affected by polo like kinase 1 which promotes PCM assembly is not known. Likewise almost nothing is known about, how these molecular interactions might help the PCM resist microtubule-dependent forces.

Here the authors use the nematode *C. elegans* as a model system to study the assembly and material properties of the PCM. *C. elegans* PCM is thought to be composed almost entirely by the coiled-coil protein SPD-5. The authors use cross-linking mass spectrometry to map

interactions sites between and within SPD-5 molecules in vitro and analyze how phosphorylation by PLK-1 leads to changes in the spectrum of interactions. They find that most interactions involve the coiled-coil regions of SPD-5 and that PLK-1 phosphorylation reduces the overall number of interactions, particularly those in linker regions while promoting interactions between coiled-coil domains. They also investigate the role of various SPD-5 motifs (helical hairpin, a long coiled-coil domain and the n-terminus) in PCM assembly and strength. Finally they analyze nine fragments of SPD-5 in vitro for the ability to form or be recruited into supramolecular structures. Their findings support the idea that multivalent interactions among the SPD-5 coiled-coil domains are a key feature of PCM assembly and strength.

From a technical standpoint, this study is truly impressive. The experiments, particularly the in vitro work, is well thought out and executed. My problem is that I don't think the results offer a significant advance in understanding PCM assembly or strength. The authors do describe the various intra and inter molecular interactions that take place before and after PLK-1 phosphorylation of SPD-5 in vitro, and that is important data, but the rest of the paper is concerned with a structure-function type approach to identify various SPD-5 motifs required for assembly or strength of the PCM. For instance, the helical hairpin and the long-cc are explored but disruption of either one of these structures doesn't even affect embryonic viability, meaning that they don't affect essential processes, such as spindle assembly or position. Moreover, all of the mutations they study strongly affect PCM assembly with only minimal effects on PCM strength (despite what the authors seem to argue in the text). This is not surprising as it seems obvious to me that the protein-protein interactions that drive assembly are also likely to impart material strength to the matrix. So what they really are studying here is mostly the interactions that drive PCM assembly but there just aren't any really exciting findings that make me feel I can strongly endorse this work for publication in JCB. I'll keep an open mind though and if the other reviewers feel differently, I might reconsider. Below are my recommendations.

We thank the reviewer for the comments on the technical aspects of this study. Regarding novelty of the study, we feel that comprehensively mapping the interactions underlying SPD-5 assembly with XL-MS is a significant achievement. This work reveals that multivalent coiled-coil interactions drive PCM scaffold assembly and that disordered linkers prevent assembly through auto-inhibition. This information will guide future studies, in terms of 1) studying structures within SPD-5 and 2) establishing the methods and metrics to analyze PCM scaffold proteins in other species, such as Centrosomin and CDK5RAP2.

By following the advice of this reviewer, we now provide structural information about how PLK-1 regulates the conformation of SPD-5, thus revealing the molecular basis of how phosphorylation potentiates assembly of the PCM scaffold. We feel this raises the significance of this study, as PLK-1-mediated multimerization of scaffold proteins is a conserved feature of PCM assembly.

Major revisions:

Line 143: I think it would be helpful here to describe the capabilities and limitations of XL-MS. For instance I found myself immediately wondering if the approach was capable of distinguishing intramolecular vs intermolecular interactions but had to wait until later in the

manuscript to find this out. Also I had questions about the quantitative nature of the approach. Are the authors able to identify the number of times a specific interaction occurs per unit mass of oligomer? Just a little bit of background here would go a long way in helping the reader appreciate the data.

We agree with the reviewer on this point and made significant effort to distinguish intra and intermolecular contacts. In the revised Figure 1, you will see that we were able to adjust our conditions to capture SPD-5 monomers (those that contain only intra-molecular interactions). Subtracting out these bona fide intra-molecular interactions from our multimer interaction map allows us to identify interactions that are highly likely to be inter-molecular.

Analysis of intra-molecular x-links revealed that unphosphorylated SPD-5 is folded on itself. Phosphorylation releases these interactions, thus opening SPD-5 so the coiled-coil domains can participate in inter-molecular interactions. This data provides structural evidence that SPD-5 is auto-inhibited, which can be relieved by PLK-1 phosphorylation. This model explains why PLK-1 potentiates SPD-5 assembly. This experiment has really added value to the study, and we thank the reviewer for the suggestion.

We also add comments on the quantification of the X-links in the main text. We typically report “occurrence”, which represents the number of replicates that contain that specific X-linked pair. For example, we performed six replicates. An occurrence of 6 means that the X-link was confidently assigned in all six replicates. High occurrence is typical of stable interactions within a folded domain, while low occurrence is typical of transient interactions.

The methods section describes in detail the analysis of peptide masses and evaluation of confidence and error (e.g., Protein ID score and False Discovery Rate). The amount of oligomer mass is constant between samples, so yes, our results are estimates of the number of X-links per unit mass. We provide more detailed information in the supplemental material, which shows the exact peptides found and their properties. Our raw MS files are also available on MASSIVE (links are listed in the Data Availability section).

Line 161. The authors state that interacting residues had overall fewer observed crosslinked partners in the phosphorylated state. How do the authors interpret this finding? Does this mean that phosphorylation results in a less dynamic PCM or does this mean that there are simply fewer static interactions allowed in the phosphorylated state? Or both?

In the revised manuscript, we now report the intra-molecular interactions. PLK-1 significantly reduced intra-molecular folding of SPD-5 (Figure 1D,E). Thus, much of the reduction in X-link number is due to the relief of auto-inhibitory folding. Secondly, we think that PLK-1 phosphorylation reduces promiscuous inter-molecular interactions between SPD-5. This is revealed when looking at the number of partners per residue (Figure S1H); PLK-1 reduces the overall partners that each residue has. We comment on this in the main text and discussion.

Line 256: The authors wish to test the role of the helical hairpin in PCM strength and assembly. For this they use the spd-5(or213) allele that converts arginine 592 to a lysine (R592K). However it is clear that the mutation does not affect the dimerization, stability or the structure of the

hairpin. Furthermore, as shown later in the paper, disrupting the hairpin by deleting a portion of the second helix does not have effects as severe as the R592K mutant. Thus the authors can't attribute the effects of the R592K mutation on a disruption of the hairpin. This means that the statement on line 315 that says the hairpin is required for full-scale assembly is incorrect.

We see the reviewer's point and have changed the text to:

"We conclude that residues within the helical hairpin are critical for SPD-5 to assemble into a strong scaffold that resists microtubule-mediated forces."

(lines 363-364)

Figure 3A and B. At what stage are the embryos being analyzed and how do the authors determine this stage? PCM levels change dramatically during embryogenesis and to properly compare embryos, they all need to be at the exact same stage. The wild-type embryo shown in panel A appears to be a mitotic two-cell embryo while the *spd-5(or213)* embryos appear to be one cell (based on centrosome number). The authors should stain for DNA so that the cell cycle stage can be determined properly.

We have since repeated this experiment and analyzed only 1-cell embryos that had begun or completed nuclear envelope breakdown (Figure 3A,B). Our conclusion remains the same: PCM size in R592K embryos can be rescued by application of nocodazole, indicating that SPD-5(R592K) can build weak PCM.

Also the authors state that failed PCM assembly in *spd-5(or213)* embryos is largely due to PCM weakness. I would say this is an overstatement as nocodazole-treatment only partially rescued PCM levels, yielding about half as much as controls.

We have reworded our conclusion to emphasize the role of R592 in PCM assembly.

We still think our data support the conclusion that R592K also affects PCM strength. Under physiological pulling forces, SPD-5(R592K) cannot make mitotic PCM at all (the small interphase layer is still there). Yet, eliminating pulling forces allows it to assemble PCM. This rescue corresponds to a 5-fold increase, on average, in PCM mass in the *spd-5(or213ts)* strain. It may not be complete rescue, but this result is dramatic. To us, this result is also unanticipated: who would have guessed that a severe PCM assembly phenotype could be rescued by nocodazole? This result reveals that is important to consider mechanical properties when assessing PCM assembly phenotypes.

Finally, the authors need controls showing that nocodazole-treatment consistently disrupts the microtubule cytoskeleton. The P1 blastomere of the wild-type nocodazole-treated embryo shown in panel A looks like it has a normal sized and correctly positioned spindle, which you would not expect if microtubules were depolymerized.

For Figure 3A (immunofluorescence), we repeated the experiments. To check the efficacy of nocodazole, we stained for alpha tubulin (not shown in Figure 3 for sake of clarity; however, we

mention this in the methods). We found that not all embryos were permeable. Thus, for the nocodazole experiment, we only measured those where microtubules were clearly depolymerized.

For Figure 3C (live cell imaging), we use spindle collapse and centrosome drifting as indicators that nocodazole is working. In the revised manuscript, we rechecked our data to ensure that nocodazole worked as expected in each experiment.

Figure 3C and D. The same criticisms that apply to panels A and B apply here. However here the authors find that nocodazole only weakly rescues the PCM accumulation defect of the *spd-5* mutant (I estimate that nocodazole-treatment of the mutant results in only 20% as much PCM as controls based on panel D). Despite this, on lines 282-287, the authors seem to be emphasizing the effect of the R592K mutation on the strength of the SPD-5 scaffold. Looking at the data, it is clear that this mutation primarily affects assembly and not strength and the text should reflect that.

The text now emphasizes that R592K primarily affects assembly and secondarily affects strength.

Line 328. The authors refer to a western blot to point out that wild-type SPD-5 and SPD-5(Δ 610-640) are expressed at similar levels. First the correct panel is S3F, and not S3E as stated. Second, it looks to me like the SPD-5(Δ 610-640) protein is expressed at twice the level of the wild-type. Again this is one of several places where the text doesn't accurately reflect the results.

We have corrected this mistake. The new text now says:

“This phenotype was not due to poor expression of mutant SPD-5, as western blotting revealed mutant protein was expressed at a higher level than control protein (Figure S4B).”

(lines 354-355)

Lines 319-229 and Figure S3G, H, and I. The authors disrupt the helical hairpin by deleting about 31 residues of the second alpha helix. Unlike the R592K mutant, his mutant is viable, but has very low levels of PCM that are only very weakly rescued by nocodazole treatment. Based on the author's own rationale for using nocodazole to test the material strength of the SPD-5 network, I would say given the results shown in Figure S3H, that the helical hairpin plays only a modest role in PCM strength and that it's predominantly required for PCM assembly. Again, the

conclusions on lines 335-339 don't line up accurately with the data as the authors stress a role for this motif in resisting microtubule-mediated pulling forces.

The text now emphasizes that deletion of a.a.610-640 primarily affects assembly and secondarily affects strength.

Also in thinking about the effects of the SPD-5(Δ 610-640) mutant on PCM size, I began to wonder if a diminutive PCM might experience relatively higher pulling forces. That is, a wild-type PCM nucleates a certain percentage of microtubules that grow long enough to reach the cell cortex, where pulling forces are concentrated. However a small PCM might nucleate a higher percentage of long microtubules that make it to the cortex. This might occur because the free tubulin pool is increased due to a reduction in overall microtubule nucleation. The result could be a higher pulling force per unit area (or mass, not sure which is most appropriate) of PCM for smaller centrosomes. This of course could affect interpretation of the nocodazole experiments. Do the authors have any way to measure the pulling forces on the variously sized centrosomes?

We know from previous studies that smaller centrosomes don't experience larger forces, which can be estimated by measuring spindle rocking (Woodruff et al., 2015; Wueseke et al., 2016). Nevertheless, we show that spindle rocking is similar between control and SPD-5(Δ 610-640) embryos (Figure S4C). Thus, larger forces are not the cause of the mutant phenotype.

Line 394. The authors state that embryos expressing SPD-5(WT) and SPD-5(566-1198) do not exhibit PCM fragmentation or premature disassembly. Where is the data for this?

This was an oversight. We have included the data in Figure 6F and S6C.

Line 402. The authors state that their finding that the N-terminus of SPD-5 is necessary for PCM assembly is inconsistent with a prior study showing that an N-terminal fragment of SPD-5 fails to assemble on its own. Why is this inconsistent? The experiments that the authors mention test two different things: necessity and sufficiency. Deletion of the N-terminus tests for necessity while expression of a n-terminal fragment tests for sufficiency. Therefore there is no inconsistency.

We have corrected this. The new text states:

"We then tested if the N-terminus is sufficient to assemble into supramolecular scaffolds. We used MosSCI to create embryos expressing N-terminal SPD-5 (a.a. 1-566) tagged with GFP (Figure 6G and S6A,B)."
(Lines 431-433)

Figure 3I. What temperature is this assay done at? Can the authors recapitulate the ts properties of the R592K mutant?

This experiment was performed at 23° C, as stated in the methods.

We don't think it is sensible to test the in vitro protein has temperature-sensitive properties for the following reason. We found that our MosSCI line expressing GFP::SPD-5(R592K) was **not**

temperature sensitive (Figure S3C). In the absence of endogenous *spd-5*, this strain is completely inviable at both 16° C and 25° C. Thus, we think the original *spd-5(or213)* strain may have a suppressor that partly rescues viability (68% viability, Hamill, 2002) at lower temperatures.

We comment:

“Unexpectedly, unlike *spd-5(or213)* embryos, the MosSCI mutant line was not temperature-sensitive: embryos expressing only SPD-5(R592K) were inviable at 16°C and 25°C (Figure S3C). The basis of this difference is unclear, but we suspect that *spd-5(or213)* worms contain a genetically linked suppressor mutation that partly restores viability at 16°C.”

(Lines 307-311)

Figure S3A. Were these animals grown at the nonpermissive temperature? Please specify.

Yes. The figure legend has been revised to include temperature.

Minor revisions:

Figure 1. Could the authors include the PReM domain in the SPD-5 schematics?

This has been corrected.

Line 359. The authors cite Figure 4A-C when they should have referenced 4A, B and D.

The figure panels have been rearranged to match the text.

Line 363. The authors cite Figure 4D when they should have cited 4C.

The figure panels have been rearranged to match the text.

Reviewer #4 (Comments to the Authors (Required)):

Centrosomes are composed of a centriole surrounded by a pericentriolar material matrix (PCM matrix). The PCM matrix is primarily composed of molecules of the CDK5RAP2 family (CDK5RAP2 in humans, Cnn in *Drosophila* and SPD-5 in *C. elegans*) which contain a series of alpha-helices that are predicted to form coiled-coils. During mitotic entry, the PCM matrix expands via Polo-like kinase 1 (PLK1) phosphorylation-dependent self-assembly of PCM matrix molecules. Major questions in the field are how the PCM matrix is organized, both how the PCM matrix molecules are structured and how they interact with each other to form a matrix, and how the assembly process is controlled by phosphorylation. Prior to this work, the authors developed an in vitro assembly system for the *C. elegans* matrix molecule SPD-5, whose assembly in vitro was augmented by PLK1 phosphorylation. In this manuscript, they combine this in vitro system with a crosslinking-based approach to try to understand how the PCM matrix is organized. They compare SPD-5 matrix assembled in the presence versus the absence of PLK1 phosphorylation and show that they get fewer crosslinks (and a higher percentage of crosslinks between predicted coiled-coil domains in the presence of phosphorylation, suggesting that phosphorylation enhances the specificity of SPD-5-SPD-5 interactions. The work goes on to characterize several regions of SPD-5 that are important for PCM assembly and/or strength; these regions include a

nice characterization of SPD-5 residues 541-677, which they show forms a helical hairpin that dimerizes to form a tetrameric coiled-coil and a second a long coiled-coil region.

Overall, the authors employ a combination of interesting approaches to tackle the difficult question of how the PCM matrix assembles around centrioles. I would support publication in the JCB if the points below can be addressed.

Main points:

1. The approach of cross-linking PCM is interesting and potentially useful. However, as it is presented here, it is quite challenging to interpret. This is because it appears that individual SPD-5 molecules may be folding back onto themselves, sort of like intramolecular origami, with an example being the coiled-coil "hairpin" that the authors describe in Fig. 2. On top of this, there is a second layer of inter-molecular interactions between SPD-5 molecules that hold together the assembled PCM network. In the crosslink maps in Fig. 1D and E these layers are mixed together, making it impossible to know how much represents molecular folding and how much represents intermolecular proximity within the assembled networks. The authors mention in passing that they could not tease apart these layers by sub-saturation crosslinking for full-length SPD-5, as they did for the smaller SPD-5 (541-677) construct in Fig. 2E-G, to identify the intra-molecular versus inter-molecular cross-links. If this cannot be done in the context of full-length SPD-5, could it be done in the context of a few smaller regions of the molecule (for example the three crosslinking "hotspots" that the authors identify (aa 1-566, 541-677 and 734-918)? ---noting that the authors already do a nice job of this for the second hotspot aa 541-677 in Fig. 2. If not, the point that these two types of crosslinks cannot be distinguished should be more extensively discussed both in the section where the crosslinking is described and in the discussion.

We agree with the reviewer on this point and made significant effort to distinguish intra and intermolecular contacts in full-length SPD-5. In the revised Figure 1, you will see that we were able to adjust our conditions to capture SPD-5 monomers (those that contain only intra-molecular interactions). Subtracting out these bona fide intra-molecular interactions from our multimer interaction map allows us to identify interactions that are highly likely to be inter-molecular.

Analysis of intra-molecular x-links revealed that unphosphorylated SPD-5 is folded on itself. Phosphorylation releases these interactions, thus opening SPD-5 so the coiled-coil domains can participate in inter-molecular interactions. This data provides structural evidence that SPD-5 is auto-inhibited, which can be relieved by PLK-1 phosphorylation. This model explains why PLK-1 potentiates SPD-5 assembly. This experiment has really added value to the study, and we thank the reviewer for the suggestion.

2. In vivo, SPD-5 only assembles if it is phosphorylated by PLK1. Thus, the meaning of the comparison between the physiological assemblies that form from phosphorylated SPD-5 and non-physiological assemblies that form in vitro from SPD-5 that has not been phosphorylated (which based on prior work from this group appears to require higher concentrations for assembly compared to phosphorylated SPD-5) is not very clear. It seems to reveal that there is a

significant increase in promiscuous crosslinks between disordered linker regions if non-phosphorylated SPD-5, which would not normally assemble in vivo, is forced to do so. One might expect that this could be molecules in an autoinhibited configuration forced together by promiscuous linker interactions. Given that the significance of this material is not clear, it seems like the manuscript might be strengthened by moving some of the panels related to this in Fig. 1D-I to the supplement to better highlight the crosslinking map for the potentially physiologically relevant material. For example, in panel F it is difficult to see the map of the pink (phosphorylated SPD-5) crosslinks because of the blue (not phosphorylated SPD-5) crosslinks that is superimposed over it.

We have followed the advice of the reviewer and moved these figure panels to the supplement and limited our description of these experiments in the main text. We now emphasize the multimeric interactions of phosphorylated SPD-5 in Figure 1.

We still believe that analysis of unphosphorylated SPD-5 multimers is informative, as there are cases where SPD-5 can weakly interact without PLK-1. In the embryo, SPD-5 phospho-mutant still binds to a pre-assembled PCM (Wueseke et al., 2016 Biology Open). SPD-5 assembles slowly in vitro without PLK-1 (Woodruff et al., 2015 Science) and its CM2-like and PReM domains interact weakly without PLK-1 (Nakajo et al., JCS 2022).

3. Lines 177-182: "In addition, we identified previously uncharacterized interaction sites, including three hotspots: 1) two helices within the PReM region (a.a. 541-677), 2) a C-terminal, long coiled-coil domain (a.a. 734-918), and 3) four coiled-coil domains in the N-terminus (a.a. 1-566)." Looking at the map in Fig. 1D, the region between aa 1 and 345 and the region between 694 and 984, do indeed appear to be cross-linking hot spots. However, the region between 541 and 677 does not appear to be a crosslinking hot spot, should this region be included in this list?

We see now that "hotspot" may not be the correct terminology. We now call them "regions of interest". The helical hairpin was interesting because it had cross-links with high occurrence across all samples. The other two regions were interesting because they contained many interactions.

As per point 1, it would be interesting to know how many of the cross-links in the two hotspot regions that are not characterized in further detail (1-566 and 734-918) are predicted to be intramolecular, perhaps the crosslinking data is revealing much more extensive intramolecular folding than expected.

We show the XL-MS map of the SPD-5 monomers in the revised Figure 1, which shows the extent of intramolecular interactions in these regions.

We also show the map of intermolecular interactions in the phosphorylated SPD-5 multimers in our revised Figure S1B. We state the numbers of intermolecular cross-links in these areas in the text:

"In the SPD-5 multimer sample (+PLK-1(CA)), we mapped 119 intermolecular cross-linked pairs where at least one residue resided within a predicted coiled-coil domain spanning a.a. 734-918"

“Our XL-MS of phosphorylated SPD-5 multimers identified 135 intermolecular cross-linked pairs where at least one residue resided within a.a. 1-566.”

4. Lines 198-201: "We identified short-range links within this region (e.g., E602-K616, E601-K618) and long-range links with other regions of SPD-5. Phosphorylation reduced the number of contacts overall originating from this region and created a new contact with the CM2-like domain (E665-K1160)(Figure S2A)." The crosslinks to this region, which are highlighted in Fig. S2 are not prominent in Fig. 1D. This seems to be because these crosslinks were only observed in one occurrence, as opposed to other crosslinks that were observed in 2+ occurrences. Are these crosslinks reliable-should any crosslinks that were only observed 1 time be featured in the manuscript.

We choose to display all cross-links that pass our confidence criteria for analyzing MS data, such as protein ID score and false discovery rate (discussed in the methods). We think that occurrence is a metric of binding affinity. High occurrence (e.g., 5-6 out of 6 replicates) likely corresponds to folded domains with stable interactions (e.g., the helical hairpin). Low occurrence (1 out of 6) likely corresponds to transient, weak interactions. We make this point in the text. This assumption is based on our own XL-MS of folded proteins (BSA, chaperones) and intrinsically disordered proteins like Tau (PMID: 31175300; PMID: 34504072).

5. In Fig. 2F,G, is it possible to color-code the crosslinks in the intra + inter-molecular maps on the right so that the intramolecular ones are one color and the ones not in the intra-molecular maps on the left are a second color? This would make it easier to spot the potential inter-molecular crosslinks.

We have colored the inter-molecular cross-links in red.

6. In general, I like the analysis in Figures 2-4 testing the role of the various regions in PCM assembly/strength. However, it is not clear that deletion of the N-terminal region of SPD-5 (the 566-1198 construct) can be analyzed in the same way. This N-terminal region of SPD-5 has previously been shown to interact with SPD-2, AIR-1 and the RSA complex (PMID:18692475) and is thought to play a signaling role in PCM assembly by delivering active PLK1 kinase to the PCM (PMID: 21802300). Thus, removing the SPD-2 binding site might prevent SPD-5 assembly in vivo by preventing SPD-5 phosphorylation, rather than due to loss of structural elements that contribute to PCM assembly via interaction with other structural elements. Thus, it cannot really be concluded that the N-terminal coiled-coils structurally contribute to SPD-5 self-assembly. The idea that the N-terminal coiled-coils do not directly mediate SPD-5 self-assembly is supported by the in vitro data shown in Figure 6, which show that the N-terminal constructs F20-F23 do not assemble well on their own and do not associate with pre-assembled PCM, suggesting that the helicies in this region do not make promiscuous contacts with helicies in the remainder of SPD-5. In addition this data shows that the F24 construct which contains only the C-terminus-assembles in vitro (which PLK1 delivery is likely not limiting) in a fashion that is pretty comparable to full-length SPD-5. These points should be discussed in the text.

We acknowledge the reviewer's point, and we now discuss this in the text.

Still, our XL-MS data indicate that the N-terminus directly participates in inter-molecular SPD-5:SPD-5 interactions. These interactions matter, as the in vitro F24 construct cannot self-assemble or bind as well as full-length protein (about 60% as well, similar to the F28 construct that lacks CC-Long; updated Figure 7). In both the results and discussion, we emphasize that the C-terminus is far more important for multimerization compared to the N-terminus (as shown in Figure 7). We also see the reviewer's point about SPD-5 binding partners and state:

"It is possible that SPD-5 clients, which bind to the N-terminus (e.g., SPD-2, AIR-1, RSA-2)(Boxem et al., 2008), provide heterotypic interactions that also contribute to SPD-5 multimerization. Future work should characterize how these clients bind to SPD-5 and tune its multimerization."

(lines 545-548)

As intuited by the reviewer, the binding with other clients could explain how SPD-5(1-566) assembles into supramolecular structures in multi-cell embryos (Figure 6).

7. Although, the data in the paper is interesting and important, it does not provide support for the multivalent "stickers" and "linkers" model outlined in Figure 7, which seems to imply promiscuous association of different coiled-coil regions with multiple binding partners. There is no data in the paper that supports the promiscuous interaction of a coiled-coil region with multiple other coiled-coil regions. Instead, when the authors dig down into a structural element that they identify as important for PCM assembly they find evidence for very specific interactions-identifying a pair of adjacent helicies that fold back on each other to form a helical hairpin that subsequently dimerizes to form a tetrameric coiled coil. Thus, one implication of the data is that there may be more intramolecular folding of the SPD-5 helicies than previously anticipated-which could be involved in both autoinhibition of the non-phosphorylated protein and in key structural elements within the interacting phosphorylated protein. The authors further show that phosphorylation suppresses the promiscuous interaction of linkers and show that the helicies in the N-terminal region of SPD-5 do not drive significant assembly or promiscuously interact with assembled PCM. The "stickers and linkers" model and accompanying discussion should be removed and replaced with a more accurate discussion of the data and significance of the findings in the paper.

In the revised discussion, we have removed mention of promiscuous stickers and spacers and instead refer to SPD-5 assembly being mediated by specific multivalent interactions between coiled-coil domains. We now include discussion of how PLK-1 releases the auto-inhibited configuration of SPD-5, thus priming it for self-assembly. The model in Figure 8 has been updated accordingly.

(lines 512-525)

Minor comments:

1. In the first paragraph of the introduction, it feels like different types of things are being mixed and matched (1) mitotic PCM in which PCM matrix molecules self-assemble in a PLK1 phosphorylation-dependent fashion, (2) interphase PCM which usually assembles in a single layer around centrioles (PMID: 36107993), but also seems to be present at the base of cilia when

centrioles are absent (PMID: 33798428, PMID: 33798427) and (3) the microtubule-nucleating centers in meiotic mouse oocytes. Also, relevant here is recent work suggesting that there are at least two robust microtubule generation pathways anchored at centrioles, one of which is independent of the PCM matrix molecules (CDK5RAP2/Cnn/SPD-5; PMID: 33170211; <https://www.biorxiv.org/content/10.1101/2022.09.23.509043v4>). Overall, the intro would be much stronger if it stayed focused on the mitotic self-assembling CDK5RAP2/Cnn/SPD-5 type matrix that is the subject of the paper.

We changed the introduction to focus on mitotic self-assembling CDK5RAP2/Cnn/SPD-5 and how it is regulated by PLK-1. We included the Conduit reference when discussing how PCM concentrates clients that nucleate MTs.

2. Lines 127-128 "Alphafold predicts that SPD-5 contains 14 alpha helices connected by disordered linker regions (Jumper et al., 2021)". It may have to do with the overlapping orange and yellow colors, but I only see 12 alpha helices indicated on Fig. 1B.

These regions overlap. Coiled-coil domains contain the alpha helices. For the sake of simplicity, the coiled-coil regions are arranged on top, which unfortunately obscures the number of predicted alpha helices. We chose this arrangement since coiled-coil domains are the primary focus of this study.

3. Please label SPD-5 amino acid numbers on the SPD-5 diagram in Fig. 1E as well as in Fig. 1D.

This has been corrected.

4. Discussion of the PReM region in the intro should include a reference to the paper where it was originally defined in *Drosophila* Cnn (PMID:24656740).

This has been corrected.

5. The reference on line 81 to Feng et al., 2017, should also include a reference to PMID: 35362532.

We have reworded this section to discuss Cnn first, then SPD-5 second. Thus, PMID: 35362532 is included when referencing SPD-5.

"Specifically, regulated assembly of Centrosomin occurs via phosphorylation-driven oligomerization of a central coiled-coil region, termed "PReM", (Conduit et al., 2014a) and its interaction with the C-terminal CM2 domain (Feng et al., 2017). Yeast-2-hybrid analyses and pull-downs of SPD-5 fragments showed a similar interaction between the CM2-like domain and a central region containing coiled-coil domains and key PLK-1 phosphorylation sites (a.a. 272-732, which the authors termed the "PReM region")(Nakajo et al., 2022)."

December 20, 2023

RE: JCB Manuscript #202306142R-A

Dr. Jeffrey B Woodruff
The University of Texas Southwestern Medical Center
Cell Biology
6000 Harry Hines Blvd
Dallas, TX 75235

Dear Dr. Woodruff,

Thank you for submitting your revised manuscript entitled "Multivalent coiled-coil interactions enable full-scale centrosome assembly and strength." The manuscript has been re-assessed by three of the original reviewers who are all enthusiastic about the work. The reviewers do have a number of comments asking for text and figure changes to improve presentation and discussion which will need to be fully addressed in the final version. We would be happy to publish your paper in JCB pending these final changes as well as revisions necessary to meet our formatting guidelines (see details below). Please note that we will likely ask some of the reviewers for feedback on the final version.

A. MANUSCRIPT ORGANIZATION AND FORMATTING:

1) Text limits: Character count for Articles is < 40,000, not including spaces. Count includes title page, abstract, introduction, results, discussion, and acknowledgments. Count does not include materials and methods, figure legends, references, tables, or supplemental legends.

2) Figure formatting: Articles may have up to 10 main text figures. Scale bars must be present on all microscopy images, including inset magnifications. Molecular weight or nucleic acid size markers must be included on all gel electrophoresis. Please add scale bars to magnifications in Figures 3A/C, 4A, 5A/E, & 6D.

Also, please avoid pairing red and green for images and graphs to ensure legibility for color-blind readers. If red and green are paired for images, please ensure that the particular red and green hues used in micrographs are distinctive with any of the colorblind types. If not, please modify colors accordingly or provide separate images of the individual channels.

3) Statistical analysis: Error bars on graphic representations of numerical data must be clearly described in the figure legend. The number of independent data points (n) represented in a graph must be indicated in the legend. Please, indicate whether 'n' refers to technical or biological replicates (i.e. number of analyzed cells, samples or animals, number of independent experiments). If independent experiments with multiple biological replicates have been performed, we recommend using distribution-reproducibility SuperPlots (please see Lord et al., JCB 2020) to better display the distribution of the entire dataset, and report statistics (such as means, error bars, and P values) that address the reproducibility of the findings.

Statistical methods should be explained in full in the materials and methods. For figures presenting pooled data the statistical measure should be defined in the figure legends. Please also be sure to indicate the statistical tests used in each of your experiments (both in the figure legend itself and in a separate methods section) as well as the parameters of the test (for example, if you ran a t-test, please indicate if it was one- or two-sided, etc.). Also, if you used parametric tests, please indicate if the data distribution was tested for normality (and if so, how). If not, you must state something to the effect that "Data distribution was assumed to be normal but this was not formally tested."

4) Materials and methods: Should be comprehensive and not simply reference a previous publication for details on how an experiment was performed. Please provide full descriptions (at least in brief) in the text for readers who may not have access to referenced manuscripts. The text should not refer to methods "...as previously described." Please also indicate the acquisition and quantification methods for immunoblotting/western blots.

5) For all cell lines, vectors, constructs/cDNAs, etc. - all genetic material: please include database / vendor ID (e.g., Addgene, ATCC, etc.) or if unavailable, please briefly describe their basic genetic features, even if described in other published work or gifted to you by other investigators (and provide references where appropriate). Please be sure to provide the sequences for all of your oligos: primers, si/shRNA, RNAi, gRNAs, etc. in the materials and methods. You must also indicate in the methods the

source, species, and catalog numbers/vendor identifiers (where appropriate) for all of your antibodies, including secondary. If antibodies are not commercial, please add a reference citation if possible.

6) Microscope image acquisition: The following information must be provided about the acquisition and processing of images:

- Make and model of microscope
- Type, magnification, and numerical aperture of the objective lenses
- Temperature
- Imaging medium
- Fluorochromes
- Camera make and model
- Acquisition software
- Any software used for image processing subsequent to data acquisition. Please include details and types of operations involved (e.g., type of deconvolution, 3D reconstitutions, surface or volume rendering, gamma adjustments, etc.).

7) References: There is no limit to the number of references cited in a manuscript. References should be cited parenthetically in the text by author and year of publication. Abbreviate the names of journals according to PubMed.

8) Supplemental materials: Articles are generally allowed have up to 5 supplemental figures and 10 videos. You currently exceed this limit but, in this case, we will be able to give you the extra space but please try not to add to the current total. Please also note that tables, like figures, should be provided as individual, editable files. A summary of all supplemental material should appear at the end of the Materials and methods section. Please include one brief sentence per item.

9) eTOC summary: A ~40-50 word summary that describes the context and significance of the findings for a general readership should be included on the title page. The statement should be written in the present tense and refer to the work in the third person. It should begin with "First author name(s) et al..." to match our preferred style.

10) Conflict of interest statement: JCB requires inclusion of a statement in the acknowledgements regarding competing financial interests. If no competing financial interests exist, please include the following statement: "The authors declare no competing financial interests." If competing interests are declared, please follow your statement of these competing interests with the following statement: "The authors declare no further competing financial interests."

11) A separate author contribution section is required following the Acknowledgments in all research manuscripts. All authors should be mentioned and designated by their first and middle initials and full surnames. We encourage use of the CRediT nomenclature (<https://casrai.org/credit/>).

12) ORCID IDs: ORCID IDs are unique identifiers allowing researchers to create a record of their various scholarly contributions in a single place. Please note that ORCID IDs are required for all authors. At resubmission of your final files, please be sure to provide your ORCID ID and those of all co-authors.

13) JCB requires authors to submit Source Data used to generate figures containing gels and Western blots with all revised manuscripts. This Source Data consists of fully uncropped and unprocessed images for each gel/blot displayed in the main and supplemental figures. Since your paper includes cropped gel and/or blot images, please be sure to provide one Source Data file for each figure that contains gels and/or blots along with your revised manuscript files. File names for Source Data figures should be alphanumeric without any spaces or special characters (i.e., SourceDataF#, where F# refers to the associated main figure number or SourceDataFS# for those associated with Supplementary figures). The lanes of the gels/blots should be labeled as they are in the associated figure, the place where cropping was applied should be marked (with a box), and molecular weight/size standards should be labeled wherever possible. Source Data files will be directly linked to specific figures in the published article.

14) Journal of Cell Biology now requires a data availability statement for all research article submissions. These statements will be published in the article directly above the Acknowledgments. The statement should address all data underlying the research presented in the manuscript. Please visit the JCB instructions for authors for guidelines and examples of statements at (<https://rupress.org/jcb/pages/editorial-policies#data-availability-statement>).

B. FINAL FILES:

Thank you for your attention to these final processing requirements. Please contact the journal office with any questions at cellbio@rockefeller.edu.

Thank you for this interesting contribution, we look forward to publishing your paper in Journal of Cell Biology.

Sincerely,

Arshad Desai, PhD
Monitoring Editor
Journal of Cell Biology

Dan Simon, PhD
Scientific Editor
Journal of Cell Biology

Reviewer #2 (Comments to the Authors (Required)):

The authors have addressed my original concerns and appear to have done a good job at addressing those of the other reviewers. Addressing how PCM proteins are regulated is not easy and I like the combination of techniques used by the authors. Now that they have better explained how they can distinguish intra vs inter molecular crosslinks, the conclusions are solid. Overall, I like the idea that when not phosphorylated (i.e. in the cytosol) SPD5 folds back on itself (intramolecular interactions) to limit interactions with other SPD5 molecules (intermolecular interactions) and therefore to limit scaffolding, but when phosphorylated SPD5 unfolds to allow intermolecular interactions that drive scaffolding. For me this is the key finding in the study and advances our understanding of regulated PCM assembly. I therefore feel the paper is ready for publication

Reviewer #3 (Comments to the Authors (Required)):

Having reviewed the revised manuscript by Rios et al., I feel the work is now suitable for publication in JCB. They have addressed all of my prior concerns and in particular have added valuable data concerning how phosphorylation of SPD-5 causes it to transform from a monomeric state to a multimeric one. This work should be a great interest to those in the centrosome field and beyond. In fact anyone interested in how the material properties of cellular constituents is regulated should find this work fascinating. I do still have some minor concerns listed below that should be addressed before publication.

1. Line 226. "AlphaFold modeling predicts that this region...". Would the authors please change "this region" to "the PReM region". They use the terms "this region" or "other regions" so frequently in this paragraph that it is eventually becomes hard to tell what region they are talking about. In the prior sentence they mentioned the CM2-like domain, so it wasn't immediately clear if they were talking about the PReM region or the CM2-like domain.
2. The authors find that the GFP::SPD-5(R592) transgene is not temperature sensitive like the original mutant carrying this same mutation. They speculate that this might be due to the presence of a linked suppressor in the original mutant. This might be true, but it could be due to the lower expression of the transgene relative to the endogenous gene. In figure S3A and in particular S3B, it is clear that the transgenic proteins are expressed at a lower levels than the endogenous protein. So the lack of viability

at all temperatures could be due to the combination of the R592 mutation and lower expression.

3. Line 321 The reference to Figure 3F and S3F should be 3F and S3E.

4. Figure S2A It would be very helpful to highlight in another color those links that are only found in the presence of active PLK-1.

5. Figure S6B Why is expression of the endogenous SPD-5 so low in the 1-566 and 566-1198 transgenic strains relative to the FL?

Reviewer #4 (Comments to the Authors (Required)):

Overall, the authors did an excellent job in addressing my concerns. I was very impressed with the additional work that they put into the crosslinking experiments to be able to analyze SPD-5 monomers in the presence and absence of phosphorylation. The dramatic effect that phosphorylation has on the intra-molecular crosslinks in SPD-5 monomers and the difference in crosslinks between phosphorylated monomers and phosphorylated multimers are both quite striking. The new data and changes have improved the manuscript a lot. However, a few additional textual/figure changes (outlined below), which should be relatively straightforward to incorporate, are still needed to catch the text up to the data and make sure that the results are communicated clearly.

1. It would potentially be useful to include the panel now in Fig. S1A in the main figure to convey how extensively SPD-5 is phosphorylated, which would align well with the dramatic changes in the intra-molecular crosslinks in SPD-5 monomers that are now shown in Fig. 1.

2. An experimental schematic is needed before Figure 1D-F to outline how these experiments are done including the means used to induce SPD-5 phosphorylation/assembly and subsequent disassembly to form the monomer population. From reading the methods, it seems that assembly and phosphorylation are induced in parallel by diluting the purified SPD-5 out of a detergent and high salt-containing storage buffer, which presumably holds the SPD-5 in a monomeric state, into a lower salt, detergent-lacking buffer with PLK1 kinase and ATP and incubating for 2 hours. Then, samples were either cross-linked and processed for Mass spectrometry (multimers) or disassembled by chilling for 45 minutes and then cross-linked (monomers). A flow chart outlining this procedure should be included as a schematic in Fig. 1.

3. The description of the crosslinking experiments in the text is confusing. The main reason for this is that the terms "monomer" and "multimer" are each being used to describe two different things. The first use of the terms "monomer" and "multimer" is to reflect the assembly state of the SPD-5 protein before cross-linking is initiated. SPD-5 was diluted to reduce the salt and detergent concentrations and warmed up with or without PLK1 phosphorylation to form larger multimeric assemblies "multimers". In some cases, reactions were then held on ice to induce disassembly to generate a monomeric assembly state "monomers". The problem is that the authors also use the terms "monomer" and "multimer" to refer to the species on SDS-PAGE gels that are used to identify intra- and inter-molecular cross-links. Regardless of whether they are derived from unassembled or assembled SPD-5, SPD-5 molecules that only have intra-molecular crosslinks after the crosslinking run as "monomers" on an SDS-PAGE gel, whereas species in which 2 or more SPD-5 molecules have been cross-linked to each other run as "multimers". Thus, crosslinks identified from SDS-PAGE monomers are "intra-molecular crosslinks" whereas crosslinks identified from SDS-PAGE multimers are a superposition of inter-molecular and intra-molecular crosslinks. This text in this section needs to be rewritten using different terms for the concepts of SPD-5 in vitro assembly state (for example "unassembled monomers" versus "large multimeric assemblies") and the concept of intra-molecular ("SDS-page monomers") versus combined intra- and inter-molecular ("SDS-page multimers") crosslinks to improve the logical clarity in this section. Also, make sure to use these same terms on Figure 1.

4. "By subtracting out crosslinks that appear in the monomer data sets (i.e., bona fide intramolecular interactions), we identified 269 crosslinks (93% of total) that likely represent intermolecular interactions (Figure S1B)." Related to point 3, I am concerned that the plot in Fig. S1B might be misleading. If I understand correctly, this is an effort to suggest which crosslinks detected in the multimeric assembly state are intermolecular crosslinks by subtracting the crosslinks shown in 1E from the crosslinks shown in 1F. This does not seem like the correct way to do this. Wouldn't identifying the true intermolecular crosslinks require subtracting the SDS-PAGE monomer-derived crosslinks obtained from the multimeric assembly state population from the SDS-PAGE multimer crosslinks obtained from the same multimeric assembly state. The set of intra-molecular crosslinks, like the presence of inter-molecular crosslinks would be expected to differ between the monomeric and multimeric assembly states.

5. The half-triangle summary diagrams for dissecting the outcome of the crosslinking experiments are great because they convey a sense of the reproducibility of the cross-linked pairs, allowing one to ignore potentially spurious crosslinks only seen in 1 or 2 of the 6 samples, which is hard to do in the linear diagrams on the left. A main finding of the analysis comparing Fig. 1E to Figure 1F seems to be that in the multimeric assembly state, but not in the monomeric state, there are a lot of cross-links on the diagonal (which represent either intra- or inter-molecular cross-linking of a residue to a residue near itself in the primary amino acid sequence). This seems like it might represent the formation of homotypic dimeric coiled-coils in these regions (roughly aa 50-300 and aa 750-1050 with small additional regions near aa 620 and 1200). The really surprising thing about this map is how clean it is, you primarily see the appearance of on-diagonal crosslinks in two regions, one in N-terminus and one in the C-

terminus and really very little going on off-diagonal. It would be good for the authors to comment on whether their interpretation of the emergence of on-diagonal crosslinks in these regions is that these regions are forming parallel homotypic coiled-coils (maybe with some additional structure-- like the dimeric hairpin helicies that they describe) within these regions as the molecules form multimeric assemblies. If this is the authors' interpretation, why is this important finding not incorporated into the model in Figure 1G? Is it possible that the N- and C-terminal regions of one SPD-5 molecule could be forming parallel homotypic coiled coils with two different SPD-5 molecules -- if so, maybe this idea could be part of the model in Fig. 1G.

6. More discussion of the key results emerging from this comparison (Fig. 1E to F) is needed. In addition to the emergence of on-diagonal crosslinks that may represent the formation of homotypic coiled-coils, there also seem to be a few high-occurrence off-diagonal longer-distance crosslinks, although surprisingly few. It might be useful to highlight the strongest (most high-confidence) of these off-diagonal crosslinks. For example, looking at the data in the triangular plot there seem to be significant crosslinks between something near aa 220 and the region between aa 30 and 60, between an amino acid near 350 and the region near 140, and one significantly off-diagonal crosslink between an amino acid near 750 and one near 250.

7. "After normalizing for differences in overall cross-link number, we found that phosphorylation of SPD-5 increases its preference for coiled-coil interactions and reduces the number of non-coiled coil interactions (Figure S1E,F)." This is written as though these interactions are general interactions between one predicted coiled-coil region and any of the other coiled-coil regions, whereas visual inspection suggests that the majority of these are crosslinks within a predicted coiled-coil region (suggesting that the crosslink may reflect homotypic coiled-coil formation as discussed above in point 5 or may reflect subsequent interactions between adjacent dimeric coiled-coils.

It seems like it might be good to replace the orange pie charts with something that makes this point. For example, addresses how many of the CC-CC cross-links are within a predicted coiled-coil region or between adjacent coiled-coil regions (potentially reflecting local structure), versus potentially more distant connections. For this, it might be useful to exclude low occurrence crosslinks only observed in 1 or 2 out of 6 samples.

8. It is hard to understand what is being shown in Fig. 1C. From the text on the figure, it seems like crosslinker is being added for varying amounts of time to generate more extensively cross-linked species (SPD-5 multimers). This is consistent with the main text which says "By titrating crosslinker concentrations and incubation times, we could generate SPD-5 species that migrate as monomers or multimers on an SDS-PAGE gel (Figure 1C, see methods)". On the other hand, in the figure legend it says that "Assembly of purified SPD-5 was induced, incubated for 0-60 min, then the cross-linking agent DMTMM was added, which suggests that the assembly reaction is being monitored by adding in x-linker at varying times after the initiation of assembly. Please clarify which of these it is in the figure/text/legend.

9. "R592 lies within the PReM region, close to PLK-1. phosphorylation sites needed for PCM expansion in preparation for mitosis (Ohta et al., 2021; Woodruff et al., 2015). These results indicate the importance of the PReM region for PCM assembly, thus warranting further structural and mechanistic investigation. We identified short-range links within this region and long-range links with other regions of SPD-5 (Figure 1D-F and S2A)." Inspection of the region near aa600 in the triangular plot in Fig. 1F suggests that this region doesn't actually exhibit many long-range interactions. There are 5 crosslinks that originate in the 541-677 region. However, these were all very low occurrence, being observed in only 1 sample out of 6. Rewriting or even deleting most of this transition paragraph might be useful. Given the low occurrence of these crosslinks, is Fig. S2A is worth including?

10. Figure 2F- on this figure the term "dimer unique link" is used. This term is not clear since the assembly state of this construct is always dimeric. Please rephrase this, maybe "likely inter-molecular cross-link"?

11. It would be useful to combine Figures 4 and 5 to allow a direct comparison of this data which looks at the effects on PCM assembly of either disrupting the dimeric hairpin in the aa541-677 region (Fig. 4) or the long coiled-coil between 734 and 918 (Fig. 5). The text describing these two perturbations is quite repetitive between the two sections and suggests that the phenotypes are pretty similar. However, in reality they are quite distinct with respect to severity. The delta 610-640 mutation that disrupts the dimeric hairpin, strongly blocks mitotic PCM assembly, reducing the mass of the mitotic PCM by about 6-fold compared to control. By contrast, disrupting the long coiled-coil has a significantly weaker phenotype, reducing PCM mass by only about 2-fold. In neither case does adding nocodazole really do much to rescue PCM mass, although it does make the mutant PCM more spherical and less prone to fragmentation (which is much more convincing for the large coiled-coil deletion since there is more mitotic PCM there). It would be nice if these sections could be written a more compact way, that would do a better job of bringing out the bigger conclusions.

12. With respect to the GFP::SPD-5 aa 566-1198 phenotype in Fig. 6B. Looking at the images it appears to be pretty similar to the phenotype for the RFP::SPD-5 delta 610-640 phenotype in Fig. 4A. Thus, it is very strange that the quantification is so different between these experiments. For the RFP::SPD-5 delta 610-640 phenotype in (Fig. 4A,B) PCM mass is reduced from ~2 A.U. to 0.35 A.U shown on a linear scale, whereas in Fig. 6C PCM mass is reduced from something like 300,000 A.U. to 10,000 A.U. and is shown on a logarithmic scale. Why are these so different-can they be made more similar?

Minor:

1. "MARCOIL (50% threshold; MTK matrix) predicts that 9 disconnected regions within the alpha helices form coiled-coils (Figure 1B)."

Recommend changing to "have the potential to form coiled-coils" as your data indicate that there are assembly states due to autoinhibition in which they may not be in a coiled-coil state.

2. In the text, be careful in use of the term interaction as crosslinks reflect proximity rather than interaction.

3. "Lines 73-75 "Specifically, regulated assembly of Centrosomin occurs via phosphorylation-driven oligomerization of a central coiled-coil region, termed "PReM", (Conduit et al., 2014a) and its interaction with the C-terminal CM2 domain (Feng et al., 2017)." This should be reworded. The first cited reference (Conduit et al., 2014a) shows that the PReM region is dimeric irrespective of its phosphorylation status. In the second reference, it is proposed that phosphorylation enables an anti-parallel interaction between the PReM coiled-coil and a second dimeric coiled-coil region, the C-terminal CM2 domain of Cnn that is central to the phosphorylation-dependent oligomerization of Cnn.

Dear Arshad,

We have addressed all of the reviewers' remaining concerns. Our responses are listed in blue below. Major text changes are highlighted in the manuscript.

Jeffrey Woodruff

Reviewer #2 (Comments to the Authors (Required)):

The authors have addressed my original concerns and appear to have done a good job at addressing those of the other reviewers. Addressing how PCM proteins are regulated is not easy and I like the combination of techniques used by the authors. Now that they have better explained how they can distinguish intra vs inter molecular crosslinks, the conclusions are solid. Overall, I like the idea that when not phosphorylated (i.e. in the cytosol) SPD5 folds back on itself (intramolecular interactions) to limit interactions with other SPD5 molecules (intermolecular interactions) and therefore to limit scaffolding, but when phosphorylated SPD5 unfolds to allow intermolecular interactions that drive scaffolding. For me this is the key finding in the study and advances our understanding of regulated PCM assembly. I therefore feel the paper is ready for publication

Reviewer #3 (Comments to the Authors (Required)):

Having reviewed the revised manuscript by Rios et al., I feel the work is now suitable for publication in JCB. They have addressed all of my prior concerns and in particular have added valuable data concerning how phosphorylation of SPD-5 causes it to transform from a monomeric state to a multimeric one. This work should be a great interest to those in the centrosome field and beyond. In fact anyone interested in how the material properties of cellular constituents is regulated should find this work fascinating. I do still have some minor concerns listed below that should be addressed before publication.

1. Line 226. "AlphaFold modeling predicts that this region...". Would the authors please change "this region" to "the PReM region". They use the terms "this region" or "other regions" so frequently in this paragraph that it eventually becomes hard to tell what region they are talking about. In the prior sentence they mentioned the CM2-like domain, so it wasn't immediately clear if they were talking about the PReM region or the CM2-like domain.

We have corrected this.

2. The authors find that the GFP::SPD-5(R592) transgene is not temperature sensitive like the original mutant carrying this same mutation. They speculate that this might be due to the presence of a linked suppressor in the original mutant. This might be true, but it could be due to the lower expression of the transgene relative to the endogenous gene. In figure S3A and in particular S3B, it is clear that the transgenic proteins are expressed at a lower levels than the

endogenous protein. So the lack of viability at all temperatures could be due to the combination of the R592 mutation and lower expression.

We included a comment in the text on this point.

“Transgenic SPD-5(R592K) contains a GFP tag and is expressed at slightly lower levels compared to SPD-5(R592K) in *spd-5(or213)* embryos (Figure S3A), which could decrease viability at 16°C.”

We removed the statement about the suppressor, as we have no experimental evidence to support that claim.

3. Line 321 The reference to Figure 3F and S3F should be 3F and S3E.

This has been corrected.

4. Figure S2A It would be very helpful to highlight in another color those links that are only found in the presence of active PLK-1.

Following the advice of Reviewer 4, we decided to remove Figure S2A. The data represented there are already found in Figure 1. In the text, we also point out the long-distance cross-links that appear between the PReM and CM2-like regions that appear after phosphorylation (lines 188-189). Furthermore, we take a closer look at this region in Figure 2 and overlay the short-range cross links onto our structural models. Therefore, we think that Figure S2A is redundant and therefore unnecessary.

5. Figure S6B Why is expression of the endogenous SPD-5 so low in the 1-566 and 566-1198 transgenic strains relative to the FL?

Please note that the antibody is against GFP, so it will only detect the transgenic SPD-5 and not the endogenous version. The faint band at ~130 kDa is non-specific. We now use green arrowheads to indicate the band corresponding to the GFP::SPD-5 fragment being expressed.

Reviewer #4 (Comments to the Authors (Required)):

Overall, the authors did an excellent job in addressing my concerns. I was very impressed with the additional work that they put into the crosslinking experiments to be able to analyze SPD-5 monomers in the presence and absence of phosphorylation. The dramatic effect that phosphorylation has on the intra-molecular crosslinks in SPD-5 monomers and the difference in crosslinks between phosphorylated monomers and phosphorylated multimers are both quite striking. The new data and changes have improved the manuscript a lot. However, a few

additional textual/figure changes (outlined below), which should be relatively straightforward to incorporate, are still needed to catch the text up to the data and make sure that the results are communicated clearly.

1. It would potentially be useful to include the panel now in Fig. S1A in the main figure to convey how extensively SPD-5 is phosphorylated, which would align well with the dramatic changes in the intra-molecular crosslinks in SPD-5 monomers that are now shown in Fig.1.

We have now added this to the main figure.

2. An experimental schematic is needed before Figure 1D-F to outline how these experiments are done including the means used to induce SPD-5 phosphorylation/assembly and subsequent disassembly to form the monomer population. From reading the methods, it seems that assembly and phosphorylation are induced in parallel by diluting the purified SPD-5 out of a detergent and high salt-containing storage buffer, which presumably holds the SPD-5 in a monomeric state, into a lower salt, detergent-lacking buffer with PLK1 kinase and ATP and incubating for 2 hours. Then, samples were either cross-linked and processed for Mass spectrometry (multimers) or disassembled by chilling for 45 minutes and then cross-linked (monomers). A flow chart outlining this procedure should be included as a schematic in Fig. 1.

The revised Figure 1D now outlines our X-linking protocol. Please note that chilling on ice was for 10 min, not 45 min (we may have incorrectly stated this as 45 min in the previous rebuttal letter). The updated figure appears below:

3. The description of the crosslinking experiments in the text is confusing. The main reason for this is that the terms "monomer" and "multimer" are each being used to describe two different things. The first use of the terms "monomer" and "multimer" is to reflect the assembly state of the SPD-5 protein before cross-linking is initiated. SPD-5 was diluted to reduce the salt and detergent concentrations and warmed up with or without PLK1 phosphorylation to form larger multimeric assemblies "multimers". In some cases, reactions were then held on ice to induce disassembly to generate a monomeric assembly state "monomers". The problem is that the authors also use the terms "monomer" and "multimer" to refer to the species on SDS-PAGE gels that are used to identify intra- and inter-molecular cross-links. Regardless of whether they are

derived from unassembled or assembled SPD-5, SPD-5 molecules that only have intra-molecular crosslinks after the crosslinking run as "monomers" on an SDS-PAGE gel, whereas species in which 2 or more SPD-5 molecules have been cross-linked to each other run as "multimers". Thus, crosslinks identified from SDS-PAGE monomers are "intra-molecular crosslinks" whereas crosslinks identified from SDS-PAGE multimers are a superposition of inter-molecular and intra-molecular crosslinks. This text in this section needs to be rewritten using different terms for the concepts of SPD-5 in vitro assembly state (for example "unassembled monomers" versus "large multimeric assemblies") and the concept of intra-molecular ("SDS-page monomers") versus combined intra- and inter-molecular ("SDS-page multimers") crosslinks to improve the logical clarity in this section. Also, make sure to use these same terms on Figure 1.

We agree completely. While we titrated our crosslinking conditions (concentration and incubation time) to capture most of the complexes, by chance a protein in a complex may not be crosslinked to its partner(s) and will run as a monomer by SDS-PAGE.

Therefore, we now call our samples "SDS-PAGE monomers" and "SDS-PAGE multimers". We include a statement explaining this designation in the main text:

"When separated by SDS-PAGE, crosslinked samples migrated as monomeric or multimeric species (MW = 135 kDa vs. multiples thereof) which could be excised and analyzed separately by MS (Figure 1D). It is possible that a protein in a complex was not crosslinked to its binding partners, in which case it would run as a monomer on an SDS-PAGE gel. Thus, we refer to our samples as "SDS-PAGE monomers" and "SDS-PAGE multimers". The SDS-PAGE monomers contain exclusively intramolecular interactions, while the SDS-PAGE multimers contain both intra- and intermolecular interactions."

4. "By subtracting out crosslinks that appear in the monomer data sets (i.e., bona fide intramolecular interactions), we identified 269 crosslinks (93% of total) that likely represent intermolecular interactions (Figure S1B)." Related to point 3, I am concerned that the plot in Fig. S1B might be misleading. If I understand correctly, this is an effort to suggest which crosslinks detected in the multimeric assembly state are intermolecular crosslinks by subtracting the crosslinks shown in 1E from the crosslinks shown in 1F. This does not seem like the correct way to do this. Wouldn't identifying the true intermolecular crosslinks require subtracting the SDS-PAGE monomer-derived crosslinks obtained from the multimeric assembly state population from the SDS-PAGE multimer crosslinks obtained from the same multimeric assembly state. The set of intra-molecular crosslinks, like the presence of inter-molecular crosslinks would be expected to differ between the monomeric and multimeric assembly states.

We did something close to what the reviewer had suggested. We realize that this is unclear because we did not properly explain our X-linking approach in the text. Also, there was some disagreement between the main text and the methods section, which we have fixed.

As shown in Figure 1D, we assembled SPD-5 scaffolds for 2 hr in the presence of PLK-1 (CA or KD) for all reactions. This was needed to ensure full phosphorylation of SPD-5. Then, we incubated one sample on ice for 10 min so that the SPD-5 scaffolds would partly disassemble, then added the x-linker. We did this in the hope that we could enrich for the SDS-PAGE monomers. Thus, these SDS-PAGE monomers are derived from a multimeric sample.

Please keep in mind that our determination of inter-molecular interactions is an estimate. The best way to identify bona fide inter-molecular interactions is to use a mixed population of isotope-labeled and unlabeled protein. However, this experiment is extremely difficult for a large protein. The isotope label (which shifts the masses of all peptides) dramatically increases the complexity of the computational search space during our analysis. We can do this, but the false discovery rate skyrockets, and the data become meaningless. We strongly feel that improving this procedure is beyond the scope of the current paper.

5. The half-triangle summary diagrams for dissecting the outcome of the crosslinking experiments are great because they convey a sense of the reproducibility of the cross-linked pairs, allowing one to ignore potentially spurious crosslinks only seen in 1 or 2 of the 6 samples, which is hard to do in the linear diagrams on the left. A main finding of the analysis comparing Fig. 1E to Figure 1F seems to be that in the multimeric assembly state, but not in the monomeric state, there are a lot of cross-links on the diagonal (which represent either intra- or inter-molecular cross-linking of a residue to a residue near itself in the primary amino acid sequence). This seems like it might represent the formation of homotypic dimeric coiled-coils in these regions (roughly aa 50-300 and aa 750-1050 with small additional regions near aa 620 and 1200). The really surprising thing about this map is how clean it is, you primarily see the appearance of on-diagonal crosslinks in two regions, one in N-terminus and one in the C-terminus and really very little going on off-diagonal. It would be good for the authors to comment on whether their interpretation of the emergence of on-diagonal crosslinks in these regions is that these regions are forming parallel homotypic coiled-coils (maybe with some additional structure-- like the dimeric hairpin helices that they describe) within these regions as the molecules form multimeric assemblies. If this is the authors' interpretation, why is this important finding not incorporated into the model in Figure 1G? Is it possible that the N- and C-terminal regions of one SPD-5 molecule could be forming parallel homotypic coiled coils with two different SPD-5 molecules -- if so, maybe this idea could be part of the model in Fig. 1G.

Yes, it is possible. The short-distance cross-links (those near the diagonal) could indicate parallel homotypic coiled-coils. We speculate on this point in the revised discussion. But, more in-depth structural modeling is needed to say for sure. It is possible that small, disordered linker regions could exist within a coiled-coil domain, allowing it to turn. For example, AlphaFold models disagree on the continuity of the first coiled-coil domain, as some insist that the helix is broken by a small turn. Therefore, we are not comfortable with stating if these coiled-coils are parallel or anti-parallel without extensive follow-up experiments. We make a statement in the Discussion on this point.

In the results, we now comment on the prevalence of short-distance cross-links (on diagonal) as evidence of homotypic interactions and mention the long-distance (off diagonal) cross-links as evidence of heterotypic interactions. We state that 72% of CC-CC links are between the same coiled-coil. The model is updated accordingly to display both kinds of interactions.

6. More discussion of the key results emerging from this comparison (Fig. 1E to F) is needed. In addition to the emergence of on-diagonal crosslinks that may represent the formation of homotypic coiled-coils, there also seem to be a few high-occurrence off-diagonal longer-distance crosslinks, although surprisingly few. It might be useful to highlight the strongest (most high-confidence) of these off-diagonal crosslinks. For example, looking at the data in the triangular plot there seem to be significant crosslinks between something near aa 220 and the region between aa 30 and 60, between an amino acid near 350 and the region near 140, and one significantly off-diagonal crosslink between an amino acid near 750 and one near 250.

We agree and have updated the text accordingly:

“We noticed that most linked residues were close in sequence space, appearing near the diagonal on our occurrence plot (middle panel, Figure 1G). Of all crosslinks between coiled-coil domains, 72% were between the same domains, suggesting that coiled-coil domains in SPD-5 can homo-multimerize. There were also several long-range crosslinks, the most prominent being K35-235E, K126-D350, and E247-K760 (occurrence >4), suggesting that certain coiled-coil domains in SPD-5 can also hetero-multimerize.”

7. "After normalizing for differences in overall cross-link number, we found that phosphorylation of SPD-5 increases its preference for coiled-coil interactions and reduces the number of non-coiled coil interactions (Figure S1E,F)." This is written as though these interactions are general interactions between one predicted coiled-coil region and any of the other coiled-coil regions, whereas visual inspection suggests that the majority of these are crosslinks within a predicted coiled-coil region (suggesting that the crosslink may reflect homotypic coiled-coil formation as discussed above in point 5 or may reflect subsequent interactions between adjacent dimeric coiled-coils.

It seems like it might be good to replace the orange pie charts with something that makes this point. For example, addresses how many of the CC-CC cross-links are within a predicted coiled-coil region or between adjacent coiled-coil regions (potentially reflecting local structure), versus potentially more distant connections. For this, it might be useful to exclude low occurrence crosslinks only observed in 1 or 2 out of 6 samples.

We agree that it is important to highlight homotypic vs. heterotypic crosslinks. We quantified this, essentially using occurrence as a weight so that low-occurrence links don't skew the data. We now state in the text:

“Of all crosslinks between coiled-coil domains, 72% were between the same domains,”

In addition, in Figure S1G, we report the distribution of x-link distances, which is an unbiased way to estimate the degree of local vs. long-distance interactions.

We still believe the current pie charts are useful to illustrate how x-links are largely between CC regions. These data support our main conclusion that coiled-coil-based interactions drive SPD-5 assembly. This is an especially important point to make for the condensate/phase separation community, who have largely espoused IDRs as the main drivers of super-stoichiometric assemblies.

8. It is hard to understand what is being shown in Fig. 1C. From the text on the figure, it seems like crosslinker is being added for varying amounts of time to generate more extensively cross-linked species (SPD-5 multimers). This is consistent with the main text which says "By titrating crosslinker concentrations and incubation times, we could generate SPD-5 species that migrate as monomers or multimers on an SDS-PAGE gel (Figure 1C, see methods)". On the other hand, in the figure legend it says that "Assembly of purified SPD-5 was induced, incubated for 0-60 min, then the cross-linking agent DMTMM was added, which suggests that the assembly reaction is being monitored by adding in x-linker at varying times after the initiation of assembly. Please clarify which of these it is in the figure/text/legend.

The old Figure 1C was meant to illustrate the crosslinked species. We now see that it is misleading. For our XL-MS data, we always use the same x-linking conditions. We have now updated Figure 1 to include a flow chart of our x-linking protocol and a representative gel that would be used for gel band excision. The text has been corrected accordingly.

9. "R592 lies within the PReM region, close to PLK-1 phosphorylation sites needed for PCM expansion in preparation for mitosis (Ohta et al., 2021; Woodruff et al., 2015). These results indicate the importance of the PReM region for PCM assembly, thus warranting further structural and mechanistic investigation. We identified short-range links within this region and long-range links with other regions of SPD-5 (Figure 1D-F and S2A)." Inspection of the region near aa600 in the triangular plot in Fig. 1F suggests that this region doesn't actually exhibit many long-range interactions. There are 5 crosslinks that originate in the 541-677 region. However, these were all very low occurrence, being observed in only 1 sample out of 6. Rewriting or even deleting most of this transition paragraph might be useful. Given the low occurrence of these crosslinks, is Fig. S2A is worth including?

We agree with the reviewer that this statement in the transition is not useful. We have modified this text and removed Figure 2A.

10. Figure 2F- on this figure the term "dimer unique link" is used. This term is not clear since the assembly state of this construct is always dimeric. Please rephrase this, maybe "likely inter-molecular cross-link"?

This has been corrected.

11. It would be useful to combine Figures 4 and 5 to allow a direct comparison of this data which looks at the effects on PCM assembly of either disrupting the dimeric hairpin in the aa541-677 region (Fig. 4) or the long coiled-coil between 734 and 918 (Fig. 5). The text describing these two perturbations is quite repetitive between the two sections and suggests that the phenotypes are pretty similar. However, in reality they are quite distinct with respect to severity. The delta 610-640 mutation that disrupts the dimeric hairpin, strongly blocks mitotic PCM assembly, reducing the mass of the mitotic PCM by about 6-fold compared to control. By contrast, disrupting the long coiled-coil has a significantly weaker phenotype, reducing PCM mass by only about 2-fold. In neither case does adding nocodazole really do much to rescue PCM mass, although it does make the mutant PCM more spherical and less prone to fragmentation (which is much more convincing for the large coiled-coil deletion since there is more mitotic PCM there). It would be nice if these sections could be written a more compact way, that would do a better job of bringing out the bigger conclusions.

We have streamlined these two sections and now emphasize the difference in severity of phenotypes, which suggests that CC-long is not as important as the Helical Hairpin. This further supports that interaction domains are not equivalent, but rather hierarchical. However, we believe that Fig. 4 (Hairpin helix mutation) and Fig. 5 (CC-long mutation) are best displayed as separate figures.

12. With respect to the GFP::SPD-5 aa 566-1198 phenotype in Fig. 6B. Looking at the images it appears to be pretty similar to the phenotype for the RFP::SPD-5 delta 610-640 phenotype in Fig. 4A. Thus, it is very strange that the quantification is so different between these experiments. For the RFP::SPD-5 delta 610-640 phenotype in (Fig. 4A,B) PCM mass is reduced from ~2 A.U. to 0.35 A.U shown on a linear scale, whereas in Fig. 6C PCM mass is reduced from something like 300,000 A.U. to 10,000 A.U. and is shown on a logarithmic scale. Why are these so different-can they be made more similar?

We have corrected this to make all results more comparable. In addition, we now show Figure 6C on a linear scale.

For all experiments, the integrated density values were in the 100,000's range. Since the unit is arbitrary, it is scalable; so, for the ease of reading the values, we eliminated the zeros, basically dividing each number by 100,000. We had not done this type of scaling for Fig. 6C in the previous draft. All integrated density values are now scaled this way.

Minor:

1. "MARCOIL (50% threshold; MTK matrix) predicts that 9 disconnected regions within the alpha helices form coiled-coils (Figure 1B)."

Recommend changing to "have the potential to form coiled-coils" as your data indicate that

there are assembly states due to autoinhibition in which they may not be in a coiled-coil state.

We have corrected this.

2. In the text, be careful in use of the term interaction as crosslinks reflect proximity rather than interaction.

We agree and have corrected this.

3. "Lines 73-75 "Specifically, regulated assembly of Centrosomin occurs via phosphorylation-driven oligomerization of a central coiled-coil region, termed "PReM", (Conduit et al., 2014a) and its interaction with the C-terminal CM2 domain (Feng et al., 2017)." This should be reworded. The first cited reference (Conduit et al., 2014a) shows that the PReM region is dimeric irrespective of its phosphorylation status. In the second reference, it is proposed that phosphorylation enables an anti-parallel interaction between the PReM coiled-coil and a second dimeric coiled-coil region, the C-terminal CM2 domain of Cnn that is central to the phosphorylation-dependent oligomerization of Cnn.

We have re-written this section to be more specific about the stoichiometry of PReM domain and how it multimerizes after Plk1 phosphorylation:

"Cnn contains a central coiled-coil region, termed "PReM", that forms dimers in isolation. Upon phosphorylation by PLK-1, PReM forms homo-pentamers (Conduit et al., 2014a) or anti-parallel, hetero-tetramers with the C-terminal CM2 domain (Feng et al., 2017)."